# Reducing methylation of histone 3.3 lysine 4 in the medial ganglionic eminence and hypothalamus recapitulates neurodevelopmental disorder phenotypes

Jianing Li [1], Anthony F. Tanzillo[1], Giusy Pizzirusso [2,3], Adam Caccavano [2], Ramesh Chittajallu[2], Mira Sohn [4], Daniel Abebe[2], Yajun Zhang[1], Kenneth A. Pelkey [2], Ryan K. Dale [4], Chris J. McBain [2] & Timothy J. Petros [1] ✉

Methylation of lysine 4 on histone H3 (H3K4) is enriched on active promoters and enhancers where it promotes gene activation. Disruption of H3K4 methylation is associated with numerous neurodevelopmental diseases (NDDs) that display intellectual disability and abnormal body growth. Here, we perturb H3K4 methylation in the medial ganglionic eminence (MGE) and hypothalamus, two brain regions associated with these disease phenotypes. These mutant mice have fewer forebrain interneurons, deficient network rhythmogenesis, and increased spontaneous seizures and seizure suscept-ibility. Mutant mice are significantly smaller than control littermates, but they eventually became obese due to striking changes in the genetic and cellular hypothalamus environment in these mice. Perturbation of H3K4 methylation in these cells produces deficits in numerous NDD-associated behaviors, with a bias for more severe phenotypes in female mice. Single nuclei sequencing reveals transcriptional changes in the embryonic and adult brain that underlie many of these phenotypes. In sum, our findings highlight the critical role of H3K4 methylation in regulating survival and cell-specific gene regulatory mechanisms in forebrain GABAergic and hypothalamic cells during neurode-velopment to control network excitability and body size homoeostasis.

Chromatin remodelers and epigenetic-modifying enzymes regulate proper gene expression throughout development. There are ~300 human genes coding for proteins that drive epigenetic modifications, and these genes are highly intolerant to genetic variation[1]. Predictably, heterozygous loss-of-function variants in these genes leads to numerous neurodevelopmental diseases (NDDs)[2], with many of these NDDs displaying a co-occurrence of intellectual disability (ID), epilepsy, and abnormal body growth[3,4].

[1]Section on Cellular and Molecular Neurodevelopment, Eunice Kennedy Shriver National Institute of Child Health and Human Development (NICHD), National Institutes of Health (NIH), Bethesda, MD, USA. [2]Section on Cellular and Synaptic Physiology, Eunice Kennedy Shriver National Institute of Child Health and Human Development (NICHD), National Institutes of Health (NIH), Bethesda, MD, USA. [3]Department of Neurobiology, Care Sciences and Society, Division of Neurogeriatrics, & Department of Women's and Children's Health, Karolinska Institutet, Stockholm, Sweden. [4]Bioinformatics and Scientific Programming Core, Eunice Kennedy Shriver National Institute of Child Health and Human Development (NICHD), National Institutes of Health (NIH), Bethesda, MD, USA. ✉e-mail: tim.petros@nih.gov

Many epigenetic machinery proteins target post-translational modifications of histones that are highly dynamic and required for proper transcriptional activation and repression[5,6]. One critical modification is methylation of histone H3 lysine 4 (H3K4), with trimethylation of H3K4 (H3K4me3) strongly correlating with promoter activation and gene expression. Dysregulation of H3K4me is associated with numerous NDDs such as autism spectrum disorder (ASD), schizophrenia and substance-related disorders[7–9]. Canonical histones H3.1 and H3.2 are enriched in fetal mouse and human brains and decrease over time, with histone H3 variant H3.3 becoming the predominant histone H3 in postmitotic neurons through adulthood[10–12]. H3.3 is encoded by 2 genes, *H3f3a* and *H3f3b*. Removing one or both genes in mice leads to variable survivability and phenotypes depending on genetic targeting strategy, compensation and onset of gene loss[13]. Notably, removal of both genes in the dorsal forebrain results in lethality several hours after birth[12], complicating assessment of H3K4me function throughout neurodevelopment.

There are 6 members of the lysine-specific methyltransferase 2 (KMT2) family that methylate H3K4 in mammals (*KMT2A-D & F-G*), and 7 histone lysine demethylases that target H3K4me (*KDM1A-B, KDM2B & KDM5A-D*)[9,14]. Patients with variants in *KMT2A* (Wiedemann-Steiner syndrome), *KMT2C* (Kleefstra syndrome) and *KMT2D* (Kabuki syndrome) display ID, microcephaly, epilepsy and short stature[6,15–18]. Variants in demethylases *KDM5A-C* are associated with ID and ASD[9], indicating a delicate balance H3K4 methylation is critical for normal brain development.

The comorbidities of epilepsy, ID and abnormal body growth in these H3K4me-associated diseases are indicative of altered excitatory-inhibitory balance and hypothalamic defects. Several studies have examined decreased H3K4me in excitatory cells[19–23], but disruption of H3K4me in forebrain inhibitory interneurons and hypothalamic cells has not been explored. Interneurons are a highly diverse cell population that arise from two transient structures in the embryonic ventral forebrain, the medial and caudal ganglionic eminences (MGE and CGE, respectively)[24–26]. Abnormal development and function of interneurons has been linked to the pathobiology of NDDs such as schizophrenia, autism and epilepsy[27–30], and many NDD-associated genes are enriched in interneuron progenitors[31–37].

Here, we have reduced H3K4me specifically in the MGE and hypothalamus, two cell populations associated with common phenotypes of H3K4me dysregulation such as ID, epilepsy and control of body growth. Our results reveal significant genetic and cellular changes of MGE-derived interneurons and hypothalamic cells, leading to behavioral phenotypes associated with NDDs. In many assays, female mice display greater deficits compared to males. Electrophysiology recordings confirmed altered intrinsic properties of hippocampal interneurons and reduced network synchrony in mutant mice. In sum, our multimodal analyses reveal that reduction of H3K4 methylation leads to similar and distinct changes in forebrain interneurons and hypothalamic cells that, in part, underlie a variety of complex phenotypes related to NDDs.

## Results

### Disruption of H3K4 methylation in the brain alters viability and body size

To manipulate H3K4me, we used the conditional transgenic mouse line *LSL-K4M* with a floxed-stop cassette followed by an HA-tagged human H3.3 sequence where the lysine 4 residue is mutated to a methionine (*H3.3K4M*)[38], thus disrupting normal H3K4 methylation (Fig. 1A). The transcription factor *Nkx2.1* is a 'master regulator' for all MGE-derived interneurons[39,40] and is also expressed in the developing hypothalamus[41] where it is critical for nuclei involved in regulating appetite and energy homeostasis[42–44]. In addition to neurons, Nkx2.1+ progenitors give rise to astrocytes and oligodendrocytes in the MGE, and oligodendrocytes, astrocytes and tanycytes in the hypothalamus[45].

Thus, *Nkx2.1-Cre* mice allow us to target two brain regions critical for the most common H3K4 methylation disease-related phenotypes: epilepsy and altered body growth.

We generated *Nkx2.1-Cre;H3.3K4M;Ai9* wild-type, heterozygous and homozygous mice (H3.3K4M WT, Het and Hom, respectively) to drive expression of H3.3K4M protein in the MGE and hypothalamus, with Nkx2.1-lineage cells expressing the red fluorescent reporter tdTomato. These H3.3K4M Hom mice contain both endogenous WT H3.3 genes (*h3f3a* and *h3f3b*) and the exogenous mutant H3.3K4M alleles, thus mimicking heterozygous disease-associated variants in humans with reduced H3K4 methylation (Fig. 1B). We confirmed strong expression of HA-tagged H3.3K4M protein throughout the ventricular zone (VZ), subventricular zone (SVZ) and mantle of the MGE and hypothalamus, and a corresponding reduction of H3K4me1, H3K4me2 and H3K4me3 in E13.5 H3.3K4M Hom MGE via immunohistochemistry (Fig. 1C, D). The strongest reduction of H3K4 methylation occurs in the mantle, where the H3.3 variant is actively replacing canonical histones H3.1 and H3.2 in postmitotic cells[12]. Western blots demonstrated a significant reduction of H3K4me3 in the E13.5 MGE and hypothalamus of H3.3K4M Hom mice, with a moderate reduction in H3.3K4M Het mice (Fig. 1E). Having H3.3K4M active prior to cell cycle exit is critical to substitute endogenous H3.3 with mutant H3.3K4M protein due to the prolonged half-life of histone proteins in the brain[46,47]. We do not observe any significant difference in total histone H3.3 levels in the E13.5 MGE or hypothalamus of mutant mice (Supplementary Fig. 1A).

Male and female H3.3K4M Hom mice were significantly smaller than WT mice during the first 6 postnatal weeks, with ~25% of Hom mice dying during this period (Fig. 2A–C and Supplementary Fig. 1B). H3.3K4M Het mice were similar size as WT littermates, but ~20% of Het mice also died by 6 weeks. H3.3K4M Hom mice brains weighed less, had larger lateral ventricles and had a moderate but significant reduction in cortical thickness, particularly layer II–IV (Fig. 2D–F and Supplementary Fig. 1C, D). While H3.3K4M Het and Hom mice continued to die over the next couple months (~55% Hom and ~30% Het die by 20 weeks), the survivors eventually surpassed the size of WT mice. From postnatal weeks 6–9, H3.3K4M Hom mice consumed more food and gained more weight compared to WT mice (Fig. 2G). By 20 weeks, male and female H3.3K4M Het and Hom mice weighed significantly more than WT mice with higher fat weight and serum leptin levels (Fig. 2H–J and Supplementary Fig. 1E). This trend continued up to 1 year old mice where H3.3K4M Hom mice weighed ~35% more, had > 4-fold more fat weight, and ~2.5-fold higher serum leptin levels compared to WT (Fig. 2K–M). Thus, perturbation of H3K4 methylation in the MGE and/or hypothalamus increases postnatal lethality and causes significant changes in brain size and body growth, likely due in part to increased food consumption and leptin levels.

### Decreased numbers and altered function of MGE-derived interneurons leading to increased seizure susceptibility

Some H3.3K4M Hom mice had spontaneous seizures suggesting a deficit in normal interneuron function. The MGE gives rise to nearly all parvalbumin- and somatostatin-expressing interneurons (PV+ and SST+, respectively) in the forebrain, as well as a population of neuronal nitric oxide synthase-expressing (nNos+) neurogliaform and ivy cells in the hippocampus[24,25]. We observed a significant decrease in the density of MGE-derived cortical interneurons in the P21 cortex of Het and Hom mice, with the strongest effect on PV+ cells (Fig. 3A, B). This decrease in PV+ cells was present in both the deep and superficial layers (Supplementary Fig. 2). Similar results were observed in the hippocampus, with a ~40% reduction in MGE-derived interneurons affecting PV+, SST+ and nNos+ subtypes (Fig. 3C, D). Similar changes in interneuron densities and percentages were observed in male and female mice. We also detected a population of MGE-derived Tom+/Olig2+ oligodendrocytes in CA3 as previously described[48]. These

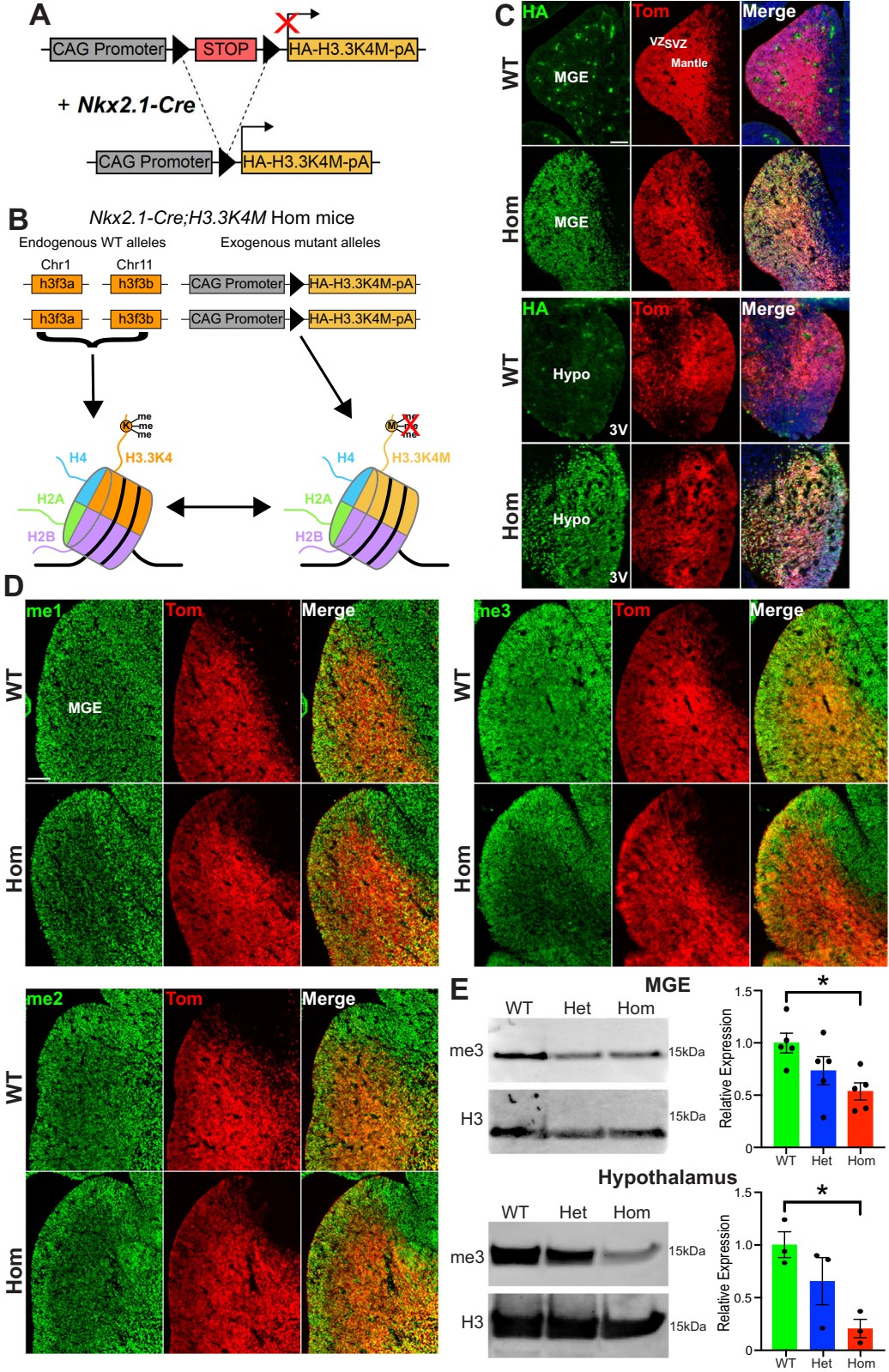

**Fig. 1 | Downregulation of H3K4 methylation in the H3.3K4M mouse. A, B** *LSL-K4M* transgene (**A**) and expression of both WT H3.3 and mutant H3.3K4M proteins in homozygous mice (**B**). **C** Upregulation of HA-tagged H3.3K4M protein in VZ, SVZ and postmitotic cells in the MGE (top) and hypothalamus (bottom) of E13.5 *Nkx2.1-Cre;H3.3K4M;Ai9* WT and Hom mice. 3V = 3rd ventricle. **D** Decrease of H3K4me1, H3K4me2 and H3K4me3 (me1, me2, me3, respectively) in the MGE of H3.3K4M Hom mice. **E** Western blots from combined male & female mice of H3K4me3 levels in the E13.5 MGE ($n = 5$ for each genotype) and hypothalamus ($n = 3$ for each genotype) with quantification. Scale bars = 100 μm. Data are presented as mean values +/− SEM. All stats are one-way ANOVA followed by Tukey's multiple comparison tests: * = $p \leq .05$. Source data are provided as a Source Data file.

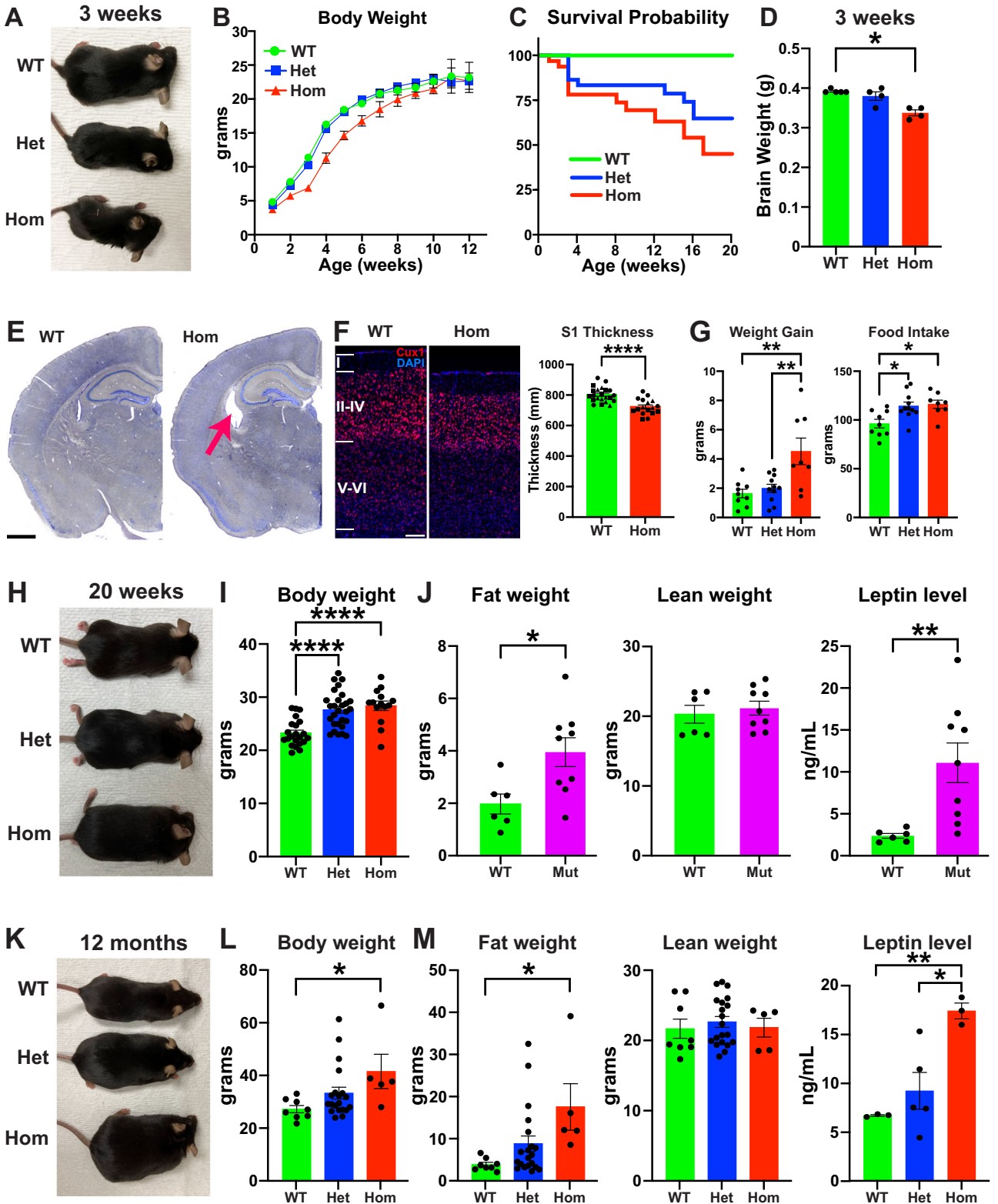

Olig2+ oligodendrocytes were almost completely lost in the P21 H3.3K4M Hom mice, and we observed an apparent corresponding decrease in myelin basic protein (MBP) in CA3 (Supplementary Fig. 3). Thus, disruption of H3K4 methylation in the MGE reduces interneurons in the cortex and hippocampus.

To specifically target H3.3K4M expression to MGE cells, we tried generating *Lhx6-Cre;H3.3K4M* Hom mice, as Lhx6 is expressed in

postmitotic MGE-derived interneurons and is critical for their development[49]. We never obtained *Lhx6-Cre;H3.3K4M* Hom offspring, and while most Het mice also died during embryogenesis or shortly after birth, several survived for one month. These *Lhx6-Cre;H3.3K4M* Het mice were smaller than WT littermates, their brains weighed significantly less, and there was a trend towards decreased PV+ and SST+ hippocampal interneurons that is similar to *Nkx2.1-Cre;H3.3K4M* Het

**Fig. 2 | Disruption of H3K4 methylation alters viability and body size.** Data is combined male & female samples. **A**–**D** 3-week-old H3.3K4M WT, Het and Hom mice (**A**) with quantification of body weight (**B**), survival probability (**C**) and brain weight (**D**). Cumulative number for body weight: 151 WT, 270 Het and 105 Hom mice. Brain weight: 5 WT, 4 Het and 4 Hom mice. **E** Nissl staining reveals enlarged lateral ventricles (red arrow) in 3-week-old H3.3K4M Hom mice. Scale bar = 1 mm. **F** Sections through S1 stained with Cux1 to label layers II-IV (left), with reduced S1 thickness in H3.3K4M Hom mice (right). Scale bar = 100 μm. WT: $n$ = 22 slices, Hom: $n$ = 17 slices, $n$ = 3 brains per genotype, sections from each brain labeled with a different shape. **G** Weight gain (left) and food intake (right) from postnatal weeks 6–9 ($n$ = 9 WT, 11 Het and 8 Hom mice). **H, I** 20-week-old H3.3K4M WT, Het and Hom mice (**H**) with quantification of body weight ($n$ = 22 WT, 28 Het and 14 Hom) (**I**). **J** Fat weight, lean weight and leptin levels of 20-week-old mice ($n$ = 6 WT and 9 Mut mice). **K, L** 12-month-old H3.3K4M WT, Het and Hom mice (**K**) with quantification of body weight ($n$ = 8 WT, 20 Het and 5 Hom mice) (**L**). **M** Fat weight, lean weight, and leptin levels of 12-month-old mice. Fat weight and lean weight: $n$ = 8 WT, 20 Het and 5 Hom mice. Leptin level: $n$ = 3 WT, 5 Het and 3 Hom mice. Data are presented as mean values +/− SEM. Kruskal-Wallis followed by Dunn's multiple comparisons (**D**) or one-way/two-way ANOVA followed by Tukey's multiple comparison tests were performed based on normal distribution (**B, G, I, L, M**). Standard Unpaired 2-tailed $t$ test (**F, J**: fat weight & lean weight) or 2-tailed Mann-Whitney test (**J**: leptin level) based on normal distribution when only WT and Hom mice tested: * = $p \le .05$, ** = $p \le .005$, **** = $p \le .0001$. Source data are provided as a Source Data file.

mice (Supplementary Fig. 4). The lethality and inefficiency of generating *Lhx6-Cre;H3.3K4M* mutant mice prohibited further analysis.

Some P10-P14 H3.3K4M Hom mice displayed robust spasm-like behaviors (e.g., rapid, full flexions and extensions of limbs, trunk flexion, trunk curling and back arching) (Fig. 4A). This seizure activity was also observed in 3-month-old H3.3K4M Hom mice, which exhibit head nodding, continuous whole-body myoclonus, rearing tonic seizure and tonic-clonic seizure with wild jumping (Fig. 4A and Supplementary Video 1). This behavior was strongly sex-biased, with 70% of H3.3K4M Hom female mice displaying spontaneous seizures while only ~14% of Hom males showed this behavior (Fig. 4B). We did not observe spontaneous seizure-like behavior in H3.3K4M Het mice.

To assess seizure susceptibility, we induced seizures with a single injection of Pentylenetetrazole (PTZ) at 5 months. We used a dose of 20 mg/kg, ~50% of the typical effective concentration[50] because some Hom mice already display spontaneous seizures. While both male and female 5-month-old H3.3K4M Hom mice displayed increased seizure susceptibility compared to WT, female Hom mice had more severe seizures, with 60% not recovering, while no males died (Fig. 4B and Supplementary Fig. 5A). Since Hom mice are already obese at 5 months, we also performed the PTZ injections at 3 months when WT and Hom mice are equal size. Since female mice are more sensitive to PTZ-induced seizures[51], we increased the concentration of PTZ (40 mg/kg) for male mice. Both male and female Hom mice displayed increased seizure susceptibility at 3 months, with the increased dose in males generating similar results to female mice (Fig. 4B, C).

The temporal lobe, particularly the hippocampus, is the most common epileptic locus, and that is where we observe the greatest interneuron loss (Fig. 3). To assess the function of MGE-derived interneurons, we performed whole-cell patch-clamp recordings on Tom+ cells in ex vivo hippocampal slices from 8-week-old female WT and H3.3K4M Hom mice. Quantifying six intrinsic properties (input resistance, Tau, capacitance, Sag ratio, maximum firing rate, AP half-width) was sufficient to define the three expected interneuron cell types via unbiased hierarchical clustering and K-means clustering (Supplementary Fig. 5B–D). Consistent with our immunohistochemical findings, putative PV+/fast-spiking (FS), SST+/non-FS (NFS), and nNos+/slow-spiking (SS) interneurons were observed in H3.3K4M Hom mice with properties largely similar to those in WT mice. However, at the population level, all three interneuron subtypes in H3.3K4M Hom mice showed greater variability in intrinsic properties compared to WT, leading to less refined clustering on the PCA maps (Supplementary Fig. 5D). This increased variance was statistically significant in some instances, such as the firing rate and capacitance in FS cells (Supplementary Fig. 5E).

Cortical circuit information coding requires precision in the timing, extent and synchrony of activity within glutamatergic principal cell assemblies that is largely orchestrated by local circuit interneurons. Brain oscillations in the gamma-frequency band are critical for higher cognitive function and critically rely on balanced phasic excitatory and inhibitory drive[52–56]. To examine if the interneuron deficits observed in H3.3K4M Hom mice disrupt network rhythmogenesis, we probed kainate (KA)-induced gamma oscillations in ex vivo hippocampal slices from 3-month-old male and female mice (Fig. 4D, E). Gamma oscillations were detected in 77% of slices from WT mice but only 54% of slices from H3.3K4M Hom mice (Fig. 4F), suggesting a reduced propensity for physiological network rhythmogenesis in H3.3K4M Hom mice. Slices from H3.3K4M Hom mice displayed significantly decreased gamma power and peak frequency compared to WT mice (Fig. 4E–H), indicating that H3.3K4M slices produce smaller and slower oscillations. In addition, multiple slices from H3.3K4M mice developed KA-induced interictal epileptiform (IE) events, which were not observed in any slices from WT mice (Fig. 4F, I, J). This finding suggests unrestrained network hyperexcitability in H3.3KM Hom mice, consistent with the increased seizure susceptibility observed in vivo (Fig. 4C and Supplementary Fig. 5A).

In sum, reducing H3K4 methylation decreases the number of MGE-derived interneurons, alters their intrinsic properties, and disrupts normal network activity, leading to spontaneous seizures and increased seizure susceptibility; this is similar to temporal lobe epilepsy (TLE) patients and animal models of TLE[57–60].

## Disrupted tangential migration of interneurons in H3.3K4M Hom mice

To identify mechanisms underlying reduced interneuron numbers, we explored changes in cell cycle dynamics, interneuron migration and programmed cell death. At E13.5, we observed no differences in the M-phase marker phospho-histone 3 (PH3) or the proliferative marker Ki-67 in the MGE between H3.3K4M WT and mutant embryos (Fig. 5A, B). However, there was a significant reduction in the number MGE-derived Tom+ interneurons migrating into the cortex in H3.3K4M Het and Hom mice (Fig. 5A–C). To determine if migration was delayed or reduced, we examined the cortex of P4 mice, before the period when ~20–40% of cortical interneurons undergo programmed apoptosis from P5–P12[61–63]. A striking reduction of Tom+ cells is still present in the hippocampus of H3.3K4M Hom mice at P4 (Fig. 5D), indicating that tangential migration is not simply delayed but instead strongly reduced in mutant mice. We also examined apoptotic cells in the E13.5 MGE and P4 cortex. Very fewer TUNEL+ apoptotic cells were detected at both ages, and there was no obvious difference between genotypes (Fig. 5E). Thus, the significant reduction of forebrain interneurons in H3.3K4M mutant mice arises primarily from decreased tangential migration of MGE-derived interneurons.

## Increased anxiety and impaired locomotor activity in H3.3K4M Hom mice

We performed a series of behavior tests on juvenile (5–11 weeks) or adult (3–5 months) H3.3K4M WT and mutant mice (Supplementary Fig. 6A) to determine if disruption of H3K4 methylation in the MGE and/or hypothalamus induces behavioral deficits associated with NDDs. To assess anxiety behaviors, we characterized mice in the open field and light-dark box tests. Both female and male H3.3K4M Hom mice spent significantly less time in the center zone of the open field compared WT mice (Fig. 6A, B). Female Hom mice displayed a trend of

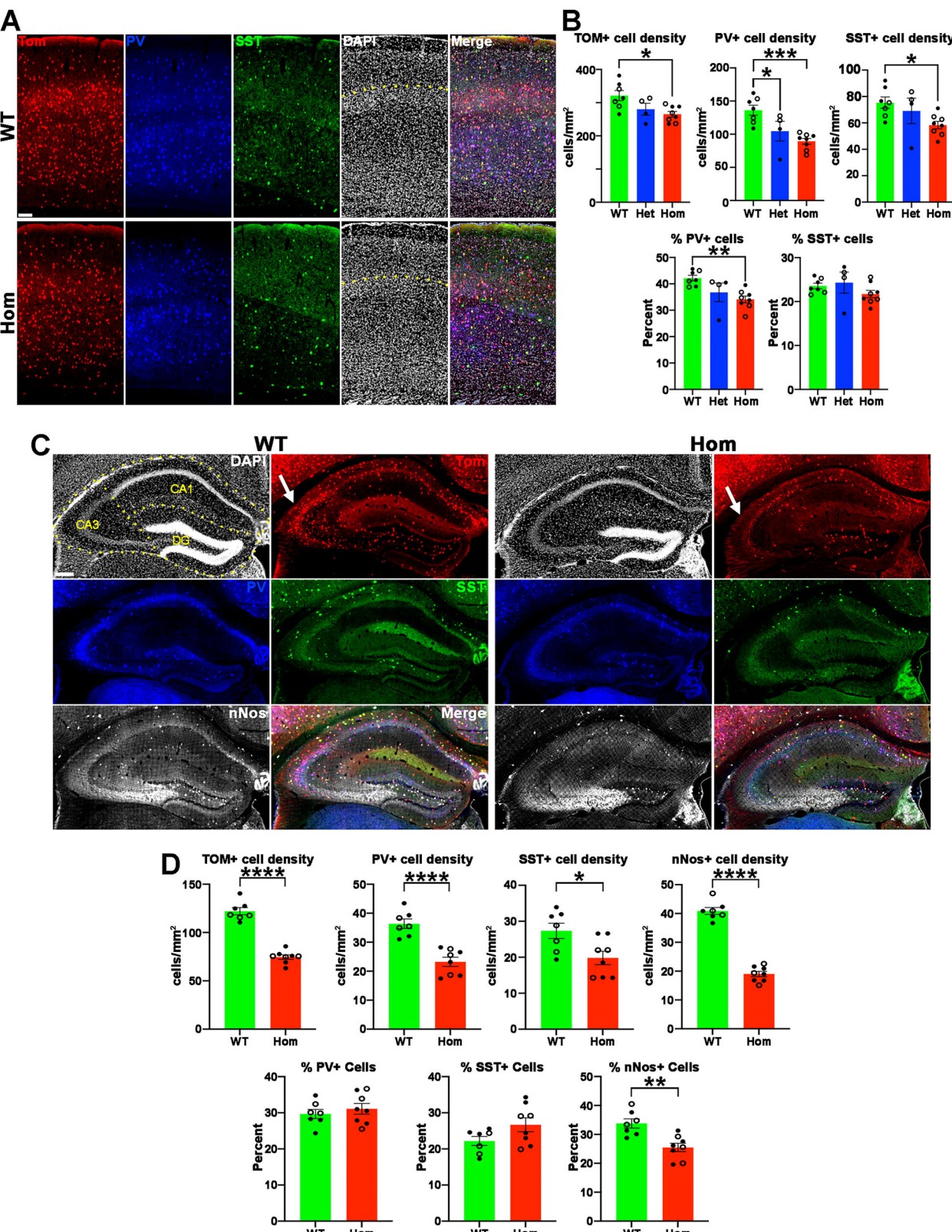

**Fig. 3 | Decreased MGE-derived interneurons in H3.3K4M Hom mice. A** P21 somatosensory cortex from H3.3K4M WT and Hom mice stained for tdTomato (red), PV (blue), SST (green) and DAPI (white). PV+ and SST+ cells that were Tom-negative were excluded from the counts. The yellow dotted line demarcates the boundary between superficial and deep cortical layers. Scale bar = 50 μm. **B** Quantification of cell density (top) and percent of PV+ and SST+ cells (bottom) in the somatosensory cortex. Open circles = females, filled circles = males. *n* = 3 WT female, 4 WT male, 2 Het female, 2 Het male, 4 Hom female, 4 Hom male. **C** P21 hippocampus from H3.3K4M WT and Hom mice stained for PV (blue), SST (green) and nNos (white). PV+ and SST+ cells that were Tom-negative were excluded from the counts. Yellow dotted lines demarcate CA1, CA3 and dentate gyrus (DG) boundaries. Scale bar = 100 μm. **D** Quantification of cell density (top) and percent of PV+ , SST+ and nNos+ cells (bottom) in the hippocampus. Open circles = females, filled circles = males. *n* = 3 WT female, 4 WT male, 3 Hom female, 5 Hom male. Data are presented as mean values +/− SEM. All stats are one-way ANOVA followed by Tukey's multiple comparison tests (B), or Standard Unpaired 2-tailed *t* test when only WT and Hom mice tested (D): * = $p \leq .05$, ** = $p \leq .005$, *** = $p \leq .0005$, **** = $p \leq .0001$. Source data are provided as a Source Data file.

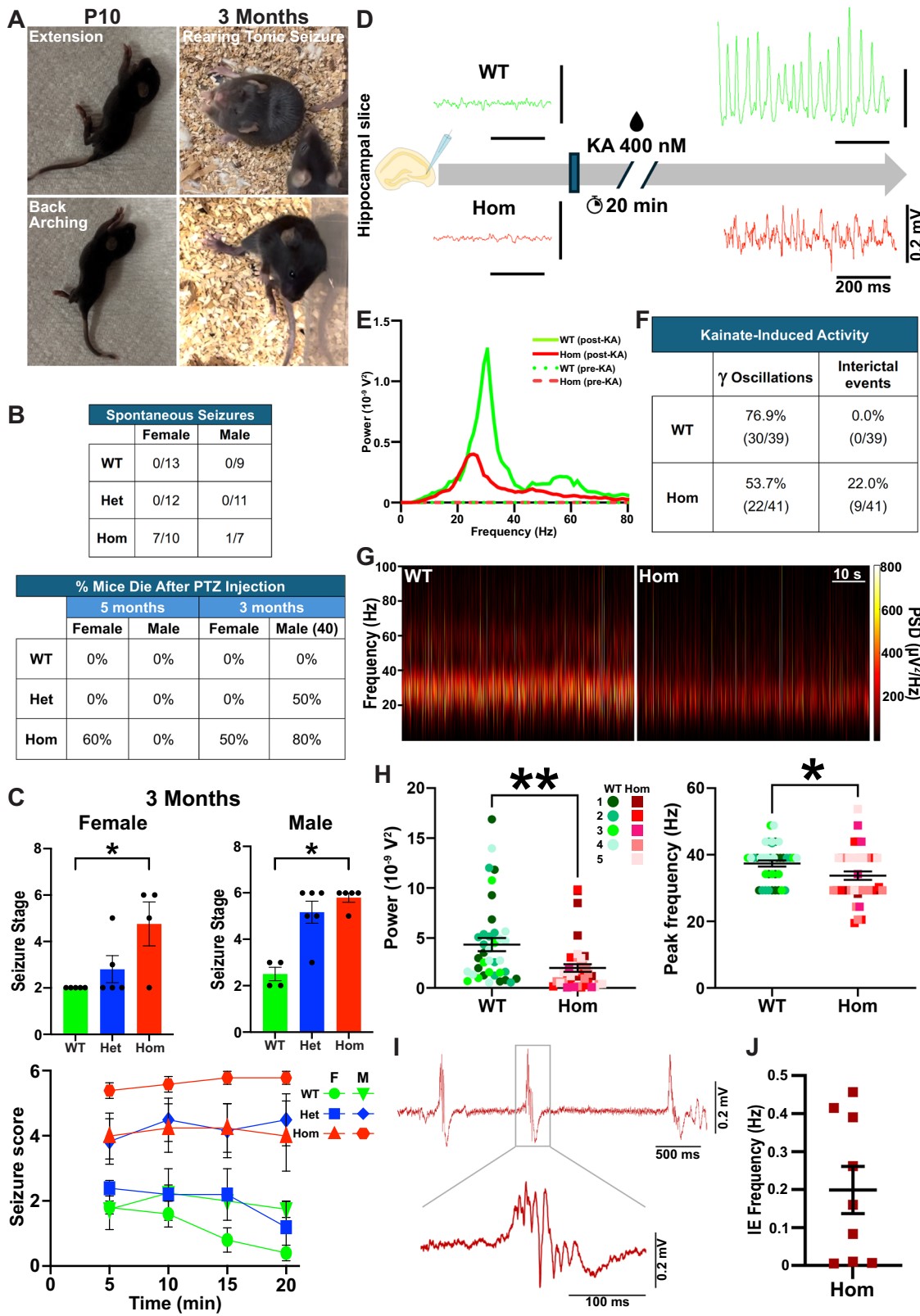

increased average speed in the open field test, which was not observed in male mice (Fig. 6B). In the light-dark box test, H3.3K4M Hom male and female male mice spent more time in the dark zone compared to WT mice (Fig. 6C). These two tests confirmed increased anxiety-like behavior with reduced H3K4 methylation.

Since H3.3K4M Hom mice sometimes jumped out of the cage during normal handling, we assessed these mice for increased impulsivity, a primary clinical symptom of ADHD that is observed in other NDDs[64,65]. We used the cliff avoidance reaction (CAR) test to assess impulsivity in these mice. Female H3.3K4M Hom mice had significantly more jump off events from the platform compared with WT (Fig. 6D and Supplementary Fig. 6B), suggesting increased impulsive-like behaviors. There was no difference in male mice.

**Fig. 4 | Spontaneous seizures and increased seizure sensitivity in H3.3K4M Hom mice.** Data is combined male & female samples unless stated otherwise. **A** H3.3K4M Hom mice undergoing infantile spasms at P10 (left) and spontaneous seizures at 3 months (right). **B** Quantification of spontaneous seizures observed in home cages from 3-5 months (top) and mortality rate after PTZ injection at 5 and 3 months (bottom). 3-month-old males received a 40 mg/kg PTZ dose (40), whereas all other conditions received a 20 mg/kg dose. **C** Maximum scores (seizure stage, top) for males (40 mg/kg) and females (20 mg/kg) following PTZ injection at 3 months (top) and during the 20 min observation period (bottom). n = 5 WT females, 4 WT males, 5 Het females, 6 Het males, 4 Hom females, 5 Hom males. **D, E** Local field potential traces recorded in ex vivo hippocampal slices from WT (green) and H3.3K4M Hom (red) brains before (left) and after (right) KA exposure (**D**), with respective power spectra (**E**). **F** Percent (and total number) of hippocampal slices displaying KA-induced gamma oscillations (left) and interictal events (right). **G** Spectrograms of gamma oscillations from WT (left) and H3.34 M Hom (right) brains. PSD: Power Spectra Density. **H** Significant decrease in power (left) and peak frequency (right) in H3.3K4M Hom mice. **I, J** Trace showing interictal events (IE) (**I**), and IE frequency from 9 slices that displayed IE (**J**) from H3.3K4M Hom mice. Data are presented as mean values +/− SEM. Kruskal-Wallis followed by Dunn's multiple comparisons (**C**: top) or two-way ANOVA followed by Tukey's multiple comparison tests (**C**: bottom) were performed based on the Normal distribution. The shapiro-wilk test was used to assess normality for electrophysiology data and Unpaired 2-tailed t-test was performed (**H**): * = $p \leq .05$, ** = $p \leq .005$. Source data are provided as a Source Data file.

H3.3K4M Hom mice also displayed unsteady gait and altered locomotion. To assess general locomotor activity, we continuously recorded singly housed mice for 4 days using the Photobeam Activity System. Both H3.3K4M Hom female and male mice displayed decreased ambulatory movements compared to WT mice, with an increase in fine movements in the center zone (Fig. 6E and Supplementary Fig. 6C). No difference in rearing activity was observed (Supplementary Fig. 6C). We used the Free-Walk system to quantify gait parameters and motor coordination in 3-month-old mice (Fig. 6F). Female H3.3K4M Hom mice displayed significantly reduced size, pressure, stance of hind-right paw, decreased BaseFront, and an increased stride length compared to WT mice (Fig. 6G and Supplementary Fig. 6D). Gait properties of male H3.3K4M Hom mice were normal except for a slight increase in BaseFront. Thus, H3.3K4M Hom mice display increased anxiety, decreased general locomotion and altered gait activity, with females generally displaying more severe symptoms compared to males.

## H3.3K4M Hom mice display impaired sensorimotor gating and memory

We then assessed higher-order cognition and behavior in H3.3K4M mice. Proper sensorimotor gating is a critical aspect of normal brain function and animal behavior, and deficits in prepulse inhibition (PPI) are observed in NDDs such as schizophrenia and OCD[66]. In the acoustic startle reflex test, female H3.3K4M Hom mice showed an increase in startle amplitude and a decrease percent of PPI compared to WT mice (Fig. 7A). No significant differences were observed in male H3.3K4M Hom mice.

We used the Barnes maze and 2-object novel recognition paradigms to explore memory deficits in H3.3K4M mice. In the Barnes maze, female H3.3K4M Hom mice exhibited a significantly shorter time around the target hole and trended to spend less time in the target quadrant compared to WT mice (Fig. 7B), indicating decreased memory performance in this task. Notably, H3.3K4M Hom mice traveled a greater total distance compared to WT mice, indicative of hyper-activity and increased anxiety (Supplementary Fig. 6E). In the 2-object novel recognition assay, WT female mice exhibited preference for novel versus familiar objects as expected, while female H3.3K4M Hom mice showed no preference for the novel object (Fig. 7C). No significant differences were observed in the male H3.3K4M Hom mice.

To evaluate sociability deficits, we performed the three-chamber test. We found no significant differences in sociability between H3.3K4M Hom and WT mice (Fig. 7D and Supplementary Fig. 6F). Curiously, H3.3K4M Hom female mice spent more time directly interacting with (sniffing) both the empty object and mice compared WT. We also observed nest-building defects in most singly housed H3.3K4M female and male Hom mice (Fig. 7E), which is observed in numerous mouse models of NDDs[67,68].

Altogether, our behavioral data indicate that reduced H3K4 methylation in the MGE and/or hypothalamus produces a broad range of behavioral phenotypes, many of which are commonly found in NDDs and other models of altered H3K4 methylation[8,9]. H3.3K4M Hom mice display increased anxiety and impulsivity, hypoactivity and decreased ambulatory locomotion in the home cage, abnormal gait, and deficits in sensorimotor gating and memory. Notably, female Hom mice displayed more severe phenotypes compared to males in many assays, indicating a clear sex bias in these mutant mice.

## Alterations in transcriptome associated with interneurons fate and seizure

We performed single-nucleus Multiome analysis (snRNA-seq + snATAC-seq) on E13.5 MGE, E13.5 hypothalamus, P60 cortical MGE-derived interneurons and P60 hypothalamus cells from male and female WT and H3.3K4M Hom mice. We generated *Nkx2.1-Cre;H3.3K4M;Sun1-sfGFP* mice to harvest *Nkx2.1*-lineage GFP+ nuclei via flow cytometry (Supplementary Fig. 7). We obtained 50,684 nuclei from embryonic mice and 22,116 nuclei from P60 mouse brains (Supplementary Data 1). There was clear segregation of both ages and brain regions when visualizing RNA-only, ATAC-only or integrated RNA and ATAC data using the weighted nearest neighbor (WNN) analysis (Fig. 8A and Supplementary Fig. 8A–D). Cells from the same age and brain region largely overlapped regardless of genotype or sex (Fig. 8B and Supplementary Fig. 8E, F).

We observed a smaller percentage of GFP+ nuclei in the MGE from H3.3K4M Hom mice (75.3% in WT vs. 62.1% in Hom) (Supplementary Fig. 7), which is consistent with fewer interneurons in the cortex and hippocampus in mutant mice. We identified 20 putative cell clusters that can be classified as apical progenitors (*Nestin*+ ), basal progenitors (*Ccnd2*+ and *Nestin*-) and postmitotic neurons (*Dcx*+ and *Ccnd2*-) following the expected developmental trajectory (Fig. 8C, D and Supplementary Fig. 9A, B). These cells can be further classified into 6 progenitor and 7 postmitotic cell types based on their molecular signatures (Fig. 8E and Supplementary Fig. 9C). The proportion of PV-fated (*Maf*+ ) and Sst-fated (*Sst*+ ) cells was decreased in H3.3K4M Hom MGE (Fig. 8E), matching reductions of SST+ and PV+ interneurons in postnatal mice.

We identified many differentially expressed genes (DEGs) in both H3.3K4M Hom female and male MGE, with the overwhelming majority of DEGs being downregulated in Hom mice (Fig. 8F and Supplementary Fig. 9D). Nearly all differentially accessible peaks were also less accessible in the MGE of H3.3K4M Hom mice (Supplementary Fig. 8G). Many genes critical for general interneuron development (*Dlx1, Dlx6, Gad1*) and specific for MGE-derived interneurons (*Nkx2.1, Lhx6, Lhx8*) (Fig. 8F and Supplementary Fig. 9D) were downregulated in Hom female and/or male mice. Of note, 2 genes predictive of PV+ inter-neurons, *Maf* and *Mef2c*[69,70], are respectively reduced in H3.3K4M Hom female MGE and male MGE, which is consistent with the specific reduction of PV+ cells in the postnatal cortex (Fig. 3). Gene ontology (GO) analysis revealed that DEGs were enriched in categories relating to forebrain development and neuron migration (Fig. 8G and Supplementary Fig. 9E).

P60 MGE-derived interneurons were cleanly divided into SST+ and PV+ interneurons, with a small population of oligodendrocytes (Fig. 8H and Supplementary Fig. 10A). Based on previous PV+

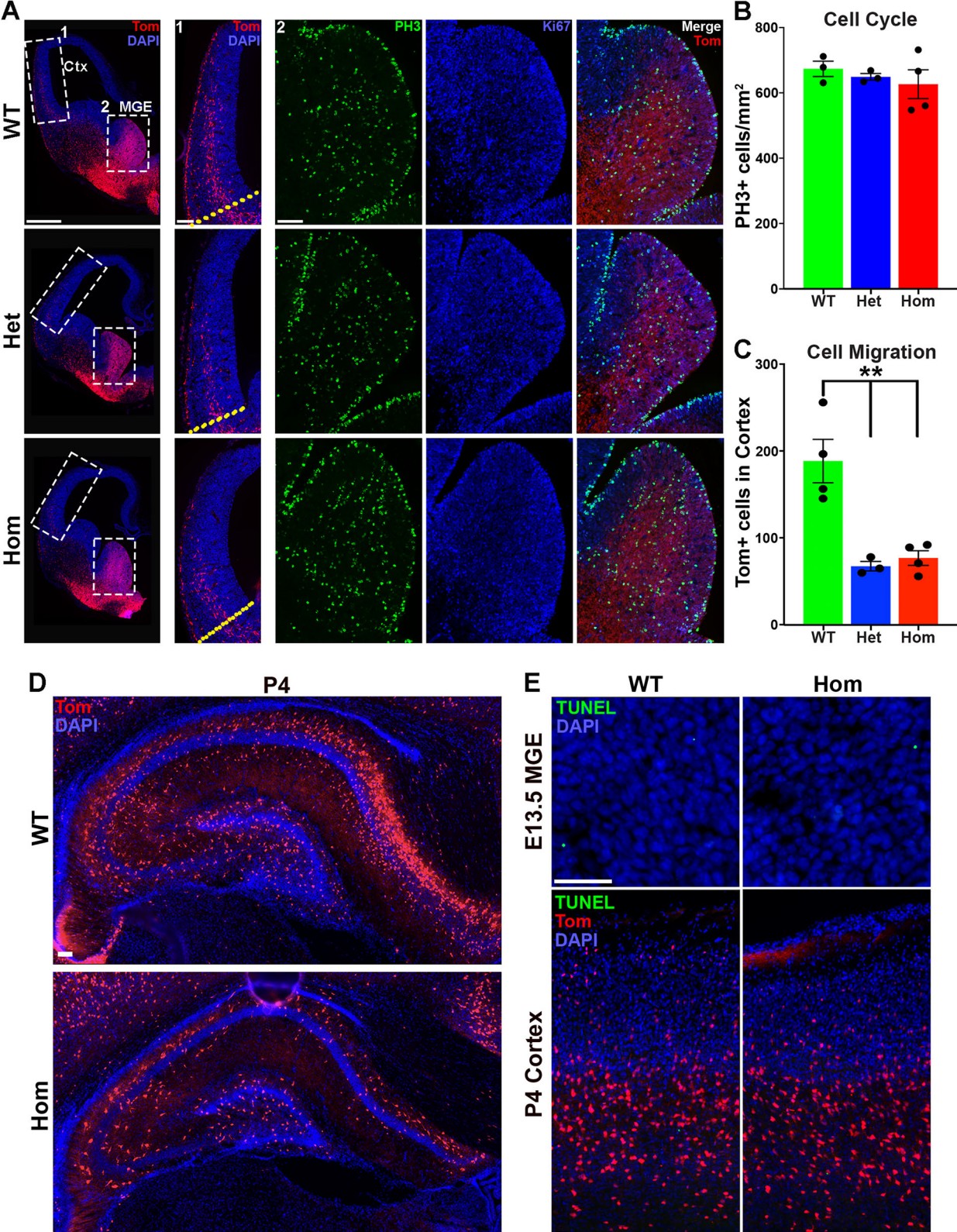

**Fig. 5 | Reduced interneuron migration in H3.3K4M Hom mice.** Data is combined male & female samples. **A** Hemisections through E13.5 MGE of H3.3K4M WT, Het and Hom mice (left), with higher magnification images of migrating Tom+ cells in the cortex (1, middle), and PH3 and Ki67 immunostaining within the MGE (2, right). White dotted lines indicate regions of higher magnification images. The yellow dotted line demarcates the pallium-subpallium boundary. Scale bar = 500 μm (left) & 100 μm (middle, right). **B** Density of PH3+ cells in the MGE of WT (*n* = 3), Het

(*n* = 3) and Hom (*n* = 4) embryos. **C** Reduced Tom+ neurons in the E13.5 cortex of Het (*n* = 3) & Hom (*n* = 4) mice compared with WT (*n* = 4). **D** Reduction of Tom+ cells in P4 H3.3K4M Hom hippocampus. Scale bar = 100 μm. **E** TUNEL signal (green) in E13.5 MGE (top) and P4 cortex (bottom) of H3.3K4M WT and Hom mice. Scale bar = 100 μm. Data are presented as mean values +/− SEM. All stats are one-way ANOVA followed by Tukey's multiple comparison tests (**B**, **C**): ** = $p \leq .005$. Source data are provided as a Source Data file.

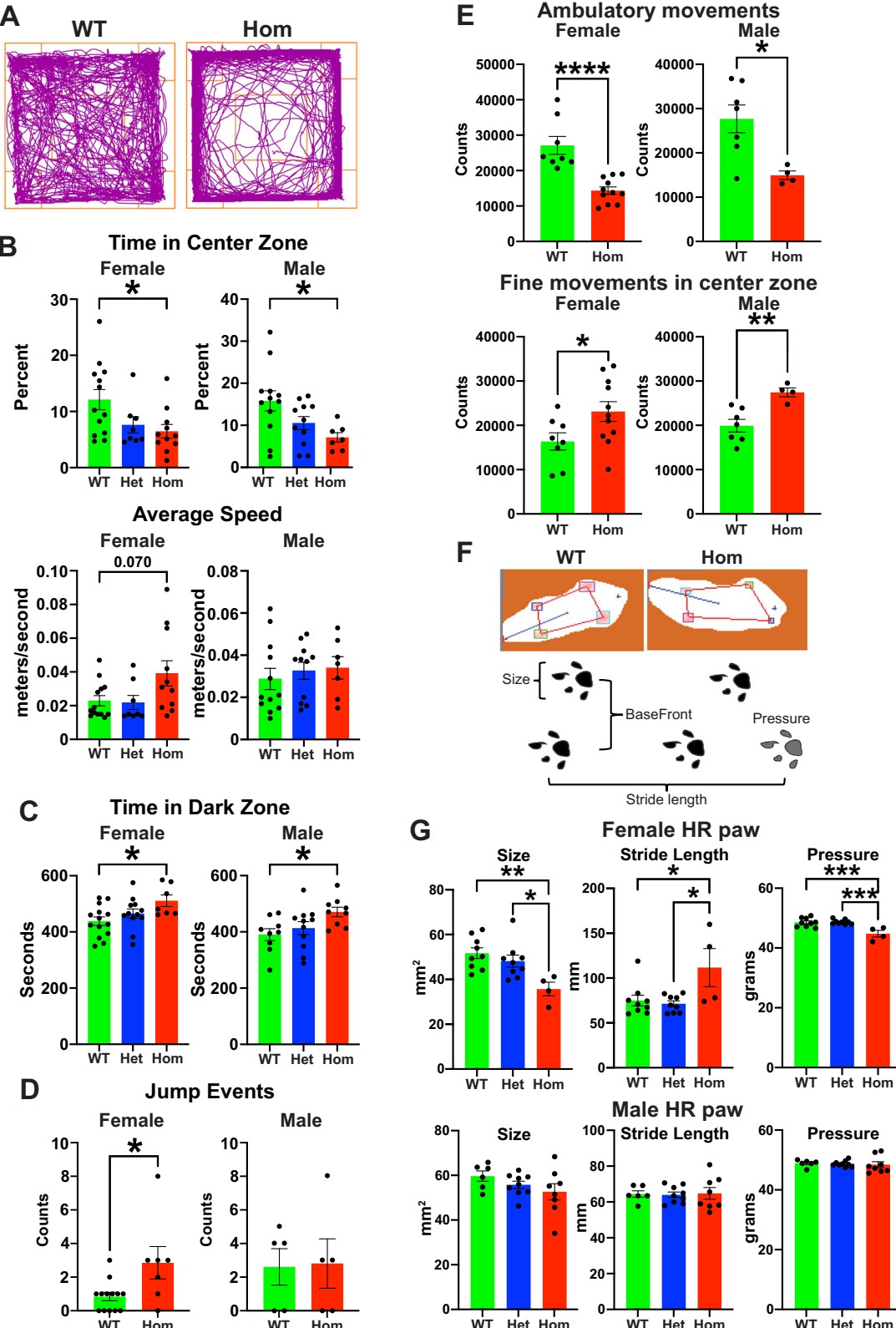

**Fig. 6 | H3.3K4M Hom mice display increased anxiety and impaired locomotion. A** Trajectory plot of H3.3K4M WT and Hom mice in the open field test. **B** Time in the center zone (top) and average speed (bottom) in the open field test. *n* = 13 females, 12 WT males, 8 Het females, 11 Het males, 11 Hom females, 7 Hom males. **C** Time spent in the dark zone of the light-dark box. *n* = 14 WT females, 9 WT males, 12 Het females, 11 Het males, 7 Hom females, 9 Hom males. **D** Jump events during the Cliff Avoidance Reaction (CAR) test. *n* = 13 WT females, 5 WT males, 7 Hom females, 5 Hom males. **E** Beam break counts of ambulatory movements in the whole zone (top) and fine movements in the center zone (bottom) in the home cage over

4 days. *n* = 8 WT females, 7 WT males, 11 Hom females, 4 Hom males. **F** Images from free walk test (top) with schematic depicting gait measurements (bottom). **G** Size, stride length and pressure of hind right (HR) paw in the free walk test. *n* = 9 WT females, 6 WT males, 9 Het females, 9 Het males, 4 Hom females, 8 Hom. Data are presented as mean values +/− SEM. All stats are one-way ANOVA followed by Tukey's multiple comparison tests when WT, Het and Hom mice (**B**, **C**, **G**). 2-tailed Mann-Whitney test (**D**) and Standard Unpaired 2-tailed t-test when only WT and Hom mice tested (**E**): * = $p \leq .05$, ** = $p \leq .005$, *** = $p \leq .0005$, **** = $p \leq .0001$. Source data are provided as a Source Data file.

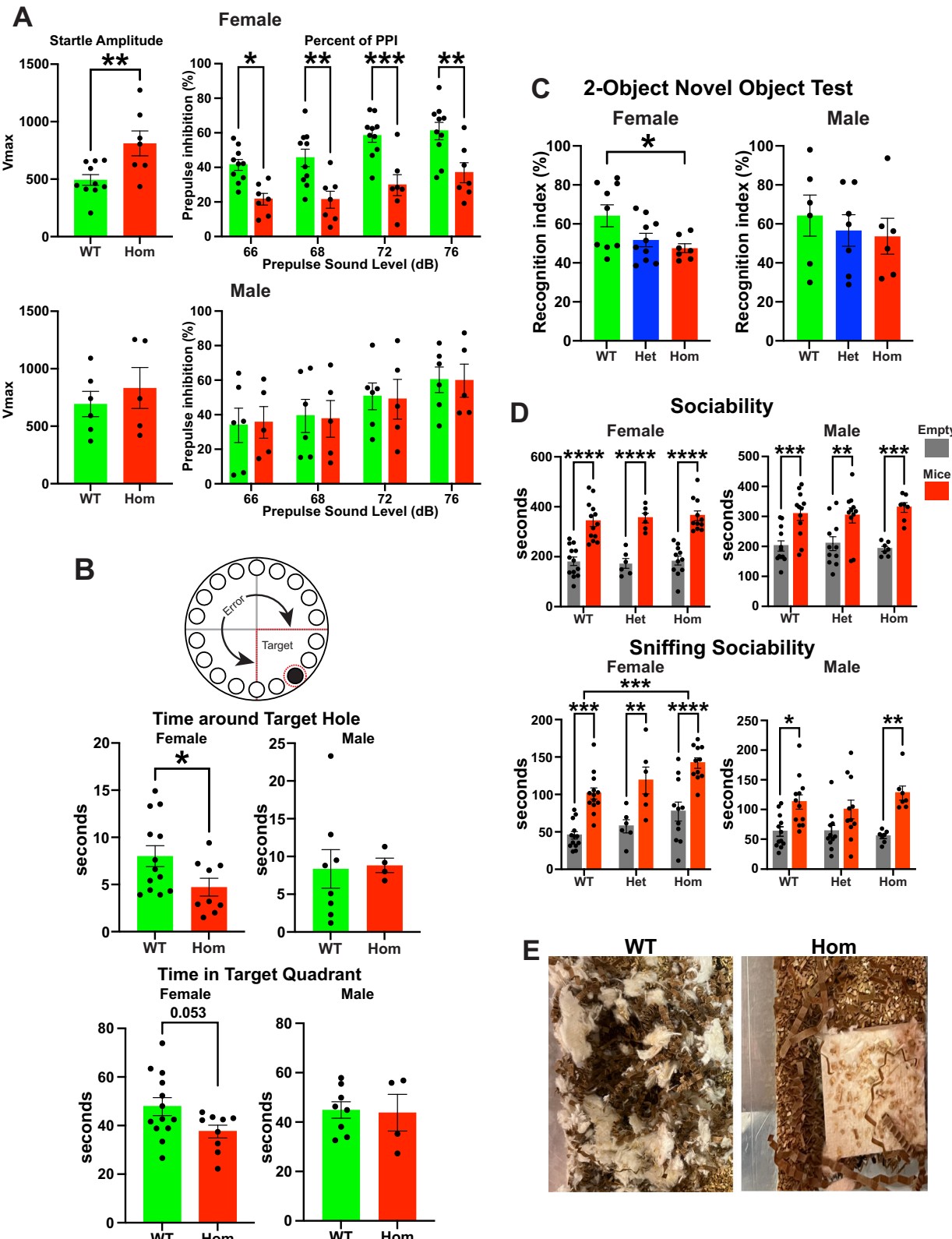

**Fig. 7 | H3.3K4M Hom mice display impaired sensorimotor gating and memory.**
**A** Startle amplitude (left) and percent prepulse inhibition (PPI) (right) in H3.3K4M WT and Hom mice. n = 10 WT females, 6 WT males, 7 Hom females, 5 Hom males. **B** Barnes maze schematic with red dotted line indicates target quadrant and region around target hole used for quantification. Time spent around the target hole (middle) and in the target quadrant (bottom) during the probe test. n = 13 WT females, 8 WT males, 9 Hom females, 4 Hom males. **C** Recognition index in the 2-object novel object recognition test. n = 9 WT females, 6 WT males, 10 Het females, 7 Het males, 7 Hom females, 6 Hom males. **D** Time spent in the chambers with (red) or without (black) stimulus

mouse in the three-chamber test (top). Time spent sniffing at the wire cage with or without a stimulus mouse (bottom). n = 13 WT females, 12 WT males, 6 Het females, 11 Het males, 11 Hom females, 7 Hom males. **E** Images of nest building for singly housed male H3.3K4M WT and Hom mice. Data are presented as mean values +/− SEM. All stats are one-way ANOVA followed by Tukey's multiple comparison tests (**C**) or two-way ANOVA followed Šídák's multiple comparisons test (**A**: percent of PPI, **D**) when WT, Het and Hom mice. Standard Unpaired 2-tailed t test when only WT and Hom mice tested (**A**: startle amplitude, **B**): * = $p \le .05$, ** = $p \le .005$, *** = $p \le .0005$, **** = $p \le .0001$. Source data are provided as a Source Data file.

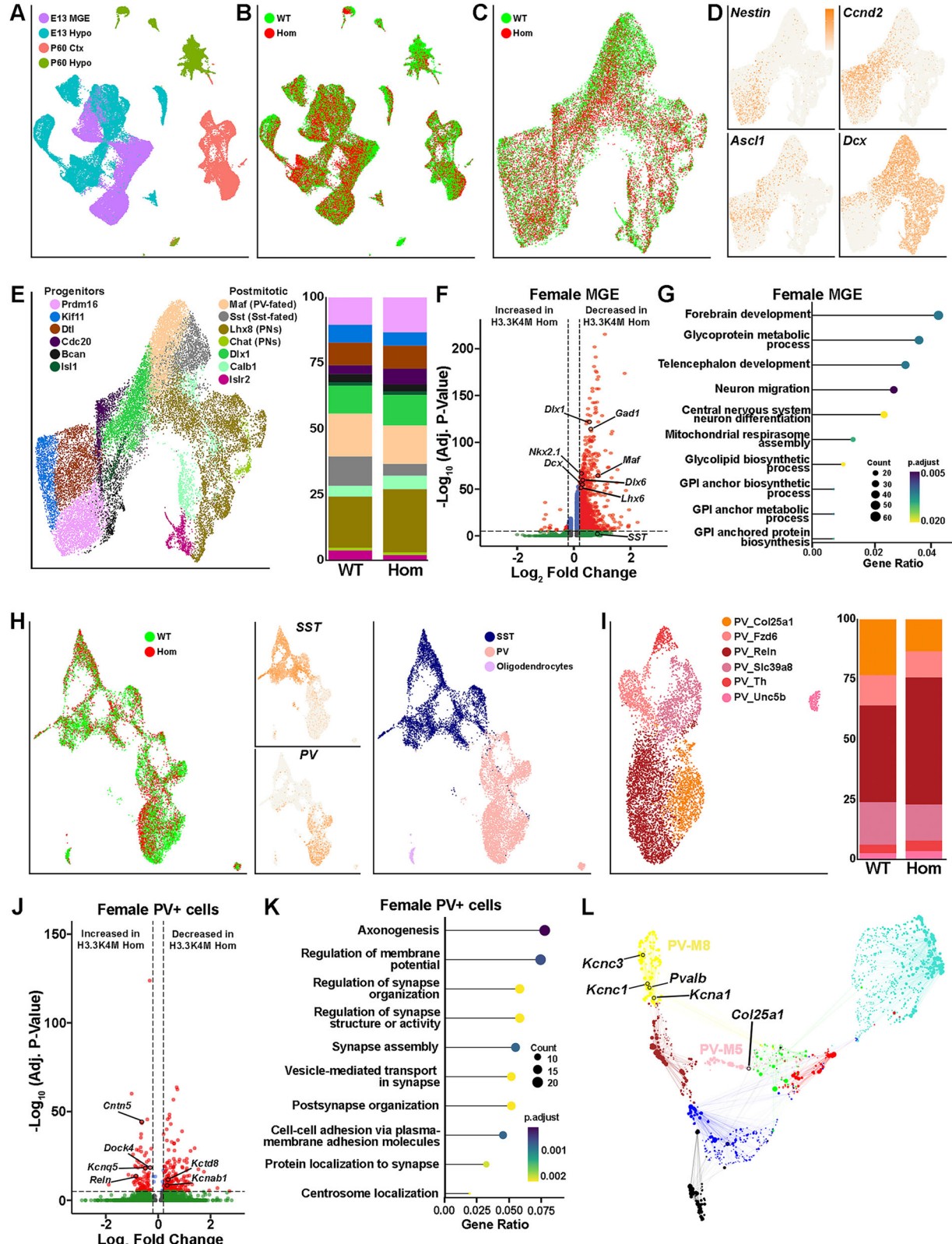

classifications[71], we identified six putative PV+ interneuron subtypes and noted a shift in the relative proportion of specific subtypes in H3.3K4M Hom mice: an increase of the PV-Reln population (from 40.3% to 53.0%) and a concomitant reduction in PV-Col25a1 cells (from 23.3% to 13.4%) (Fig. 8I and Supplementary Fig. 10B, C). We confirmed this predicted increase of PV/Reln+ cells in the cortex of H3.3K4M Hom mice (Supplementary Fig. 10D). Genes related to cell-cell adhesion

(*Dock4, Cntnap2, Cntn5*) were strongly upregulated in PV+ cells in H3.3K4M Hom mice, and the top gene families were related to axonogenesis and cell adhesion in both female and male mice (Fig. 8J, K and Supplementary Fig. 10E, F). Several genes involved in potassium ion transport (*Kcnq5, Kcnab1, Kcna1, Kcnc1, Kcnc3*, etc.) were dysregulated in the either female or male mice (Fig. 8J and Supplementary Fig. 10E). Of note, *Reln* was upregulated in Hom mice, in agreement

**Fig. 8 | Altered transcriptomes of interneurons associated with neuronal maturation and epilepsy.** Data is combined male & female samples unless stated otherwise. **A, B** Weighted nearest neighbor (WNN) integrated single nuclei RNA and ATAC UMAP plots of Nkx2.1-lineage cells from E13.5 MGE, E13.5 hypothalamus, P60 cortex and P60 hypothalamus annotated by age and tissue (**A**), and genotypes (**B**). **C, D** Integrated UMAP plots of E13.5 MGE cells annotated by genotypes (**C**) and marker genes *Nestin* (radial glia), *Ccnd2* and *Ascl1* (basal progenitors) and *Dcx* (postmitotic neurons) (**D**). **E** Integrated UMAP plot annotated by genetically defined cell types (left), with bar graph showing relative proportion of cell types in E13.5 WT and Hom MGE (right). **F** Volcano plot depicting DEGs in E13.5 H3.3K4M Hom female MGE. **G** clusterProfiler GO enrichment top biological process for DEGs of E13.5 female MGE. **H** Integrated UMAP plots of MGE-derived cells in the P60 cortex cells annotated by genotypes (left), *SST* and *PV* (middle), and SST+ and PV+ clusters (right). **I** RNA-only UMAP of PV+ cluster replotted and annotated by 6 PV+ subtypes (left), with relative proportions of PV+ subtypes in H3.3K4M WT and Hom cortices (right). **J** Volcano plot depicting DEGs in female P60 PV+ interneurons in H3.3K4M Hom female mice. **K** clusterProfiler GO enrichment top biological process for DEGs of female P60 PV+ interneurons. **L** Network visualization of the 8 PV modules identified by WGCNA with several hub genes labeled. Two-sided Wilcoxon rank sum test with Bonferroni correction (**F, J**) and one-sided Fisher's exact with BH correction (**G, K**) were used. Source data are provided as a Source Data file.

with the proportional increase of this PV+ subtype (Fig. 8I). Comparing DEGs in PV+ cells from male and female H3.3K4M Hom mice reveals both unique and shared features (Supplementary Fig. 10G).

We performed weighted gene co-expression network analysis (WGCNA)[72,73] to explore changes in gene regulatory networks in PV+ cells upon reduction of H3K4 methylation. Eight co-expression modules were found, with 5 of these modules being differentially regulated between WT and H3.3K4M Hom mice (Supplementary Fig. 10H). Notably, the downregulated potassium transporter genes *Kcna1, Kcnc1* and *Kcnc3* were found in module PV-M8, which has the largest change between WT and H3.3K4M Hom mice (Fig. 8L). *Col25a1* was found in another module downregulated in Hom mice (PV-M5), which matches the reduction of PV-Col25a1 subtype. Potassium ion channels, particularly Kv3 channels, are highly expressed in PV+ interneurons, and dysregulation of Kv3 channels alters their fast-spiking properties and increases seizure susceptibility[74,75], which matches the behavioral and electrophysiological changes in H3.3K4M Hom mice.

We identified 11 putative SST+ interneuron subtypes based on previous SST+ classifications[71,76] (Supplementary Fig. 11A, B). The most notable change was the proportional increase of the SST_Plpp4 subtype in H3.3K4M Hom mice, which we confirmed via FISH (Supplementary Fig. 11A, C). Genes related to neuron cell adhesion and regulation of synapse organization (*Lama4, Spock1, Stxbp6*) were downregulated in H3.3K4M Hom female and male SST+ cells (Supplementary Fig. 11D–F), which could indicate altered synaptic connectivity with pyramidal neurons. Several genes were increased in female Hom mice but decreased in males (*Ubb, Atp6v0c, Stmn3*), which could be related with biased sex-based behaviors.

In sum, perturbation of H3K4 methylation alters the transcriptome of embryonic MGE and adult MGE-derived interneurons that are highly correlative, and likely causative, to interneuron-related phenotypes in the mutant mice. This includes a decrease of subtype predictive genes in the MGE, changes in the proportion of PV+ and SST+ subtypes in the adult cortex, and dysregulation of gene families involved in potassium ion transport and cell adhesion that are associated with synaptic physiology and seizure susceptibility.

## Dysregulation of hypothalamic gene expression, cellular composition and structure in H3.3K4M mutant mice

We subsetted the E13.5 hypothalamus datasets and used gene expression studies[77–80] to identify *Nes*+ neural progenitor cells (NPCs) and 11 putative hypothalamic nuclei consisting of both glutamatergic (*Slc17a6*+) and GABAergic (*Slc32a1*+) cells (Fig. 9A, B and Supplementary Fig. 12A–C). While interneurons undergo extensive postnatal diversification and transcriptional changes, hypothalamic subtypes emerge early during embryogenesis with minimal postnatal diversification[45,81], which allows us to identify putative hypothalamic cell types with high confidence.

Like the MGE, we observed a lower percentage of GFP+ nuclei in the H3.3K4M Hom hypothalamus compared to WT during sorting (41.5% GFP+ nuclei in WT vs 25.5% in Hom) (Supplementary Fig. 7). There was a more even distribution of upregulated and downregulated DEGs compared to MGE, with a notable increase in differentially

accessible peaks in the H3.3K4M Hom mouse (Fig. 9D and Supplementary Fig. 8G and 12E). We observed an increased proportion of NPCs in H3.3K4M Hom mice (27.2% vs. 16.4%) (Fig. 9C). Numerous genes associated with mitotic cell cycle phase transition (*Ccnd2, Mki67, Cdk6*) were upregulated in both H3.3K4M Hom female and male hypothalamus (Fig. 9D, E and Supplementary Fig. 12E-F). We confirmed increased Ki-67+ cells in the E13.5 hypothalamus of H3.3K4M Hom mice (Supplementary Fig. 12G). In addition, there is increased accessibility at the *Nes* promoter and intron enhancer in H3.3K4M Hom female and male mice compared to WT (Fig. 9F and Supplementary Fig. 12H).

The greatest proportional reduction in H3.3K4M Hom mice were the arcuate nuclei (ARC), which is critical for regulating energy metabolism and food intake[82], and the ventromedial hypothalamus (VMH) nuclei that control neuroendocrine functions such as appetite and glucose regulation[83] (Fig. 9C and Supplementary Fig. 12D). The ARC marker gene *Pomc* was decreased in the Hom hypothalamus (Fig. 9D), consistent with a decrease proportion of ARC-fated nuclei. We also identified both sex-specific and shared DEGs in E13.5 male and female hypothalamus (Supplementary Fig. 12I). Similar to the MGE, we did not identify any differences in apoptosis in the hypothalamus between WT and Hom mice (Supplementary Fig. 12J).

In the P60 hypothalamus, we identified 40 putative cell clusters with clear separation of neurons (*Syt1*), astrocytes (*Agt*), oligodendrocytes (*Mog*) and tanycytes (*Rax*) (Fig. 9G–I and Supplementary Fig. 12A, B). Specific hypothalamic nuclei could not be cleanly resolved in our dataset because we obtained too few cells relative to the heterogeneity of cell types and nuclei in the adult hypothalamus (Supplementary Data 1); a recent study compiled ~400,000 mouse hypothalamic cells and identified 185 distinct cell clusters[84]. DEGs in adult hypothalamic neurons were associated with axonogenesis, cell adhesion and NADH metabolic process (Supplementary Fig. 12C–E).

We found a striking increase in the proportion of astrocytes (8.2% in WT vs. 27.3% in Hom) and decrease in oligodendrocytes (12.3% in WT vs. 2.2% in Hom) (Fig. 9I) in H3.3K4M Hom mice. We confirmed an increase in hypothalamic GFAP+/Tom+ Nkx2.1-lineage astrocytes, and overall astrocyte numbers, in both male and female H3.3K4M Hom mice (Supplementary Fig. 14A–C). This increase in astrocytes is also found in diet-induced obese mice[85]. Astrocytes play critical roles in controlling glucose metabolism and energy balance[44,86,87], inflammatory processes[83], maintaining homeostasis of neurotransmitter and ions[88], regulating synapse plasticity linked to behaviors[89]. Unlike other cell types, hypothalamic astrocytes from WT and Hom mice form distinct clusters in the UMAP plots (Fig. 9G). Re-clustering the astrocytes revealed sex and genotype-specific clusters; in particular, we observed cluster 5 was enriched in innate immune reactivity-related genes (*Usp18, Oasl2, Ifit1*) and was only present in H3.3K4M Hom female and male astrocytes (Fig. 9J–L and Supplementary Fig. 13F). DEGs in astrocytes include those involved in regulation of monoatomic ion transport (*Asic2, Grm5, Gabrb1*), synaptic transmission (*Nrxn3, Nrxn1*), glial cell differentiation and maturation (*Npas3, Gpc5*) and regulation of membrane potential (*Kcnq3*), which may indicate

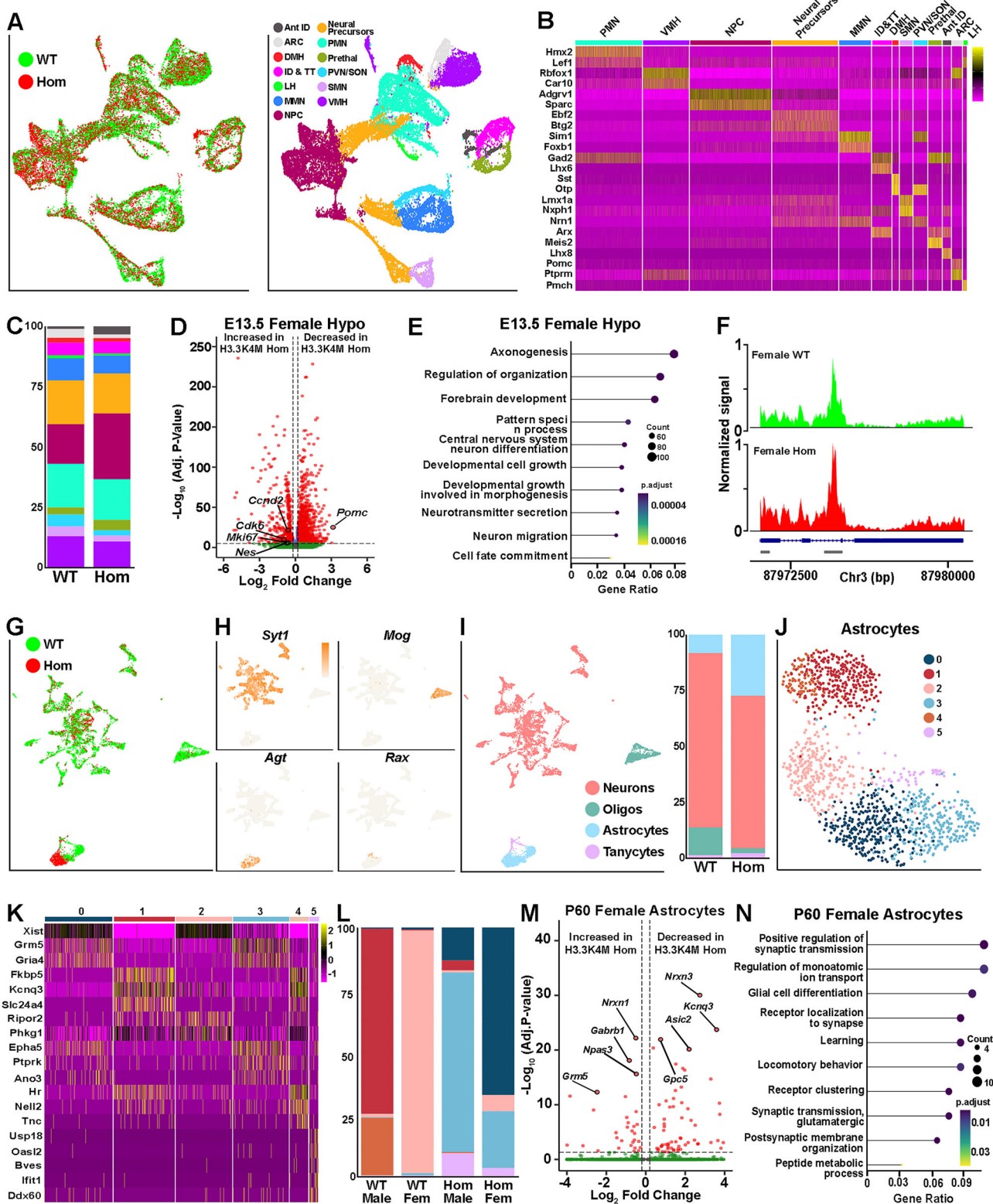

disruption of normal hypothalamic homeostasis in H3.3K4M Hom female and male mice (Fig. 9M, N and Supplementary Fig. 13G–I).

In sum, perturbation of H3K4 methylation alters the transcriptome of embryonic and adult hypothalamic cells, with a notable increase in the proportion of NPCs and reduction in the proportion of ARC and VMH nuclei, which both play critical roles in feeding behavior and growth hormone regulation. We also observed a large increase in hypothalamic astrocytes in H3.3K4M mice and distinct transcriptional astrocyte sub-types that likely alter hypothalamic function.

## Abnormal cellular organization in the hypothalamus of H3.3K4M Hom mice

We explored the hypothalamic cellular organization in more detail. Tanycytes are specialized glial cells within the blood-brain barrier of the hypothalamus that can detect nutrients and metabolites in the blood and cerebrospinal fluid, sending this information to the ARC and VMH to regulate body weight and energy metabolism[90,91]. They also act as neural progenitor cells, giving rise to new neurons and glia cells in the adult hypothalamus[90]. In adult WT mice, Nkx2.1-lineage Tom+/

**Fig. 9 | Significant changes in gene expression and cellular composition in the hypothalamus of H3.3K4M Hom mice.** Data is combined male & female samples unless stated otherwise. **A** WNN integrated single nuclei RNA and ATAC UMAP plots of Nkx2.1-lineage cells from E13.5 hypothalamus annotated by genotypes (left) and hypothalamus nuclei (right). **B** Heatmap depicting genes enriched in specific hypothalamic nuclei. **C** Relative proportions of hypothalamic nuclei in H3.3K4M WT and Hom, with color scheme matching panel (**A**). **D** Volcano plot depicting DEGs in E13.5 H3.3K4M Hom female hypothalamus dataset. **E** clusterProfiler GO enrichment top biological process for DEGs of E13.5 female hypothalamus. **F** Tracks showing increased genomic accessibility at *Nes* promoter and intronic enhancer (gray bars) in female E13.5 H3.3K4M Hom hypothalamus. **G, H** Integrated UMAP plots of P60 hypothalamus annotated by genotypes (**G**), and marker genes *Syt1* (neurons), *Mog* (oligodendrocytes), *Agt* (astrocytes) and *Rax* (tanycytes) (**H**). **I** Integrated UMAP plot annotated by cell type (left) and proportion of each cell type in WT and Hom mice (right). **J** RNA-only UMAP plot of astrocytes annotated by putative astrocyte subtypes. **K** Heatmap depicting genes enriched in astrocyte subtypes. **L** Proportion of astrocyte subtype in WT and Hom female & male samples. **M** Volcano plot depicting DEGs in female P60 Hom astrocytes. **N** clusterProfiler GO enrichment top biological process for DEGs of female P60 astrocytes. Ant ID, anterior intrahypothalamic diagonal; ARC, arcuate nucleus; DMH, dorsomedial nucleus; ID&TT, intrahypothalamic diagonal & tuberomammillary terminal; LH, lateral hypothalamus; MMN, mammillary nucleus; NPC, neural progenitor cells; PMN, premammillary nucleus; Prethal, Prethalamus; PVN/SON, paraventricular nucleus/supraoptic nucleus; SMN, supramammillary nucleus; VMH, ventromedial hypothalamus. Two-sided Wilcoxon rank sum test with Bonferroni correction (**D**, **M**) and one-sided Fisher's exact with BH correction (**E**, **N**) were used. Source data are provided as a Source Data file.

Vimentin+ (Vim+) tanycytes line the third ventricle (α-tan) and median eminence (ME, β-tan), with their processes extending out in an organized manner as previously described[92] (Fig. 10A). A layer of Tom-/GFAP+ astrocytes lie just below these tanycytes in the ME. In male and female H3.3K4M Hom mice, this organization is completely disrupted, with Vim+ tanycytes scattered throughout the ME and GFAP+ astrocytes clumped at the lateral edges of the ME (Fig. 10A). α-tan processes along the wall of the third ventricle are more disorganized and fasciculated compared to WT mice. In addition, oligodendrocytes line the dorsal portion of the ME, where they are involved in the regulation of energy availability and leptin sensitivity[93]. This myelin band was present in the ME of WT mice but was absent in Hom mice, with myelin instead restricted to the lateral edges of the ME, similar to astrocytes (Fig. 10B). Some hypothalamic neurons project to the ME and release the neuropeptide SST that is involved in growth hormone release and food intake[94]. We observed significantly less Sst in the ME in both Hom female and male mice (Fig. 10C).

We subsetted out the *Rax*+/*Col23a1*+ tanycytes in the hypothalamus and identified the 2 tanycyte subtypes as previously described[95]: *Vcan*+ α-tans and *Col25a1*+ β-tans (Fig. 10D). We identified very few DEGs in tanycytes, in large part due to low cell numbers. *Fgf14* was the most downregulated gene in H3.3K4M Hom tanycytes (Fig. 10D), and FGF signaling in tanycytes regulates their morphology and proliferation[92,96], lipid homeostasis[97] and numerous hypothalamic metabolic process[98,99]. While a reduction in *Fgf14* in tanycytes was difficult to confirm via FISH, we observed slightly decreased *Fgf14* levels in the ARC nucleus adjacent to tanycytes in H3.3K4M Hom mice (Fig. 10F) that may indicate altered FGF14 signaling, but this still needs further confirmation.

In sum, reduced H3K4 methylation alters astrocyte, oligodendrocyte and tanycyte organization in the hypothalamus, specifically in the ME. Disruption of normal cellular architecture at this critical blood-barrier junction will alter hypothalamic function and hormonal regulation that drives initial growth retardation, followed by obesity, increased leptin levels, and likely contributes to altered behavioral phenotypes in H3.3K4M Hom mice.

## Discussion

Dysregulation of H3K4 methylation is associated with ASD, schizophrenia and numerous other NDDs with ID[8]. While several studies have utilized lysine-to-methionine (K-to-M) mutations to study histone modifications[100], the Cre-dependent H3 variant H3.3K4M transgenic mouse permits precise spatial and temporal control to study H3K4 methylation during neurodevelopment. We expressed this mutant histone in the embryonic MGE and hypothalamus, two brain regions associated with H3K4 disease-related phenotypes such as ID, epilepsy and abnormal body growth[3,4]. Reduction of H3K4 methylation in the embryonic MGE and hypothalamus recapitulates these most consistent phenotypes observed in H3K4-associated diseases. H3.3K4M Hom mice had fewer interneurons, reduced network synchrony, and

dysregulation of many genes in PV+ interneurons associated with potassium ion transport, ultimately leading to increased seizure activity in these mice. In the hypothalamus, there was a significant change in the relative proportion of cells and striking cellular disorganization along the blood-brain barrier. These changes, combined with altered transcriptome profiles, likely disrupt normal hormonal regulation, energy metabolism and other critical hypothalamic functions, leading to altered body growth and behavioral phenotypes in these mice.

In *Nkx2.1-Cre;H3.3K4M* mice, the H3.3K4M allele is present in Nkx2.1+ MGE and hypothalamus progenitors and is expressed throughout the life of the cell. While we demonstrate genetic and cellular changes in the embryonic brain, we cannot definitively assign specific phenotypes to perturbation of H3K4 methylation during embryogenesis, cell maturation, or at adult stages. Cells develop along a continuum, with different characteristics maturing at distinct developmental timepoints, many of which are dependent on earlier processes. Future studies using other Cre-driver lines or viral injections to activate H3.3K4M at different developmental timepoints are required to determine the relationship between H3.3K4M onset and disease phenotype.

Both male and female H3.3K4M Hom mice had fewer MGE-derived interneurons in the cortex and hippocampus, particularly PV+ interneurons, which is predicted by the decrease of PV-fated genes *Maf* (in females) and *Mef2c* (in males) in the MGE. A likely cause for reduced forebrain interneurons was decreased cortical migration. There is evidence that disruption of interneuron fate and migration can lead to non-cell autonomous changes in excitatory cells, including altered neurogenesis and defects in cortical lamination and neuron activity[101,102]. Whether the reduced cortical thickness in H3.3K4M mutant mice is solely from loss of interneurons or also arises from altered excitatory neuron development is not clear.

While all H3.3K4M Hom mice displayed significantly increased seizure scores upon PTZ injections compared to WT mice, 5-month-old female H3.3K4M Hom mice had a much higher seizure score and higher mortality rate compared to 5-month-old Hom males. As the PTZ dose was the same at 5 months, this likely represents the true sex difference in seizure susceptibility of H3.3K4M Hom mice. In addition, many 3-month-old male H3.3K4M Hom mice died upon PTZ injection, while no 5-month-old males died. As the 3-month male mice received a higher dose of PTZ, it's unclear if these mortality differences are due to dosage effects or differential response of H3K4 methylation between these ages.

In addition to the changes in interneuron numbers, the increased variability in electrical properties is reminiscent of the developing hippocampus[103,104], suggesting interneuron maturation deficits in H3.3K4M Hom mice. As widespread perisomatic inhibition is critical for both circuit gamma entrainment and seizure control[105,106], it is likely that the deficits in hippocampal FS/PV+ cells in H3.3K4M Hom mice directly contribute to disrupted rhythmogenesis and epilepsy.

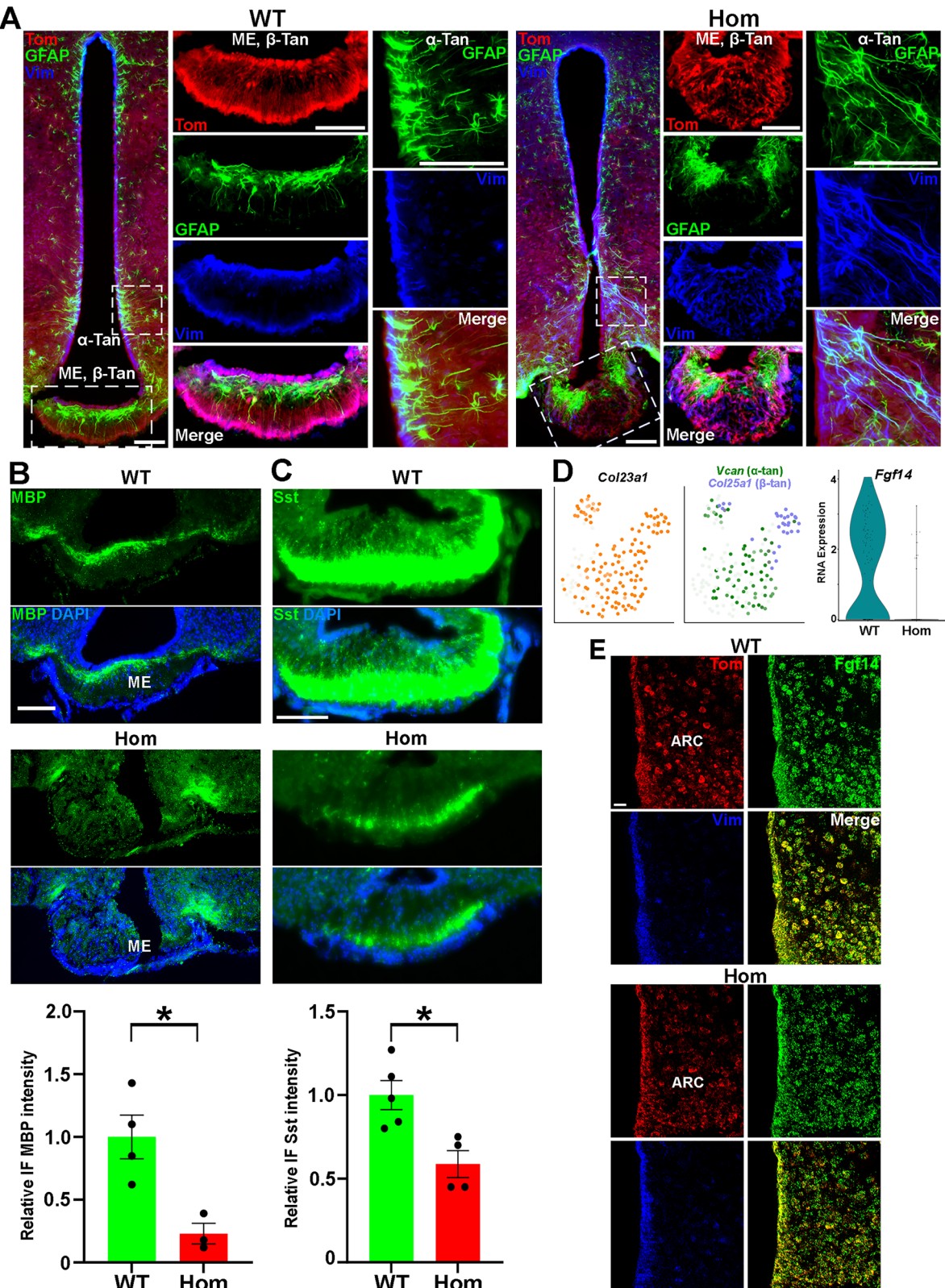

**Fig. 10 | Abnormal cellular organization in the hypothalamus of H3.3K4M Hom mice. A** Representative image of -P60 hypothalamus from H3.3K4M WT and Hom mice stained for GFAP (green) and Vimentin (Vim, blue). White boxes indicate higher magnification areas of median eminence (ME) and hypothalamus lining 3rd ventricle for each genotype. Scale bars = 100 μm. **B** P60 median eminence (ME) from H3.3K4M WT ($n = 4$) and Hom ($n = 3$) mice stained for MBP with quantification. Scale bar = 100 μm. **C** P21 ME from H3.3K4M WT ($n = 5$) and Hom ($n = 4$) mice stained for SST with quantification. Scale bar = 100 μm. **D** RNA-only UMAP plots of all *Col23a1*+ tanycytes (left), *Vcan*+ α-tanycytes and *Col25a1*+ β-tanycytes (middle), and a Violin plot showing significant downregulation of *Fgf14* in H3.3K4M Hom tanycytes (right). **E** FISH images of P60 hypothalamus from H3.3K4M WT and Hom mice stained for *tdTomato* (red), *Fgf14* (green) and *Vimentin* (blue). ARC = Arcuate nucleus. Scale bar = 100 μm. Data are presented as mean values +/− SEM. Standard 2-tailed *t* test was tested: * = $p ≤ .05$. Source data are provided as a Source Data file.

Notably, perturbation of potassium ion channels and transporters are well-established causes of epilepsy[107,108], and potassium channels are critical regulators of gamma oscillations[109,110]. Many of these genes were also enriched in the top-downregulation co-expression module of PV population in H3.3K4M Hom mice. This finding, in combination with proportional changes of specific PV+ subtypes, indicates that distinct cohorts of genes in certain PV+ interneuron subtypes are preferentially affected upon perturbation of H3K4 methylation.

Mutations that alter histone modifications can have differential effects based on cell types, development stage, and resistance or susceptibility at specific genomic loci[48,100,111]. Consistent with these observations, we found differential effects of perturbed H3K4 methylation on the MGE and hypothalamus. The overwhelming majority of DEGs were downregulated in the MGE of H3.3K4M Hom mice (as were differentially accessible peaks), and this trend was maintained in PV+ and SST+ cortical interneurons. DEGs in the embryonic hypothalamus were more evenly distributed between upregulated and downregulated in the H3.3K4M Hom mouse, while there was a strong bias for differentially accessible peaks being enriched in the mutant hypothalamus. Despite these different trends of upregulated and downregulated DEGs in H3.3K4M Hom mice, the resulting neurogenic phenotype is similar because we observed a decrease in transcription factors driving postmitotic differentiation in the MGE and an increase in genes promoting proliferation in the hypothalamus. This decrease in neurogenic genes and increase in NPC proliferative genes was also observed in the cortex *H3f3a/H3f3b* double KO mice[12].

What mechanisms could underlie these differential effects between the MGE and hypothalamus? First, there is evidence that critical fate-determining genes may be resistant to K-to-M mutations[112] and other genetic manipulations that target histone modifications[48]. This innate resistance of certain genomic loci likely varies between cell types. Second, H3K4me3 can alter chromatin state in trans by recruiting ATP-dependent chromatin-remodeling complexes, leading to opening of local chromatin and allowing gene transcription at distinct loci in different cells[113,114]. Third, the H3K4M mutation also has a strong binding affinity to several demethylases that target H3K4[115], which could differentially exacerbate this loss of methylation in distinct cell types. Fourth, in addition to blocking H3K4 methylation, the H3K4M mutation can lead to decreased acetylation of H3K27 (H3K27ac)[14], increased H3K27 methylation[116], and decreased protein levels of Mll3/KMT2C and Mll4/KMT2C, two methyltransferases that target H3K4[38]. Haploinsufficiency of *MLL4* in humans results in Kabuki syndrome, whose symptoms include short stature, ID and microcephaly[117,118]. In *Mll4*[+/-] mice, growth hormone-releasing hormone (GHRH)-neurons in ARC are particularly affected because ~90% of these neurons express *Mll4*, whereas a much lower proportion of other ARC and other hypothalamic neurons express *Mll4*[119]. Thus, depletion of Mll3 and/or Mll4 could affect growth-regulating cells in the hypothalamus more so than other cell types.

ARC integrates hormonal and nutrient signals to regulate feeding and metabolism[120]. It receives information through from the ME as well as direct input from tanycytes lining the third ventricle[121,122]. Leptin-sensing ARC neurons project to the PVN to promote or inhibit food intake[123]. Serum leptin levels are higher in the satiated state and are correlated with body fat weight[124], while an increase in leptin levels along with obesity in the H3.3K4M Hom mice indicates leptin resistance in these mice, which is found in most obese individuals[125]. Notably, we observed the largest downregulation of ARC-fated neurons in the embryonic hypothalamus of H3.3K4M Hom mice compared to other hypothalamic nuclei.

Hypothalamic astrocytes also play important roles in energy metabolism by sensing and transporting nutrients in the interface blood vessels and neurons[44]. We observed an increase of hypothalamic astrocytes in H3.3K4M Hom mice, as well as a clear segregation of WT and Hom astrocytes that was not observed in any other cell type. Astrocyte DEGs are associated with cell adhesion, lipid metabolic processes, and glutamate or GABA receptor function, suggesting disruption of multiple astrocytic functions in mutant mice. In addition, we observed a striking disorganization and altered morphology of tanycytes and glia cells in the hypothalamus of H3.3K4M mice. Tanycytes can act as neural progenitor cells, generating new neurons and astrocytes[90], and they control leptin transport into the third ventricle[126]. Taken together, these cellular and genetic changes likely underlie the increased food intake and fat weight gain leading to obesity observed in the H3.3K4M Hom mice. A similar growth trajectory of initially underweight mice eventually developing obesity was observed in another H3 point mutation (H3.3G34W)[111]. This could indicate a particular sensitivity of the hypothalamic feeding circuit to manipulation of histone modifications, which may explain why so many diseases related to altered histone modifications demonstrate short stature and other growth defects[3,4].

Mouse models using different strategies to genetically perturb H3K4 methylation are associated with abnormal behaviors, including deficits in learning and memory, impaired sociability, severe hyperactivity, ASD-like behaviors, ID and epilepsy[9,21–23,127]. Here, the H3.3K4M Hom mice display abnormal locomotor and gait activity, increased anxiety, impaired spatial memory and sensorimotor gating, hypoactivity and impulsive activity (ADHD-like phenotypes) compared with WT. It is well-established that many NDDs affect one sex more than the other[114,128–131]. We found that H3.3K4M Hom female mice displayed more severe deficits in many of these behaviors compared to males. While males and females have a similar prevalence of having epilepsy, females have a greater risk for generalized-onset epilepsies, such as juvenile myoclonic epilepsy and juvenile absence epilepsy, with differential sex hormone control by the hypothalamus as one strong candidate driving these differences[132]. In addition, there is evidence for differential developmental expression and function GABA-related signaling between males and females that may contribute to increased female susceptibility in H3.3K4M Hom mice[133,134].

NDDs such as autism, schizophrenia, ADHD, ID, and various learning and motor disorders are more prominently diagnosed in males[128,135], whereas obsessive-compulsive-like tic behaviors, anxiety, bipolar disorder, depression and personality disorders are more prominent in females[136,137]. Female H3.3K4M Hom mice displayed greater deficits in numerous motor-related assays compared to males. Stimulation of Nkx2.1+ neurons in the ventromedial hypothalamus (VMH) generated a female-specific increase in movement, and loss of these cells resulted in females becoming obese, but not males[138]. In addition, the MGE gives rise to interneurons and GABAergic projection neurons that populate the basal ganglia, a structure critical for voluntary movements, which shows sexually dimorphism in humans[139,140], and specifically with MGE-derived interneuron distribution in the striatum[141]. The preoptic area in the hypothalamus and the bed nucleus of the stria terminalis (BNST), both of which contain Nkx2.1-lineage cells, display sexual dimorphism[142]. Notably, these regions also display sex differences in H3K4me3 levels, with ~70% of differential peaks enriched in females, and many at genomic loci associated with seizures, emotion/affective behavior, and learning and memory[131]. In addition, H3K4me is correlated with escape from X inactivation[143], and sex steroid hormones such as estrogens and androgens can directly affect H3K4 methylation. For example, the methyltransferase MLL2 methylates H3K4 but is also a coactivator for the estrogen receptor α (ERα) and is recruited to promoters of ERα target genes upon estrogen stimulation[144]. Thus, a bias for H3.3K4M Hom female-enriched deficits in various locomotor and behavior assays likely arises from differential transcriptome, cellular composition and circuit alterations in the hypothalamus and MGE-derived GABAergic cells.

One advantage of our study is targeting the H3.3K4M mutation to the two brain regions most associated with H3K4 methylation disease-

related phenotypes, as this replicates many of these disease phenotypes. However, this approach makes it challenging to definitively assign causation to specific cell types or brain regions. For example, the link between interneuron dysfunction and seizures is quite clear and supported by our data. But we do observe differences in spontaneous seizures & seizure susceptibility between males and females, as well as significant differences between sexes in many behavior assays. It is unclear whether these sex-specific effects are due to differences in interneuron function between H3.3K4M Hom males and females, hormonal changes arising from H3K4 methylation effects in sexual dimorphic regions of the hypothalamus, non-cell autonomous changes in response to altered H3K4 methylation in *Nkx2.1*-lineage cells, or some combination of these effects.

Similarly, transcriptional changes and altered cellular organization in the hypothalamus of H3.3K4M Hom mice is the most likely cause of altered feeding behavior, body weight and other metabolic processes. However, it's possible that the altered function of MGE-derived interneurons also plays a role in these processes. For example, disrupting the activity of SST + interneurons in the gustatory cortex can alter taste sensitivity and sucrose consumption[145], which can lead to changes in body weight.

In addition, we have shown alterations in glial cells in both the hippocampus and hypothalamus. Whether these changes arise from cell-autonomous effects during gliogenesis of *Nkx2.1*-linage glia or instead are non-cell-autonomous changes arising from altered H3K4 methylation in *Nkx2.1*-lineage neurons remains unclear. As MGE-derived interneurons, hypothalamic cells, and glia are intricately involved in nearly every aspect of brain function, it's likely that no single cell type or brain region are solely responsible for specific phenotypes observed in these mice.

## Methods

### Animals
All experimental procedures were conducted in accordance with the National Institutes of Health guidelines and were approved by the NICHD Animal Care and Use Committee (protocol #23-047). The following mouse lines were used in this study: *LSL-K4M* (gift from Dr. Kai Ge, NIDDK)[38]; *Nkx2.1-Cre* (Jax# 008661)[41], *Lhx6-Cre* (Jax# 026555)[146], *Ai9* (Jax# 007909)[147], *Sun1-sfGFP* (Jax# 030952)[148]. In the *LSL-K4M* mouse, the transgene construct is CAG promoter-LoxP-Stop-LoxP-H3.3K4M-FLAG/HA-polyA, using the human H3.3b gene sequence (https://www.addgene.org/128743/). The insertion site of the transgene is unknown. *LSL-K4M* mice were genotyped with forward primer (5'-GGGTCTGTTCGAAGATACCAACC-3') and reverse primer (5'-GTAGTCGGGCACGTCGTA-3'), with heterozygous and homozygous mice determined via qPCR. Mice were housed under standard conditions (12 h light and 12 h dark). The morning on which the vaginal plug was observed was denoted E0.5.

### Brain harvesting
Embryonic brain dissections: Pregnant dams were anesthetized with an i.p. injection of Euthasol (270 mg/kg, 50 µL injection per 30 g mouse), E13.5 embryos were removed and placed in ice-cold carbogenated artificial cerebral spinal fluid (ACSF, in mM: 87 NaCl, 26 NaHCO₃, 2.5 KCl, 1.25 NaH₂PO₄, 0.5 CaCl₂, 7 MgCl₂, 10 glucose, 75 sucrose, saturated with 95% O₂, 5% CO₂, pH 7.4). Embryonic tails were cut for genotyping. MGE and hypothalamus were dissected from individual brains in ice-cold ACSF, transferred to individual 0.5 mL Eppendorf tubes, then flash frozen on dry ice and stored at −80 °C. For analyzing fixed tissue, E13.5 brains were removed and drop-fixed in 4% paraformaldehyde (PFA) in PBS overnight, washed in PBS, incubated in 30% sucrose in PBS at 4 °C overnight, embedded in OCT and stored at −80 °C.

Adult brain dissections: P21-P150 mice were anesthetized with Euthasol (270 mg/kg, 50 µL injection per 30 g mouse). For harvesting brain regions, the somatosensory cortex and hypothalamus were

dissected, transferred to 1.5 mL Eppendorf tubes, then flash frozen on dry ice and stored at −80 °C. For analyzing fixed brains, mice were perfused with PBS followed by 4% PFA, brains were removed and post-fixed in 4% PFA overnight, washed in PBS, incubated in 30% sucrose in PBS at 4 °C overnight, then embedded in OCT and stored at −80 °C.

### Tissue staining and microscopy
P21-P150 brains were cryosectioned at 30 µm, sections transferred to 96-well plates containing antifreeze solution (30% ethylene glycol, 30% glycerol, 40% PBS) and stored at −20 °C. Prior to staining, brain sections were washed in PBS (3 × 10 min). Brain sections were incubated in block solution (10% Normal Donkey Serum in PBS + 0.3% Triton X-100) for 1 hour RT, then incubated for 24–48 hour in primary antibodies in block solution at 4 °C. Then sections were washed in PBS (4 × 10 min) and incubated with secondary antibodies and DAPI for 2 h at RT or overnight at 4 °C. Brain sections were washed and mounted.

E13.5 embryonic brains were cryosectioned at 14 µm, mounted on slides, dried and stored at −20 °C. Slides were incubated in block solution for 1 h RT, then incubated overnight with primary antibodies in block solution and secondary antibodies for 1-2 h RT.

The following primary antibodies were used in this study: rat-anti SST (1:300, Millipore MAB354), goat-anti PV (1:1000, Swant PVG213), rabbit-anti nNos (1:500, Millipore MAB5380), rabbit-anti H3K4me3 (1:500, Cell Signaling TECHNOLOGY 9751S), rat-anti HA (1:200, Roche 11867423001), mouse-anti GFAP (1:500, Sigma-Aldrich G3893), rabbit-anti NeuN (1:500, abcam ab177487), rabbit-anti Vimentin (1:500, Cell Signaling TECHNOLOGY 5741 T), rabbit-anti Olig2 (1:500, Sigma-Aldrich AB9610), mouse-anti MBP (1:1000, Invitrogen MA1-10837), rat-anti Ki67 (1:100, Invitrogen 14-5698-82), rabbit-anti Phospho-Histone H3 (Ser10) (1:400, Invitrogen PA5-17869), rabbit-anti Cux1 (1:300, Proteintech 11733-1-AP). Species-specific fluorescent secondary antibodies (1:500) were used conjugated to AlexaFluor 488, 647 and 790.

Nissl Staining: Slides were warmed to RT and then treated in the following solution sequence for 5 min each: 70% EtOH, 95% EtOH, 100% EtOH, 95% EtOH, 70% EtOH and 50% EtOH. Slides were then rinsed with ddH₂O, incubated in 0.1% warm crystal violet stain for 10–20 min, then washed with ddH₂O. Slides were then incubated in 70% EtOH and 95% EtOH for 5 min each, then 100% EtOH for 2 min twice. Slides were then incubated with a xylene for 5 min twice, air dried overnight and mounted with coverslips.

Fluorescent in situ Hybridization: Brains were cryosectioned at 14 µm. RNAscope Multiplex Fluorescent v2 in situ hybridization assays (Advanced Cell Diagnostics) were performed according to the manufacturer's instructions. All probes and reagents from Advanced Cell Diagnostics. 1: probe-Mm-Reln-C3 (405981-C3), probe-Mm-Pvalb (421931) and tdtomato-C2 (317041-C2); 2: probe-Mm-Sst-C3 (404631-C3), probe-Mm-Plpp4 (471731) and tdtomato-C2 (317041-C2); 3: probe-Mm-Fgf14-C3 (462801-C3), probe-Mm-Vim (457961) and tdtomato-C2 (317041-C2).

All images were captured on an Olympus VS200 scanner (VS200 ASW) or Zeiss Axioimager.M2 (with Zen Blue software). Post-processing was performed with Adobe Photoshop or ImageJ.

### Electrophysiology experiments
Adult mice were anesthetized with isoflurane and decapitated. Brains were quickly dissected and placed in ice-cold artificial cerebrospinal fluid (ACSF) modified for dissection containing (in mM): 80 NaCl, 25 NaHCO₃, 10 glucose, 1.25 NaH₂PO₄, 3.5 KCl, 4.5 MgCl2, 0.5 CaCl₂, 90 sucrose (Sigma-Aldrich) and bubbled with carbogen (95% O₂ and 5% CO₂). Horizontal hippocampal sections were prepared from the two hemispheres using a Leica VT1200S vibratome (Leica Microsystems). Recordings were performed using a Multiclamp 700B amplifier and acquired using pCLAMP 10,4 software (Molecular Devices). Signals were acquired at 5 kHz, digitized, and stored using Digidata 1322 A and pCLAMP 10,4 software (Molecular Devices, CA, USA).

## Ex vivo gamma oscillation recordings and analysis

Nine 3-month-old mice were used for local field potential (LFP) recordings (WT $n = 4$ and Hom $n = 5$ with both male and female). 350 μm thick hippocampal slices were placed into an interface holding chamber containing ACSF modified for LFP: 124 mM NaCl, 30 mM NaHCO₃, 10 mM glucose, 1.25 mM NaH₂PO₄, 3.5 mM KCl, 1.5 mM MgCl₂, 1.5 mM CaCl₂ (Sigma-Aldrich) continuously supplied with humidified carbogen gas (95% $O_2$ and 5% $CO_2$). The chamber was held at 34 °C during the slicing process and subsequently allowed to cool down to room temperature ( ~22 °C) for at least 1 h to let the slices recover. Hippocampal gamma oscillations experiments were performed as described previously[149]. Oscillations were recorded at 34 °C using borosilicate microelectrodes (1.5–2.5 MΩ) filled with ACSF placed in the CA3 stratum pyramidale and elicited by adding kainic acid (400 nM; Tocris Bioscience) to the extracellular bath. The oscillations were allowed to stabilize for at least 20 min before recordings were performed. Signals were conditioned using a HumBug 50 Hz noise eliminator (Quest Scientific).

Before analysis, signals were low pass filtered at 0.1 KHz and high pass filtered at 10 Hz. For oscillation power spectra, Fast Fourier Transformations were obtained from 60 s of LFP recording (segment length 8192 points). Frequency variance data was obtained from the power spectra described above using the Axograph software (Kagi, Berkeley, CA, USA). Gamma power was calculated by integrating the power spectral density from 20 to 80 Hz using Clampfit 11.2.

## Whole-cell patch clamp recordings and analysis

Six 8-week-old mice ($n = 3$ WT and Hom female brains) were used to perform whole cell patch clamp recordings of MGE derived hippocampal interneurons. 300 μm thick hippocampal slices were immediately placed into a submerged holding chamber containing standard ACSF: 130 mM NaCl, 24 mM NaHCO₃, 10 mM glucose, 1.25 mM NaH₂PO₄, 3.5 mM KCl, 1.5 mM MgCl₂, 2.5 mM CaCl₂ (Sigma-Aldrich) continuously supplied with humidified carbogen gas (95% $O_2$ and 5% $CO_2$). Whole-cell patch clamp recordings were performed on slices in the submerged recording chamber at 34 °C with borosilicate glass electrodes filled with intracellular recording solution containing 130 mM K-gluconate, 5 mM KCl, 10 mM HEPES, 3 mM MgCl₂, 2 mM Na₂ATP, 0.3 mM NaGTP, 0.6 mM EGTA, and 0.2% biocytin. Voltage clamp and current clamp configurations were used.

Analysis of whole cell patch clamp recordings was performed using the AutoAnt Software[150]. Principal Component Analysis (PCA) was used to classify 93 interneurons (WT $n = 49$ and Hom $n = 44$) based in their intrinsic properties (Input resistance, membrane time constant, membrane capacitance, sag ratio, maximum firing rate and action potential half-width). The PCA was performed in Python with the Sklearn package. The kmeans clustering algorithm was used to divide the standardized data using 3 principal components. The adequate number of clusters for the dataset was identified using the Silhouette Score.

## Pentylenetetrazole (PTZ) injection and recording

PTZ was prepared on the day of use with in sterile 0.9% (w/v) NaCl. PTZ injection and recording was performed on 3-month and 5-month-old mice. All 5-month animals will receive an IP injection of PTZ (Sigma-Aldrich, P6500) at 20 mg/kg and were recorded immediately via video tracking system ANY-maze (Stoelting Company) for 60 min. 3-month female animals will receive an IP injection of PTZ at 20 mg/kg, while 3-month male animals will receive an IP injection of PTZ at 40 mg/kg due to decreased PTZ sensitivity. Epileptic behaviors were scored based on the modified Racine scoring system[151]: stage 0 = normal behavior; stage 1 = immobility and rigidity; stage 2 = head bobbing, facial, forelimb, or hindlimb myoclonus; stage 3 = continuous whole-body myoclonus, myoclonic jerks or tail held up stiffly; stage 4 = tonic seizure, continuous rearing and falling, stage 5 = clonic-tonic seizure, stage 6 = death. Total seizure scores were calculated at every 5 min

block in the first 20 min. The mortality was also noted during the recording process.

## Behavioral tests

All behavioral tests were performed on WT, Het and Hom littermates according to approved NICHD and/or NIMH standard operating procedures (SOPs) and cleaning protocols. Animal behavioral tests were carried out during the daylight cycle in the NICHD animal behavioral room or at the NIMH Rodent Behavior Core. All mice were habituated to the testing environment for 30–60 min before behavior tests. All behavior assays were performed blind to genotype, and 1-3 independent cohorts were used for each behavioral test. The sequence of tests was as follows, with at least one week elapsed between any two behavior tests. Juvenile and young adults (6–11 weeks old): Home-cage test (5-6 weeks old); Open-field test (7-8 weeks old); Three-chamber test (9-10 weeks old); Barnes maze test (10-11 weeks). Adults (3–5 months old): Free-walk test; Light-dark box test; Novel object recognition test; Cliff avoidance reaction; Prepulse inhibition startle test.

## Home-cage test

Individually housed juvenile mice were observed in their home cages placed in the Photobeam Activity System-Home Cage (PAS-HC, SD Instruments) to assess activity (ambulation, fine movements and rearing) that can be quantified by beam breaks. Mice were kept in the cage for 4 days with appropriate food and water. During this testing, mice were housed in a 'quiet' room to minimize outside interactions. Mice were returned to group housing after the experiment. The total ambulatory movements, fine movements, rearing and general locomotor activity were analyzed and processed using the PAS-HC data collection transferred to Excel. $n$ (female) = 8 WT and 11 Hom; $n$ (male) = 7 WT and 4 Hom.

## Open-field test

Mice were originally placed in the center of open field apparatus (40 cm × 40 cm x 30 cm Plexiglas box) and recorded for 25 minutes. The average speed and time spent in the different open field areas were recorded and analyzed with the video tracking system ANY-maze (Stoelting Company). $n$ (female) = 13 WT, 8 Het and 11 Hom; $n$ (male) = 12 WT, 11 Het and 7 Hom.

## Three-chamber test

Tests were performed in an opaque Plexiglas rectangular box (60 cm × 45 cm x 20 cm) consisting of 3-chambers, with clear dividing walls and an open central chamber allowing free access between chambers. Wire cups were used to hold a single stimuli mouse (C57BL/6 J mice that were sex and age-matched with test mice). The test consists of three 10-minute phases that were all recorded using the ANY-maze system. For phase 1, the test mouse was placed in the center chamber with closed walls. For phase 2, two empty wire cups were placed in each side chambers and the closed walls were opened, which let the mouse explore all three chambers. For phase 3, the test mouse was confined in the center chamber and a stimuli mouse was placed in a wire cup on one side of the chamber. Both walls were removed, and the test mouse explored chambers for 10 minutes. Sniffing (investigating) was defined as being within 3 cm of either cup's edge. The total time spent in each region and time investigating each stimulus (stimuli mouse or empty cup) was recorded and quantified. $n$ (female) = 13 WT, 6 Het and 11 Hom; $n$ (male) = 12 WT, 11 Het and 7 Hom.

## Barnes maze test

The Barnes maze consists of a white elevated circular platform (diameter = 122 cm) with 80 cm support frame containing 20 equally distributed holes (diameter = 5 cm), with 1 'escape' hole leading to a

hidden black drop box. A light source was affixed above the platform to brightly illuminate the maze, and 90 dB white noise generated from loudspeakers was used to motivate mice to enter the dark escape box. Three distal visual cues surrounded the platform.

The protocol used a 6-day paradigm that included a habituation, training and probe trial phase. During habituation (day 1), the mouse was placed in the escape tunnel for 1 min and then placed in the center of platform until it enters the escape hole or 3 minutes elapse. In the training days (days 2–5), the mouse is placed in a start chamber located in the center of one of the four quadrants, and the mouse explores the platform until it enters the escape hole or 3 min elapse. If a mouse fails to find the escape tunnel within 3 min, the mouse is guided to the dark escape tunnel and remains there for 15 s. Mice underwent 4 trials on each training day. For the probe trial (day 6), the escape tunnel is removed, and mice are placed in the center of the platform to explore the maze for 90 seconds. Time spent around the target hole was defined as being within 1 cm of the target hole edge. All trials were recorded using the ANY-maze system allowing quantification of the time each mouse spent in the target hole and target quadrant. $n$ (female) = 13 WT and 9 Hom; $n$ (male) = 8 WT and 4 Hom.

### Free-walk test
FreeWalkScan 2.0 (CleverSys Inc.) was used to assess mice gait. 3-month-old mice with similar body weight move freely in a 40 cm × 40 cm × 30 cm (length × width × height) chamber. A high-speed camera below a clear bottom plate captures mouse movement for 5 minutes with red light in a dark room. Videos are analyzed using FreewalkScanTM 2.0 software for various characteristics of gait, including BaseFront (distance between sequential footprints of front limbs), StrideLength (distance between two sequential footprints of the same paw), Contact Size, Average Pressure and Stance Time (time corresponding paw stays on ground) of each paw. $n$ (female) = 9 WT, 9 Het and 4 Hom; $n$ (male) = 6 WT, 9 Het and 8 Hom.

### Light-dark box test
The apparatus (40 cm × 40 cm x 30 cm) consists of dark chamber (40 cm × 15 cm x 15 cm) and light chamber compartment. There is a door between two chambers which allow mice to freely move between chambers. Bright white lights illuminated the box from above as mice were placed in the box for 10 min. The test was recorded by an ANY-maze video trace camera to determine the amount of time each mouse spent in the light chamber. $n$ (female) = 14 WT, 12 Het and 7 Hom; $n$ (male) = 9 WT, 11 Het and 9 Hom.

### 2 Object Novel object recognition test
Novel object recognition test was conducted in an open field arena (40 cm × 40 cm white Plexiglas box). The protocol used a 3-day paradigm that includes habituation, training, and a testing phase. During habituation (day 1), mice were placed into the center of an open field arena and allowed to explore for 10 minutes. For training (day 2), two identical objects were placed on either side of the center of the arena, and the mice explored the arena for 10 minutes. For testing (day 3), a familiar object and a novel object (dissimilar object) were placed in the same position as in the training day, and mice explored the arena for 10 minutes. ANY-maze was used to record each trial. The amount of time each mouse spent with their nose oriented toward the object within 2.5 cm of the object edge was considered 'exploration time'.

The recognition index was calculated as: Recognition index = ( time spent exploring novel − time spent exploring familiar)/(total time spent exploring both objects). $n$ (female) = 9 WT, 10 Het and 7 Hom; $n$ (male) = 6 WT, 7 Het and 6 Hom.

### Prepulse inhibition (PPI) of the acoustic startle reflex
Startle response and pre-pulse inhibition were performed via SR-LAB-Startle Response System (San Diego Instruments). Mice are placed in a startle apparatus with sound-attenuated chamber. Whole-body startle movements were detected by a piezoelectric accelerometer mounted beneath the cage, converted to electrical signals, and digitized and stored by a computer. A loudspeaker mounted to the side of the cage produced white noise acoustic stimuli. A continuous 64 dB background noise was present throughout the test. The test began with a 5-minute acclimation to the apparatus, followed by 12 pulse-alone trials. To measure PPI, 12 blocks containing 6 trials were presented: 1 pulse-alone trial, 4 prepulse-pulse trials, and 1 no-stimulus trial. Pulse-alone trials consisted of a 40 ms startle pulse at 120 dB; prepulse-pulse trials consisted of a 20 ms prepulse at 66, 68, 72, or 76 dB followed 100 ms later by the pulse; and no-stimulus trials consisted of background noise only. Trials will be presented in pseudorandom order, with an inter-trial interval of 10–20 s (mean of 15 s). Startle response will be determined as the peak amplitude within 100 ms of the startle pulse onset. The entire test lasted 30 minutes. $n$ (female) = 10 WT and 7 Hom; $n$ (male) = 6 WT and 5 Hom.

### Cliff avoidance reaction (CAR) test
The CAR test was performed as described previously[65] with slight modifications to assess impulsive-like behaviors. The apparatus consists of a square platform (20 cm×20 cm) and support by a rod (Height = 40 cm) that was placed in an open field box with soft padding. After habitation, the mouse was gently placed onto the platform, and they were also placed back onto the platform after falling. The latency to first jump off and numbers of jump off were recorded. The test was performed and analyzed for 15 min under white lights. $n$ (female) = 13 WT and 7 Hom; $n$ (male) = 5 WT and 5 Hom.

### Body composition and Elisa analysis
Analysis of body composition in live mice using the EchoMRI100 analyzer (Echo Medical Systems) in the Mouse Metabolism Core of NIDDK. This test revealed both the fat weight and lean weight for each mouse. Serum leptin levels were measured using a mouse leptin ELISA kit (ThermoFisher, #KMC2281) in accordance with their standard protocol.

### Cell counting and analysis
All cell counts on *Nkx2.1-Cre;K4M;Ai9* mice were performed manually using Photoshop and were blind to genotypes. Cortex: Cells were counted from 3 non-consecutive brain sections through the somatosensory cortex for each P21 brain, from 7 WT (4 male and 3 female), 4 Het (2 male and 2 female) and 8 Hom (4 male and 4 female) mice. DAPI staining was used to divide the cortex into superficial (I-III) and deep (IV-VI) layers. Tom+ , PV+ and SST+ cells were counted, and the density was calculated by cells/mm². Any PV+ and SST+ cells that were Tom-were excluded from the counts. Hippocampus: Cells were counted from 4 non-consecutive sections spanning the anterior-to-middle P21 dorsal hippocampus, from 7 WT (4 male and 3 female) and 8 Hom (5 male and 3 female) brains. Tom+ , PV+ , SST+ and nNos+ cells counted, and the density was calculated by cell numbers/mm². Small Tom+ cell bodies in CA2/3 that are primarily Olig2+ oligodendrocytes were excluded from these interneuron counts and counted as a separate group. The thickness of the primary somatosensory cortex (S1) (bregma: from −1.67 to −1.79 mm) and the primary visual cortex (V1) (bregma: from −2.69 to −2.91 mm) from P21 mice were measured. Cux1 immunostaining was used to label layers II-IV. 3 mice per genotype, 5–8 sections measured per brain. Hypothalamus: Six coronal brain sections (bregma: from −1.58 to −1.94 mm) were analyzed per mouse. The mean fluorescent intensity of GFAP in the region of interest was measured using ImageJ software (v1.53; NIH) from 4 WT and 3 Hom of adult mice (3-5 months old). Median eminence (ME): The mean fluorescent intensity of MBP was measured using ImageJ software from 4 WT and 3 Hom of ~P60 adult mice, while the intensity of Sst was measured from 5 WT and 4 Hom of P21 mice. MGE: PH3+ cells in the E13.5 MGE (3 WT, 3 Het and 4 Hom, combined male and female

samples, 3–6 brain sections per mouse) and Tom+ cells in the E13.5 cortex (4 WT, 3 Het and 4 Hom mice, combined male and female samples, 4 brain sections per mouse) were counted using ImageJ software. Cell counts from *Lhx6-Cre;K4M* hippocampus were performed from 3 WT and 3 Het of P30 mice (3-5 sections per mice).

## TUNEL apoptosis assay

For detecting DNA fragmentation by fluorescence microscopy, we used the ApoBrdU-IHC DNA fragmentation assay kit (Abcam, ab66098). Apoptosis signals were detected in accordance with standard protocol for tissue sections, including sections preparation, DNA labeling and antibody detecting. Brain tissues with mixed sex from E13.5 and P4 WT and Hom mice were used.

## Western blot

Total core histone proteins were extracted using the EpiQuik Total Histone Extraction Kit (EpigenTek). Total proteins were extracted using the RIPA buffer (ThermoScientific). E13.5 MGE and hypothalamus samples were obtained as described above. Protein amount was qualitied using Pierce BCA Protein Assay Kit (ThermoScientific), typically ~35 µg protein was obtained from tissue from 1 embryo. Total 40 µL Loading samples were prepared with Sample Reducing Agent (ThermoFisher, B0009) and Bolt LDS Sample Buffer (ThermoFisher, B0007) and were load on a 4–12% Bolt Bis-Tris Plus Mini Protein Gel (ThermoFisher, NW04120BOX), running 20–30 min at 200 V with Blot MES SDS Running buffer (ThermoFisher, B0002). After that, gel was transferred to PVDF membranes using the iBlot 2 Gel Transfer Device (Invitrogen) and transferred for 7 min at 20 V. Membrane was blocked 30 min in block buffer (SuperBlock Dry Blend Blocking Buffer, ThermoFisher #37545). Membranes were incubated in primary antibody overnight at 4 °C, secondary antibody for 1 h at RT, washed 3 times with TBST washing buffer, and then blots were imaged on the ChemiDoC MP Imaging System (Bio-Rad). The following primary antibodies were used: mouse anti-H3 (1:1000, Cell Signaling Technology #3638), rabbit anti-H3K4me3 (1:500, Cell Signaling Technology #9751), rabbit anti-H3K4me1 (1:1000, Abcam ab8895), rabbit anti-H3K4me2 (1:500, Abcam ab7766), hFAB rhodamine anti-Tubulin (1:2000, Bio-rad #12004166), rabbit anti-H3.3 (1:1000, Invitrogen #PA5-22388). The following secondary antibodies were used: anti-mouse-Starbright Blue-520 (Bio-Rad# 64456855; 1:2000) and anti-Rabbit-Starbright Blue-700 (Bio-Rad# 64484700; 1:2000).

## Multiome sequencing and analysis

Pregnant dams and P60 mice were terminally anesthetized with an i.p. injection of Euthasol (270 mg/kg, 50 µl injection per 30 g mouse). Tissue from male and female mice were processed separately for E13.5 and P60 samples. Frozen E13.5 and P60 tissue from *Nkx2.1-Cre;H3.3K4M;sun1-GFP* mice were mechanically lysed, washed and filtered to generate single nuclei suspensions for the 10x Genomics Multiome assay as previously described[152]. These suspensions were then sorted via flow cytometry (Sony SH800) to harvest ~100,000 GFP+ nuclei. The percent of GFP+ nuclei/Draq5+ nuclei was recorded for all sorts to determine the average percent of GFP+ nuclei for each condition. Nuclei solutions were centrifuged in a bucket centrifuge at 500 × *g* for 5 min at 4 ºC. Supernatant was removed and nuclei reconstituted in 1x Nuclei buffer at a density of 3000 nuclei/ml. 5 µl of this solution was then used for the 10x Genomics Multiome Assay per manufacturer's protocol. Our single-cell Multiome workflow 'multiome-wf' can be found at GitHub (https://github.com/NICHD-BSPC/multiome-wf) with documentation located here (https://nichd-bspc.github.io/multiome-wf/). Sample details and sequencing information are described in Supplementary Data 1.

Sequencing analysis: Gene expression and open chromatin state were profiled using 10x Chromium Multiome ATAC + Gene Expression kit (10x Genomics, 1000285). Joint libraries were simultaneously created by following the standard protocol. Sequencing was conducted with paired-end (50 × 50 bp) using an Illumina HiSeq 2500 or NovaSeq 6000. The raw sequencing data were processed using the Cell Ranger ARC (v2.0.0) pipeline. The cellranger-arc mkfastq command was used to generate the demultiplexed FASTQ files from BCL files. The sequencing reads were aligned to custom built mouse (GRCm38/mm10) reference genome using cellranger-arc count. The cellranger-arc count command was used to generate gene-by-barcode (snRNA-seq) and peak-by-barcode (snATAC-seq) matrices. The cellranger-arc aggr command, without depth normalization (--normalize = none) was used to aggregate sample datasets into a single feature-barcode matrix file.

snRNA-seq data analysis: The aggregated feature-barcode matrix was used as input to Seurat (v5.2.1)[153] in R (v4.4.2, https://cran.r-project.org). Low-quality cells were removed based on the following QC metrics: RNA feature counts, UMI counts, mitochondrial counts and ribosomal counts. Cells that have RNA feature counts more than 100 and UMI counts less than three standard deviations (SD) of the mean were kept. Cells that have less than 5% mitochondrial counts and 10% ribosomal counts were kept. In addition, R package scDblFinder (v1.20.0)[154] was used to remove the computationally likely doublets. After filtering out low-quality cells and doublets, we performed the standard Seurat workflow with default parameters unless otherwise noted: NormalizeData (LogNormalize), FindVariableFeatures (nfeatures = 2000), ScaleData, RunPCA, FindNeighbors, FindClusters and RunUMAP steps. Dim was set to 30 based on inspection of variance across the principal components (PCs) using the ElbowPlot function.

snATAC-seq data analysis: The aggregated peak-by-barcode matrix was used as input to Signac (v1.14.0)[155] in R. Low-quality cells were removed based on the following QC metrics: number of chromatin accessibility peaks, nucleosome signal and transcription start site enrichment score. Cells with more than 1000 chromatin accessibility peaks, a transcription start site enrichment score greater than 2, and a nucleosome signal less than 4 were kept. Doublets were also removed by using scDblFinder (v1.20.0)[154]. After filtering out the low-quality cells and doublets, we proceeded with normalization, identified highly variable peaks and reduced dimensions using the function of RunTFIDF, FindTopFeatures (min.cutoff = "q0") and RunSVD in Signac with default parameters. The first Latent Semantic Indexing (LSI) component was not used to downstream analysis due to its strong correlation with the sequencing depth. Then, regular non-linear dimension reduction and clustering were performed by RunUMAP (dims = 2:30), FindNeighbors (dims = 2:30) and FindClusters in Seurat. The gene activity scores were calculated using the GeneActivity function and added to the Seurat object.

Multimodal analysis: The shared cells across the two modalities (snRNA-seq and snATAC-seq datasets) were merged, and the RNA and ATAC dimensionality reduction was recomputed using default parameters. Multimodal data integration was performed by finding the Weighted Nearest Neighbor (WNN)[156] using Seurat::FindMultiModalNeighbors. The dimensionality reduction was separately set to 1:30 (snRNA assay) and 2:30 (snATAC assay). The joint UMAP and clustering was performed using the WNN graph with default parameters.

Subcluster and annotation analysis: To better analyze the different tissue, cells belonging to E13.5 MGE, E13.5 hypothalamus, P60 cortex and P60 hypothalamus from total Multiome data were separately subsetted. Shared cells across the two modalities were merged, and the RNA and ATAC dimensionality reduction was recomputed using default parameters. Multimodal data integration was recomputed by finding the Weighted Nearest Neighbor (WNN)[156] using Seurat::FindMultiModalNeighbors. All cell types were defined using the RNA expression only. Cell types of E13.5 MGE were identified with the FindAllMarkers function within clusters and integrated with mature cell type biomarkers. Putative hypothalamic nuclei of E13.5

hypothalamus datasets were annotated via gene expression studies[77–79]. In E13.5 hypothalamus datasets, we identified some cells corresponding to the adjacent prethalamus and telencephalon area via region-specific markers[77], and the telencephalon cells were removed before performing other analysis. We mapped our E13.5 hypothalamus dataset to the Hypothalamic Development Database 2[80] with the TransferData function. P60 cortex datasets were annotated using the Allen Brain Altas scRNA dataset[157] with the TransferData function, and subtypes with low cell numbers were removed. PV+ and SST+ interneuron subtypes were annotated based on previous publications[71,76] and FindAllMarkers function. For P60 hypothalamus, Seurat clusters with low cells were removed and the remaining cells were annotated with mature cell type biomarkers. Astrocytes sub-clusters from P60 hypothalamus datasets were defined with default Seurat algorithm.

Differential analysis: Differentially expressed genes between genotypes and sexes were computed using the Wilcoxon rank sum test implemented in the Seurat FindMarkers function with Bonferroni correction (significance: $\log_2$FC > ± 0.2, adjusted $P$-value < $10^{-6}$) for the E13.5 MGE, E13.5 hypothalamus and P60 cortex datasets. For P60 hypothalamus, significant differentially expressed genes were defined with $\log_2$FC > ± 0.2, adjusted $P$-value < 0.05 due to low cell numbers. DEGs in tanycytes were analyzed with combined female and male samples due to low cell numbers. Differentially expressed peaks between different genotypes were computed using the Wilcoxon rank sum test implemented in the Seurat FindMarkers function with Bonferroni correction (significance: $\log_2$FC > ± 0.2, adjusted $P$-value < $10^{-6}$) with combined sex. The EnhancedVolcano function from the EnhancedVolcano package (v1.24.0, https://github.com/kevinblighe/EnhancedVolcano) was used to visualize the DEGs between different conditions per population and parameters.

Enrichment analysis: Gene ontology pathway analysis was performed on each group's (e.g., genotype and sex) DEGs with adjusted $p$-values < 0.05 using the enrichGO function from the clusterProfiler package (v4.14.0)[158]. The parameters were set as following: OrgDb = org.Mm.eg.db (v3.20.0), keyType = "SYMBOL", ont = "BP", universe = "All detected genes in specific population". ClusterProfiler::simplify function was also used to reduce the redundancy among enriched terms.

Co-expression network analysis: The weighted gene co-expression network analysis in high-dimensional transcriptomics (hdWGCNA) was performed by the R package hdWGCNA (v0.4.03)[73]. We set up Seurat objects for WGCNA using cells from P60 cortex interneurons with the SetupForWGCNA function and constructed metacells in each group with the MetacellsByGroups function. The expression matrix of the PV+ population from combined sex and genotypes was prepared with the SetDatExpr function. The co-expression relationships between genes were raised by the power to focus on strong connections. Thus, the soft power threshold was set to 6 by performing a parameter sweep with the TestSoftPowers function, and the co-expression in the PV+ population from combined sex and genotypes was constructed with the ConstructNetwork function. Module Eigengenes and connectivity were computed with default parameters in the PV population. Then, the top hub genes were determined with the GetHubGenes function. We ran the UMAP algorithm on the hdWGCNA topological overlap matrix with the RunModuleUMAP function and visualized the UMAP co-expression network with the ModuleUMAPPlot function in hdWGCNA. Differential Module Eigengene (DME) analysis was also performed between different genotypes in the PV population using the FindDMEs function (test.use = "wilcox") and visualized with the PlotDMEsVolcano function in hdWGCNA with default parameters.

Pseudo-time analysis: Single-cell trajectory analysis was performed using Monocle 3 (v1.3.7)[159] workflow. MGE Seurat objects with combined sex and genotypes were converted to a CellDataSet object using as.cell_data_set function. The reduce dimensionality of CellDataSet object was set to WNN.UMAP of MGE Seurat object. Learning trajectory graph using the learn_graph function. The root of the trajectory was set to *Nes+* clusters and order cells using the order_cells function with default parameters. Plotting trajectory colored by pseudotime with the plot_cells function.

Visualization: Uniform Manifold Approximation and Projection (UMAP) coordinates and WNN clustering, computed by Seurat on multimodal integrated datasets, were visualized using the DimPlot function. The expression of genes of interest was visualized using FeaturePlot, VlnPlot or DoHeatmap function. Packages ggplot2 (v3.5.1) and patchwork (v1.3.0) were used to visualize during the Multiome datasets analysis process.

### Statistics and reproducibility

Statistical tests were performed using GraphPad Prism and R. Unless otherwise stated, Standard Unpaired 2-tailed $t$ test, 2-tailed Welch's $t$ tests and 2-tailed Mann-Whitney test were performed based on normality when comes to two groups; Ordinary one-way ANOVA followed by Tukey's multiple comparison tests, Kruskal-Wallis followed by Dunn's multiple comparisons based on normality when comes to three groups; Two-way ANOVA followed by Tukey's multiple comparison tests or Šídák's multiple comparisons test when comes to multiple groups. Unless otherwise stated, data are presented as mean values +/- SEM. 2-tailed Wilcoxon rank sum test with Bonferroni correction was performed in Multiome-seq data, and 1-tailed Fisher's exact with BH correction was performed in GO enrichment data. In Figs. 1C, D, 2E, 5D, E, 10A, assays were repeated ≥ 3 biological replicates with technical replicates with similar results. In Fig. 10E, assays were repeated 2 biological replicates with technical replicates.

### Reporting summary

Further information on research design is available in the Nature Portfolio Reporting Summary linked to this article.

## Data availability

The single-cell sequencing data generated in this study have been deposited in the Gene Expression Omnibus (GEO) repository under Superseries accession code GSE293881, which includes our single-cell ATAC-seq (GSE293655) and single-cell RNA-seq (GSE293751) datasets, all of which are publicly available. Details on cell counts, number of animals used per experiment, and analyses of differentially expressed genes for all comparisons described in the manuscript are provided in the Supplementary Data files. Additional data & information used in this study are available at Synapse.org. For any additional inquiries about data accessibility and analysis, please email tim.petros@nih.gov.

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

## Acknowledgements

We thank all members of the Section on Cellular and Molecular Neurodevelopment, as well as Pedro Rocha, Ariel Levine and Kai Ge for discussion and comments on this project and manuscript. We thank Kai Ge (NIDDK) for the *LSL-K4M* mice. We thank the NICHD Molecular Genomics Core, specifically Fabio Faucz, Vivek Mahadevan, Tianwei Li and James Iben; the NIDDK Mouse Metabolism Core, particularly Oksana Gavrilova and Naili Liu, for assistance with measuring body composition; the NINDS and NICHD animal facility for mouse husbandry assistance. This work utilized the computational resources of the NIH HPC Biowulf cluster (http://hpc.nih.gov). This work was supported, in part, by the NIMH IRP Rodent Behavioral Core (MH002952). This project was funded by NICHD intramural projects HD008962 (T.J.P.), HD008986 (R.K.D.), HD001205 (C.J.M.); NICHD Scientific Director's Award (T.J.P); NICHD Intramural Research Fellowship (J.L.); NICHD Career Development Award (J.L.). This research was supported by the Intramural Research Program of the National Institutes of Health (NIH). The contributions of the NIH author(s) are considered Works of the United States Government. The findings and conclusions presented in this paper are those of the author(s) and do not necessarily reflect the views of the NIH or the U.S. Department of Health and Human Services.

## Author contributions

Conceptualization – J.L. and T.J.P.; Investigation – J.L., A.F.T., G.P., A.C., R.C., D.A., Y.Z., and K.A.P.; Formal analysis – J.L., G.P., K.A.P., and M.S.; Software – M.S.; Supervision – R.K.D., C.J.M., and T.J.P.; Funding acquisition – J.L., R.K.D., C.J.M., and T.J.P.; Writing, original draft – J.L. and T.J.P.; Writing, review & editing – all authors.

## Funding

## Competing interests

The authors declare no competing interests.
