## [Transparent Peer Review file · Nature Communications]

Reducing methylation of histone 3.3 lysine 4 in the medial ganglionic eminence and hypothalamus recapitulates neurodevelopmental disorder phenotypes

Corresponding Author: Dr Timothy Petros

Version 0:

Reviewer comments:

Reviewer #1

(Remarks to the Author)

Li and colleagues present a very thorough and well written study on the role of histone 3.3 lysine 4 methylation depletion upon neurological development and functional changes due to targeted gene manipulation in either the MGE or hypothalamic regions. The authors employed a strategic approach to mimic humans with loss of function mutations in this gene, by expressing a version of histone 3.3 that was incapable of having lysine 4 methylated and in turn competed with endogenous histone 3.3 like humans with mutations in this gene. The authors also studied dosage of this effect by assessing both het and homozygous expression of this interfering mutant. The extent of approaches to characterize these mice and the relevance to the human condition are paramount and I think the authors did a superb job but just have some suggestions to help better establish some mechanisms underlying their findings and other minor things that I hope will improve the manuscript below:

- 1) For Figure 1 text in the results section, I would like to see more detail in the text describing aspects of experiments detailed in panels D and E to help readers.
- 2) Just to be safe, in line 142 where it first comes up, can you change "MGE and hypothalamus" to "MGE and/or hypothalamus." Also make this edit later as this comes up a few times.
- 3) In Figure 2D the Y-axis values seem off, can you check and make sure the values reflect the weight in grams of the mice or change?
- 4) In Figure 2F, it is interesting that Nkx2.1-Cre manipulation of cortical interneurons would result in microcephaly, which is normally due to changes in pyramidal neuron morphology. Not asking for additional experiments here but this is an interesting and potentially novel phenomenon that should be mentioned/speculated in the discussion.
- 5) Starting in Figure 5, some panels are hard to see, including the Tom and PV images. Can the authors make these brighter in the figures?
- 6) In figure 4B, bottom table, the authors mention they used a higher dose of PTZ to induce seizures in males. Can the authors color-code or do something else to reflect this in the table? This would help readers with the presentation of the data.
- 7) In Figure 4E, this is a minor thing but could the authors correlate the language using epileptiform bursts mentioned in the text to the interictal events presented in Figure 4F? Just wanted the language to match so readers had an easier time understanding.
- 8) The loss of cortical interneurons is a major phenotype in these mice but how this occurs is not known. The authors have done a great job with their RNA-seq suggesting that there are less mutant MGE cells later in the paper but this is not yet tested. Can the authors look at embryonic WT and mutants to compare proliferation and apoptosis to better understand how this loss of cells is propagated? Maybe something as simple as either PH3 or SOX2 staining (proliferation) and cleaved caspase-3 (apoptosis) staining in the MGE, and potentially hypothalamus, to answer?
- 9) In lines 391-392 of the results section, it is mentioned that the hypothalamus has a reduction in GFP+ cells. Are there any volumetric/regional assessments that could be done to assess how this impacts the hypothalamus and various nuclei? It would be very interesting to see if any changes occur in distinct regions of the hypothalamus already implicated herein.

Reviewer #2

(Remarks to the Author)

In this manuscript, Li and colleagues described experiments in which they perturbed H3K4 methylation in the developing medial ganglionic eminence (MGE) and the hypothalamus of mice. To this end, they express (at different doses in heterozygous and homozygous mice) a mutated form of H3.3 in which the lysine 4 residue is mutated to a methionine. These mutant mice have persistent body weight and growth problems (initial undergrowth, followed by obesity), fewer GABAergic interneurons, abnormal network dynamics, and increased seizure susceptibility. Mutant mice also exhibit behavioural abnormalities, which were more prevalent in females than males. The authors also performed single-cell sequencing experiments to identify transcriptional changes that could underlie the cellular phenotypes described in mutant mice.

This paper demonstrates a critical role for H3K4 methylation in the development of the MGE and the hypothalamus of mice. Disrupting H3K4 methylation impacts gene expression during embryonic development, causing multiple cellular alterations that, in turn, affect the neuronal populations generated in the brain regions studied. However, while the paper does a very nice job at identifying phenotypes caused by the likely insertion of mutated histones in the chromatin (although this is not directly shown in this manuscript), there is little mechanistic insight into the process that H3K4 methylation regulates to cause the abnormalities. Are the phenotypes caused by abnormal patterning, neurogenesis, migration, survival, or wiring? Are the phenotypes independent or causally linked? Are the transcriptional changes sufficient to explain the cellular abnormalities?

Main issues

The authors report that disrupting H3K4 methylation in the MGE and hypothalamus has a massive impact on food consumption and, consequently, growth. With this in mind, it is difficult to rule out that some of the changes observed in the brain are not indirectly caused by abnormal metabolism and growth in these animals, rather than a direct effect on the methylation profile of H3.3 at specific genomic loci. In this context, while the single-cell analyses in embryos may hint at possible mechanisms underlying the phenotypes described by the authors, the data obtained from the adult mice are hard to interpret. Since the hypothalamic deficits are likely at the core of the metabolic alterations, one possible way to circumvent this problem is to express H3.3K4M only in the MGE and study what features this manipulation recapitulates from those observed in Nkx2.1-Cre mice.

The authors describe differences in seizure susceptibility between male and female mice. However, sex differences were not explored for GABAergic interneurons in the cortex, which the authors suggest are at the core of these problems. Are there differences in the density and relative proportion of interneurons in male and female mice? Are the analyses performed in Figure 3 powered to detect differences in SST interneurons? There is a clear trend in Figure 3B.

The characterization of hypothalamic deficits is primarily based on single-cell analyses. It would be appropriate to identify key cellular populations and examine cell densities, proportions, and other relevant characteristics using histology, which would help build a picture of the deficits more similar to what is described for cells derived from the MGE. This would allow the authors to explore common themes emerging from the analysis of the two brain regions. As it stands, it is unclear why studying the MGE and the hypothalamus simultaneously adds to our understanding of the role of H3K4 methylation in brain development and function.

Other issues

LSL-K4M mice are not described in any detail. What locus is it used to drive the expression of H3.3K4M? Does the exogenous expression of H3.3K4M influence the endogenous expression of H3.3K4? One would infer that the reduced methylation is caused by the availability of mutated H3.3 but, is it possible that levels of wild type H3.3 are changed in these mice? This could be explored through Western blots.

The authors should also carry out Western blots for me1 and me2 to quantify how the expression of H3.3K4M impact those levels in heterozygous and homozygous mice (as shown for me3 in Figure 1E).

The age of the embryos analyzed in Figure 1 is not included in the text or figure legends.

There are several statements at the end of some sections of the Results that belong to the Discussion. Some examples: (1) "Such increased variability in electrical properties is reminiscent of the developing hippocampus, suggesting interneuron maturation deficits in H3.3K4M Hom mice". This is speculative and not a direct conclusion of the experiments; there are several alternative explanations. (2) "As widespread perisomatic inhibition is critical for both circuit gamma entrainment and seizure control, it is likely that the deficits in hippocampal FS/PV+ cells in H3.3K4M mice directly contribute to disrupted rhythmogenesis and epilepsy". Again, this is speculative and not a direct conclusion of the experiments. The authors described defects in other interneuron populations that shape gamma rhythms. Indicating that PV interneurons cause the network dynamic changes and seizure susceptibility is a stretch, especially in the Results section. Please revise the Results to avoid these and similar statements.

Reviewer #3

(Remarks to the Author)

Summary of the Study

This manuscript investigates how reduced H3K4 methylation in the medial ganglionic eminence (MGE) and hypothalamus affects cortical interneuron (cIN) generation and function, as well as hypothalamic circuits controlling body growth. By using

a conditional H3.3K4M mouse model driven by Nkx2.1-Cre, the authors describe reductions in interneuron numbers, altered network excitability and seizures, as well as notable changes in postnatal growth and obesity. The data include morphological, behavioral, electrophysiological, and single-cell analyses that point toward significant, cell-type-specific deficits in both MGE-derived interneurons and hypothalamic populations.

Major Points

Validation of the mouse model:

The authors would need to verify or test the actual timing of Cre expression.

The reduction of H3K4me3 in MGE-derived cINs is not properly shown. First, I wonder why the tdTomato signal is so blurry. In numerous manuscripts, including Tanuguci et al., 2011 (DOI: 10.1016/j.neuron.2011.07.026), where Nkx2.1-Cre mediated reporter expression is shown in the embryonic MGE, discrete reporter positive cells are visible in the VZ and SVZ. However, the tdTom signal in Figure 1C and D is very blurry. This hampers proper visualization of analyzing H3K4me3 levels in the recombined cells. However, at adult stages (Fig. 3A) clear tdTom cells are visible. As Nkx2.1 is down regulated in cortical fated interneurons upon becoming postmitotic, while staying expressed in e.g. striatal fated cells, such discrete cellular signals would help to show downregulation in cortical fated cells, either by analyzing cells that have reached the cortex or by co-labeling e.g. with Calbindin antibodies. Also, the authors state in line 124: „Having H3.3K4M active prior to cell cycle exit is critical to substitute endogenous H3.3 with mutant H3.3K4M protein due to the prolonged half-life of histone proteins in the brain...“ However, there are no clear tdTom+ cells visible in the ventricular zone (VZ), where the progenitors reside. Also, the reduction of H4me2/3 level in the VZ is not properly visible (rather in the SVZ). I wonder, whether successful Cre-recombination could be shown in the sc-transcriptome analyses? This would also allow proper discrimination of progenitors versus postmitotic cINs (however, here a different mouse line was used, commented later).

For the hypothalamus, no immunostainings are shown in that regard, which should be provided as well. Here, I would similarly expect clear cellular tdTomato labeling, which allows determination and quantification of H3K4me3 fluorescence intensity in tdTom cells.

In addition, the authors provided Western Blot data, which showed a strong reduction for the hypothalamus in the HOM mice, but less clear reduction for the MGE. I assume from the methods, that bulk tissue was used here. Maybe FACS-mediated enrichment of the recombined cells would yield in clearer reductions in K4me3 levels.

Embryonic fate-mapping and characterization of potential defective processes is missing

What I miss in general in this manuscript is fate mapping of the cells at embryonic stages. By providing embryonic sections with proper tdTomato cell labeling not only allows the determination of H3K4me3 reduction in e.g. cells that have reached the cortex, it would also allow to assess which developmental defects occur that lead to reduced cIN numbers in the adult mice. Is the migration altered (e.g. misrouted)? Or do HOM-cINs die? At which stage (embryonically or postnatally)? Same for the reduced numbers of nNos+ interneurons seen in the hippocampus.

More comprehensive embryonic phenotyping would also serve to better understand the findings the authors report in postnatal / adult mice, such as the reduced cortical thickness in postnatal brains of mutant mice (Fig. 2). It is known that invading cINs can influence cortical progenitors (DOI: 10.1016/j.cell.2018.01.031), and thus potentially cortical neurogenesis and layer thicknesses. Embryonic fate-mapping could serve to investigate potential migration defects that correlate with changes in cortical progenitor proliferation and lamination. This would strengthen the mechanistic link to cortical thinning.

Adult phenotyping:

In matters of cortical thinning, it should also be clearly stated which Bregma and cortical area was analyzed. In Fig. 2F, S1 is stated. Is the effect only seen in S1? Analyzing whether the reduction affects all layers or distinct layers, could provide valuable information. In the magnifications provided in 2F, it seems that II-IV are similar, and that the reduction is mainly seen in layer I and layer VI. Here, a more detailed analysis would be favorable (and could help to correlate the effects with potential embryonic defects, such as changes in neurogenesis during e.g. the formation of layer VI).

To discriminate between the different layers, layer-specific markers would be needed, which is also required for the cell density measurements depicted in Extended Figure 1. Here, PV and SST+ cIN density was claimed to be analyzed in the superficial layers (I-III) and deep layers IV-VI, based on DAPI staining. However, using DAPI staining to discriminate layer IV from layer III is very imprecise...see also Figure 3A. Also I would not necessarily agree with the demarcation of the deep layer VI shown in 3A. Layer specific labeling needs to be included, such as RORB to stain layer IV or marker for VIb. For the analysis performed in 2B it needs to be mentioned in the legend or the graph axis, whether PV+ or PV/tdTom+ cells were analyzed? As not every PV cells is also positive for tdTom. Same for SST, and the analysis in Fig. 3D (Hippocampus). In the hippocampus, the authors observed also PV+ cell reduction and a more severe reduction of nNOS+ interneurons. Mutant mice showed increased seizure susceptibility, for which PTZ injections and patch clamp-recordings in slices were performed to characterize electrophysiological properties. The authors revealed changes at the population level (greater variability) for three interneuron subtypes in H3.3K4M Hom mice, which they interpret as impaired maturation. Moreover, altered gamma oscillations were found.

In Figure 4 (Differences in Seizure-Related Mortality at 3 vs. 5 Months) the authors report different seizure susceptibilities and mortality rates at 3 vs. 5 months (e.g., female mortality is ~50% at 5 months, but at 3 months, males show higher seizure mortality). The text suggests that the higher PTZ dosage for males at 3 months may underlie this effect, yet it remains somewhat confusing why the absolute differences (between males and females, at each time point) shift so dramatically. It should be discussed whether the PTZ dose alone explains male-female differences or if age-related changes in brain maturity or H3K4 methylation might also be factors.

For PTZ experiments, did the authors used Racine scoring?

Developmental vs. Adult-Stage Functional Changes It would be helpful to discuss to what extent the observed electrophysiological and behavioral phenotypes arise from early developmental disruption of H3K4 methylation versus

potential ongoing dysfunction in adult Nkx2.1-lineage cells. Since the H3.3K4M mutation is constitutively expressed, both mechanisms are possible. Although inducible Cre lines or temporally restricted models may be beyond the scope of this study, a brief discussion of this limitation, and how future studies might approach developmental versus adult-stage contributions, would strengthen the interpretation of the results.

Sex Differences in Phenotypes

Certain figures (e.g., Figures 4 and 5) present male and female data separately (e.g., PTZ-induced seizures and related behavioral outcomes), revealing sex-specific phenotypes. However, earlier figures (Figures 1–3), which assess H3K4 methylation levels, cell counts, cortical thickness, survival and body weight, do not consistently stratify the data by sex. This inconsistency limits the interpretation of sex-specific findings. The authors should clarify whether male vs. female comparisons were performed for these earlier datasets and either present the data accordingly or explicitly state if no sex differences were observed. As the behavioral phenotyping reveals intriguing sex-dependent effects, aligning these outcomes with corresponding cellular phenotypes would strengthen the study and provide a more integrated understanding of the observed differences.

Although the manuscript highlights sex-specific behavioral and transcriptional effects in H3.3K4M Hom mice, the mechanistic link to H3K4 methylation remains somewhat speculative. The Discussion could be strengthened by elaborating on known epigenetic or neuroendocrine sex differences that may modulate H3K4 methylation patterns, such as hormonal control of chromatin modifiers or differences in X-inactivation escape genes. Furthermore, it would be helpful to clarify whether the authors observed comparable reductions in H3K4me3 or chromatin accessibility in male and female samples, or whether sex-specific effects may reflect differential sensitivity to epigenetic dysregulation.

Weight Gain: Hypothalamic vs. cIN Contributions While the manuscript focuses on hypothalamic disruptions leading to overeating/obesity, it is worth considering the possibility that altered cIN functionality could contribute to weight changes indirectly through mood or motivational deficits (for instance, depression-like or other behavioral changes can also predispose to weight gain). The causal link between hypothalamic gene expression changes and increases in body weight could be more strengthened or discussed.

Single nuclear RNA seq experiments

In subsequent single-cell analyses (Figure 7, Extended Data 6), the manuscript transitions from using an Ai9/tdTomato reporter to a Sun1-GFP reporter system, and thus to another mouse line. Please justify why a different fluorescent reporter (Sun1-GFP) was used for the single-nucleus analysis. Clarify that the experiment is directly comparable to the earlier Ai9 lines.

What about Sex-Specific Differences in the snRNA-Seq/ATAC-Seq analysis? The documentation of how sex was incorporated into the overall experimental design and analysis pipeline is inconsistent also for the seq-data. Specifically, the figure legends, methods and main results section do not clearly state which datasets (e.g., snRNA-seq, snATAC-seq, WNN-integrated data, pseudotime, and WGCNA analyses) were generated from male, female, or mixed-sex animals. It is also not evident whether sex was treated as a covariate during data integration and normalization, or whether sex-specific analyses were systematically performed across all comparisons. Given the increasing recognition of sex as a critical biological variable and its relevance to the phenotypes described (e.g., seizure susceptibility), I recommend that the authors provide a clearer account of the sex composition of each dataset and clarify in the figure legends and Methods whether data were pooled, stratified, or sex-balanced. This information is essential for interpreting the generalizability of the findings and for evaluating the extent to which observed effects are sex-dependent. Same accounts for Figure 8. Information on sex should also be mentioned in the results e.g. when describing altered expression in Mef2c.

In the provided sex-specific comparisons, where differences in gene expression (e.g., potassium channel genes in PV+ cells) are reported, refer to the H3.3K4M HOM samples. However, it remains unclear whether sex-specific differences were also present in the adult cortical control (WT) samples, or whether these only emerged in the mutant condition. This distinction is important for interpreting whether the observed differences reflect baseline sexual dimorphism or genotype–sex interactions. For PV+ cells sex specific divergent regulation was shown (Extended Data 6H). However, for SST cells, the analysis is not shown sex-specifically. Are there now sex-specific effects? Illustrating these differences more systematically and clearly could significantly strengthen the discussion about sexual dimorphism.

In Figure 7C and Extended Data 6A, more detail on cluster annotation (including specific marker genes) would help the reader interpret the results.

Does the color code presented in Figure 8C refers to the UMAP in 8A (right)? Should be mentioned in the legend.

Extended Fig. 5G: why are the accessible peaks different/complementary in MGE versus Hypothalamus (Numerous decreased peaks in HOM-MGE but increased in hypothalamic samples)?

Hypothalamic findings:

The hypothalamic analysis reveals substantial changes in cellular composition and organization in H3.3K4M Hom mice, including increased NPCs, loss of ARC/PVN populations, and expansion of astrocytes. The disorganization of tanycytes and myelin in the median eminence is striking and well illustrated histologically. However, transcriptomic analysis of tanycytes was limited by low cell numbers, with few DEGs detected. The interpretation of specific transcriptional changes, such as the downregulation of Fgf14, would benefit from orthogonal validation (e.g., in situ hybridization or qPCR) to confirm this finding independently of the single-nucleus dataset. Strengthening the support for these changes would enhance the mechanistic link between H3K4me dysregulation and hypothalamic function.

The reduction in Olig2+ cells is an interesting finding and points to fate regulation by H3K4me3. The authors report a

„corresponding decrease in myelin basic protein (MBP) in this region (Extended Data Fig. 2).“ However, I could not find a quantification here. This would need to be provided for that statement.

While the expansion of hypothalamic astrocytes in H3.3K4M Hom mice is clearly documented, the manuscript would benefit from additional developmental context clarifying how Nkx2.1-expressing progenitors contribute to glial lineages, particularly in the hypothalamus. Since Nkx2.1-lineage cells are widely known for giving rise to cortical interneurons, it would be helpful to note that these progenitors can also generate astrocytes and tanycytes in ventral forebrain regions, as supported by lineage tracing studies, e.g. in the introduction. This background would help interpret whether the observed glial expansion reflects an expected regional fate or a fate shift resulting from altered H3K4 methylation.

The Discussion clearly integrates behavioral, transcriptomic, and cellular data to support a model of H3K4 methylation-dependent regulation of interneuron and hypothalamic development. However, the authors should consider including additional developmental context for the gliogenic potential of Nkx2.1+ progenitors, particularly in the hypothalamus, to better support the interpretation of astrocyte expansion.

Overall, this manuscript presents an ambitious and multidimensional study of H3K4 methylation in forebrain development and function. The integration of single-cell epigenomics, electrophysiology, and behavior is commendable, and the focus on both MGE-derived interneurons and hypothalamic circuits is very comprehensive. Addressing the points raised here — particularly the developmental issues, glial fate interpretation, sex-stratified analyses, and figure clarity — will help to enhance the robustness and interpretability of the findings.

Version 1:

Reviewer comments:

Reviewer #1

(Remarks to the Author)

The authors have answered all my questions.

Reviewer #2

(Remarks to the Author)

The authors have addressed all the concerns I raised in the previous version of the manuscript, providing explanations for those that could not be resolved experimentally. I believe the expanded characterisation of the mutant mice, including the new western blots only included in the rebuttal letter, should be incorporated into a revised version of the manuscript, as it will make the paper clearer without requiring reference to the original paper describing the generation of the mouse mutants. I have no further comments for the authors.

Reviewer #3

(Remarks to the Author)

The authors have made a substantial effort to address the comments of all reviewers, and they conducted additional analyses (e.g., PH3, Ki67, and migration assays), as well as corrections / additions of text throughout the manuscript. All of this indeed strengthen the manuscripts' conclusions. However, some issues remain that, in my view, should be clarified or further improved before acceptance.

1. Figure quality and interpretation of the VZ signal:

In Figure 1c, the ventricular zone (VZ) still lacks a clearly discernible cellular signal (it has not been replaced), which I interpreted as background rather than specific staining. The authors state that this is a long-standing model and provided a staining form the original publication.

The new Figure 5a (very left panel) actually seems to show a more convincing VZ signal, but the resolution and magnification are too low to allow a definitive assessment. I still would recommend to replace Figure 1c with a higher-quality image that clearly shows tdTom cells in the VZ.

2. in matters of cortical layer markers: relying on DAPI alone is not sufficient for defining laminar boundaries, albeit I do agree that layer 4 is nicely visible. However, the border of layer 6 is not clearly visible without a marker, and this might impair the density measurements. Please also compare 3A DAPI with 2F DAPI. In 2F the cortex seems measured to more „ventral“ levels. So there is inconsistency in that.

3. I do agree with the authors, that interneuron–progenitor communication is not the primary focus of the study, which already is very very comprehensive. Yet, I felt that the observed adult phenotypes could be better contextualized by discussing

potential developmental correlates. Immunolabeling for TBR1 (deep-layer neurons at E13.5) or progenitor markers such as PAX6 and EOMES (TBR2) at embryonic stages could have helped to link developmental alterations (reduced invasion of cINs in to the cortex at embryonic stages) to changes in cortical progenitor populations and to the later layer shifts. I wonder, whether the newly included KI67 staining and PH3 stainings, analyzed in the MGE, could also be analyzed in the cortex? Less numbers of cycling progenitors might already provide some insights. Even if such experiments are beyond the current scope, an explicit acknowledgment of this limitation and of possible mechanistic links would strengthen the discussion. In that matters, the authors wrote in line 528: "The primary cause for reduced forebrain interneurons was decreased cortical migration, leading to smaller brains and reduced cortical thickness."

This causality is not proven. As cINs make up rather a minority, I doubt that their reduced density makes up the reduced thickness. The possible effect on cortical progenitors and excitatory neurons needs to be at least discussed.

Speaking of the cortical thickness reduction: it is a strong reduction in Figure 2F, whereas this reduction is absolutely not visible in Figure 3 A, although it is both S1 of 3 weeks old mice. As there is only one scale bar shown for the WT in 3A, I assume it is the same scaling. The cortex thickness in WT and HOM is nearly the same... How do the authors explain that? Layer 1 in 2F WT seem very large. Are the regions chosen for measuring well controlled and matched between the genotypes?

Version 2:

Reviewer comments:

Reviewer #3

(Remarks to the Author)

The revised manuscript has improved in presentation and wording and is suitable for publication. For completeness, I would like to note two points that, in my view, are not fully addressed and should be kept in mind when interpreting the data.

First, the pronounced thinning of upper cortical layers is unlikely to be explained by the reported changes in interneuron abundance, which appear rather modest. Given that later embryonic stages of corticogenesis were not examined, alterations in excitatory neuron development remain a plausible alternative explanation. Excluding this possibility would have required analyses at later developmental time points, when upper cortical layers are generated.

Second, if the reported differences in cortical thickness are robust and systematic, they would be expected to be detectable across sections. The discrepancy between the quantitative measurements shown in Figure 2F and the qualitative appearance of cortical sections in Figure 3 therefore remains difficult to reconcile. The authors' response clarifies their interpretation but does not fully resolve this issue. I note this point for completeness.

These points do not detract from the overall quality or significance of the study.

made.

Response To Reviewers

We thank the reviewers for their overall positive comments and feedback on our manuscript. We found many of their suggestions to be helpful in strengthening the data presentation and conclusions of our study. Our point-by-point response to the each of the reviewers' suggestions is below, with reviewers' comments indented & italicized, and our response in blue font. All significant changes in the Revised Manuscript are in red text.

Reviewer #1:

Li and colleagues present a very thorough and well written study on the role of histone 3.3 lysine 4 methylation depletion upon neurological development and functional changes due to targeted gene manipulation in either the MGE or hypothalamic regions. The authors employed a strategic approach to mimic humans with loss of function mutations in this gene, by expressing a version of histone 3.3 that was incapable of having lysine 4 methylated and in turn competed with endogenous histone 3.3 like humans with mutations in this gene. The authors also studied dosage of this effect by assessing both het and homozygous expression of this interfering mutant. The extent of approaches to characterize these mice and the relevance to the human condition are paramount and I think the authors did a superb job but just have some suggestions to help better establish some mechanisms underlying their findings and other minor things that I hope will improve the manuscript below:

We thank the reviewer for these positive comments.

1. For Figure 1 text in the results section, I would like to see more detail in the text describing aspects of experiments detailed in panels D and E to help readers.

Thank you for this suggestion, we have added additional details in the Results to better explain the logic for Figure 1D-E.

2. Just to be safe, in line 142 where it first comes up, can you change "MGE and hypothalamus" to "MGE and/or hypothalamus." Also make this edit later as this comes up a few times.

We changed 'and' to 'and/or' at line 142 and at several other locations where this edit was warranted.

3. In Figure 2D the Y-axis values seem off, can you check and make sure the values reflect the weight in grams of the mice or change?

The graph in Figure 2D displays brain weight of WT, Het and Hom mice at 3 weeks. The Y-axis is correct, with the brains weighing between 0.3 and 0.4 grams. To clarify this point, we have relabeled the Y-axis to read 'Brain weight (g)'.

4. In Figure 2F, it is interesting that Nkx2.1-Cre manipulation of cortical interneurons would result in microcephaly, which is normally due to changes in pyramidal neuron morphology. Not asking for additional experiments here but this is an interesting and potentially novel phenomenon that should be mentioned/speculated in the discussion.

The reviewer raises an interesting point about brain size. However, we do not claim that the H3.3K4M mice are microcephalic in the manuscript. Microcephaly is diagnosed when a baby/infant's head circumference is in the third percentile or lower. While we observe a statistically significant decrease in brain weight between WT and H3.3K4M Hom mice, we do not know if this would fall into this medical definition of microcephaly (and we don't know of a good way to define the 3rd percentile of mouse brain weight). While many mutations resulting in smaller brains do directly affect pyramidal cell development, there are examples where mutations restricted to GABAergic progenitors results in smaller brains, such *Arx* & *Nkx2.1* (reviewed in PMID: 38094004). Thus, we don't believe that smaller brains in our mutant mice are a novel phenomenon whereby mutations in GABAergic progenitors result in smaller brains.

5. Starting in Figure 5, some panels are hard to see, including the Tom and PV images. Can the authors make these brighter in the figures?

Thank you for this suggestion, we have increased the intensity of the Tom and PV channels and improved the signal-to-background ratio on immunohistochemistry images in several figures.

6. In figure 4B, bottom table, the authors mention they used a higher dose of PTZ to induce seizures in males. Can the authors color-code or do something else to reflect this in the table? This would help readers with the presentation of the data.

We have changed the 3-month male column heading to read 'Male (40)' to highlight this distinction, and we added the following sentence in Figure 4B legend: "3-month-old males received a 40 mg/kg PTZ dose (40) whereas all other conditions received a 20 mg/kg dose."

7. In Figure 4E, this is a minor thing but could the authors correlate the language using epileptiform bursts mentioned in the text to the interictal events presented in Figure 4F? Just wanted the language to match so readers had an easier time understanding.

Thank you for pointing this out, we now refer to these as interictal events in the Results & Figure legend.

8. The loss of cortical interneurons is a major phenotype in these mice but how this occurs is not known. The authors have done a great job with their RNA-seq suggesting that there are less mutant MGE cells later in the paper but this is not yet tested. Can the authors look at embryonic WT and mutants to compare proliferation and apoptosis to better understand how this loss of cells is propagated? Maybe something as simple as either PH3 or SOX2 staining (proliferation) and cleaved caspase-3 (apoptosis) staining in the MGE, and potentially hypothalamus, to answer?

We thank the reviewer for this suggestion. To gain additional mechanistic insight, we have assessed cell proliferation (via PH3 & Ki67 staining), apoptotic cell death (via TUNEL staining) and interneuron migration in WT, Het & Hom mice. These new data are summarized below:

1. Cell cycle dynamics: We quantified the density of PH3+ cells in the MGE and did not identify any differences in PH3+ cells between any genotypes. We also did not observe any striking differences in Ki67 expression between genotypes, though not quantified. These findings demonstrate that there are no changes in cell proliferation in H3.3K4M mutation mice. This result is not surprising since histone variant H3.3 is weakly expressed in cycling VZ/SVZ cells and gets strongly upregulated in postmitotic neurons (Funk...Kwan *PNAS* 2022, see Reviewer #3, point #3 below).

2. Apoptosis: We performed TUNEL staining to detect apoptotic cells in the E13.5 MGE, E13.5 hypothalamus and P4 cortex. We did not observe any obvious difference of TUNEL positive cells either at E13.5 MGE or P4 cortex. Since there were very few TUNEL+ cells in all genotypes, we did not perform quantitative analysis on this tissue.
3. Cell migration: We did observe clear differences in the number of tdTom+ cells migrating into the cortex, with significantly fewer interneurons entering the cortex in E13.5 H3.3K4M Hom mice compared to WT. To determine if this migration was simply delayed in mutant mice or rather led to significant reductions in interneurons, we also examined P4 WT and Hom brains. This is prior to the intrinsic programmed interneuron apoptosis that removes 20-40% of interneurons from ~P6-P12. We observe a striking decrease in Tom+ cells in the forebrain of Hom brains, most notably in the hippocampus. These findings indicate that reduced tangential migration into the cortex is likely the primary cause for reduced cell numbers in these Hom mice.

These data are presented in New Fig 5 & Extended Data Figure 12, described in the Results and integrated into the Discussion. This data provides significant mechanistic insight into how reduced H3.3K4 methylation reduces interneuron numbers in the forebrain, and we again thank the reviewer for this suggestion.

9. In lines 391-392 of the results section, it is mentioned that the hypothalamus has a reduction in GFP+ cells. Are there any volumetric/regional assessments that could be done to assess how this impacts the hypothalamus and various nuclei? It would be very interesting to see if any changes occur in distinct regions of the hypothalamus already implicated herein.

The reviewer raises an interesting question about assessing cellular/volumetric changes in specific hypothalamic nuclei. The decrease in percent of GFP+ nuclei in the Hom mice mentioned here is based on our flow cytometry analysis of E13.5 hypothalamus. While we can assign many of these cells to putative mature hypothalamic nuclei based on transcriptomes at this age (Figure 9 A-C & Extended Data Figure 12A-D), that does not mean that the structure & organization of these nuclei are representative of the mature brain. Additionally, hypothalamic neurogenesis will continue for several days after this E13.5 timepoint, with postmitotic cells still migrating to their proper hypothalamic nuclei. And different hypothalamic nuclei may be generated across different timescales. Thus while we agree with the reviewer's point that distinct hypothalamic nuclei could be differentially affected in this mutation (and this is supported by Extended Data Figure 12D), it's unclear that different regional and/or volumetric analysis of GFP+ nuclei at E13.5 would be informative to address this question.

We now include representative examples of our flow cytometry plots in Extended Data Figure 7 and have cited this figure in the relevant portions of the manuscript. We have also clarified in the Methods how we calculated the average % GFP+ nuclei in these experiments.

Reviewer #2:

In this manuscript, Li and colleagues described experiments in which they perturbed H3K4 methylation in the developing medial ganglionic eminence (MGE) and the hypothalamus of mice. To this end, they express (at different doses in heterozygous and homozygous mice) a mutated form of H3.3 in which the lysine 4 residue is mutated to a methionine. These mutant mice have persistent body weight and growth problems (initial undergrowth, followed by obesity), fewer

GABAergic interneurons, abnormal network dynamics, and increased seizure susceptibility. Mutant mice also exhibit behavioural abnormalities, which were more prevalent in females than males. The authors also performed single-cell sequencing experiments to identify transcriptional changes that could underlie the cellular phenotypes described in mutant mice. This paper demonstrates a critical role for H3K4 methylation in the development of the MGE and the hypothalamus of mice. Disrupting H3K4 methylation impacts gene expression during embryonic development, causing multiple cellular alterations that, in turn, affect the neuronal populations generated in the brain regions studied. However, while the paper does a very nice job at identifying phenotypes caused by the likely insertion of mutated histones in the chromatin (although this is not directly shown in this manuscript), there is little mechanistic insight into the process that H3K4 methylation regulates to cause the abnormalities. Are the phenotypes caused by abnormal patterning, neurogenesis, migration, survival, or wiring? Are the phenotypes independent or causally linked? Are the transcriptional changes sufficient to explain the cellular abnormalities?

We thank the reviewer for this accurate summary of our manuscript and the positive feedback, and we have addressed the questions raised here more specifically below.

Main issues

1. The authors report that disrupting H3K4 methylation in the MGE and hypothalamus has a massive impact on food consumption and, consequently, growth. With this in mind, it is difficult to rule out that some of the changes observed in the brain are not indirectly caused by abnormal metabolism and growth in these animals, rather than a direct effect on the methylation profile of H3.3 at specific genomic loci. In this context, while the single-cell analyses in embryos may hint at possible mechanisms underlying the phenotypes described by the authors, the data obtained from the adult mice are hard to interpret. Since the hypothalamic deficits are likely at the core of the metabolic alterations, one possible way to circumvent this problem is to express H3.3K4M only in the MGE and study what features this manipulation recapitulates from those observed in Nkx2.1-Cre mice.

One strength of our study is that we have modified H3.3K4 methylation in 2 brain regions that are associated with the most common phenotypes of perturbed H3K4 methylation in human disease: epilepsy, ID and abnormal body growth. However, the reviewer is correct that it is challenging to definitively link specific phenotypes to either interneuron or hypothalamic dysfunction, or possibly that changes in both cell populations could underlie some phenotypes. And we cannot definitively rule out indirect effects of the H3.3K4M mutation. To highlight this issue, a recent study demonstrated that disrupting activity of SST+ interneurons in the gustatory cortex can alter taste sensitivity and sucrose consumption (Yevo...Maffei, *Current Biology* 2025). Thus, it's possible that altered interneuron function in the gustatory cortex could, in part, play a role in appetite drive and/or obesity. We have expanded on these issues in the Discussion.

As the reviewer suggested, ideally we could express the H3.3K4M mutation only in MGE-derived cells (or hypothalamic cells) to circumvent this problem. There are 2 Cre-driver mouse lines that are widely used in the field to target the MGE: *Nkx2.1-Cre* (used here) and *Lhx6-Cre* (as *Lhx6* is expressed in postmitotic MGE cells but not LGE or CGE cells). However, *Lhx6* is also expressed in a subpopulation of hypothalamic nuclei during embryogenesis (Kim...Blackshaw *Comm Bio* 2021), so even this Cre-driver is not clean to target MGE cells.

We tried generating *Lhx6-Cre;H3.3K4M* mice but never obtained any homozygous offspring. (We had similar results in a previous study where we could not generate *Lhx6-Cre;Ezh2* mutant mice, Rhodes...Petros *Frontiers* 2024). *Lhx6* is expressed in the mesenchymal lineage and is critical for palate development (PMID: 26071365), so it's possible that perturbation of H3K4 methylation in *Lhx6*-lineage mesenchymal cells leads to aberrant palate development and lethality.

While most *Lhx6-Cre;H3.3K4M* Het mice also died during embryogenesis and early postnatal period, a few of them survived to one month. These *Lhx6-Cre;H3.3K4M* Het mice appeared underweight compared to WT littermates and their brains weighed significantly less. We detected H3K4M protein via HA immunostaining in the postmitotic cells of MGE as expected. There was a trend towards decreased PV+ and SST+ interneurons in the hippocampus, like the trends observed in *Nkx2.1-Cre;H3.3K4M* Het mice. These data are presented in New Extended Data Figure 4 and in the Results. While this does not eliminate possible contribution from H3.3K4 methylation defects in the hypothalamus (since *Lhx6* is also expressed in hypothalamus ID&TT nuclei), it does confirm some of the basic brain weight & interneuron changes in a second driver line targeting the MGE.

Other driver lines used to target MGE-derived interneurons are *SST-Cre* and *PV-Cre*, which label separate populations of MGE-derived interneurons. But PV is not expressed until the second postnatal week. SST begins to be expressed in postmitotic MGE cells, but SST expression (and thus Cre) slowly ramps up in all SST+ cells over several days. As histone variant H3.3 begins to replace H3.1 and H3.2 shortly after becoming postmitotic (see Reviewer #3, Point #3 below), we do not believe either of these Cre-drivers is ideal to introduce the H3.3K4M mutation in MGE precursors. Additionally, neither *SST* nor *PV* are restricted to MGE-derived interneurons (*PV* is highly expressed by Purkinje cells for example), so altered H3.3K4 methylation in other cell types in these lines could still be confounding. Thus, while we agree with the reviewer's suggestion, we do not know of a 'clean' Cre-driver to specifically label only MGE cells. One could use viral injections or in utero electroporation to genetically target the MGE, but these are technically very challenging experiments, would be unilateral, and would only target a subset of MGE cells, thus complicating data interpretation with these approaches.

2. The authors describe differences in seizure susceptibility between male and female mice. However, sex differences were not explored for GABAergic interneurons in the cortex, which the authors suggest are at the core of these problems. Are there differences in the density and relative proportion of interneurons in male and female mice? Are the analyses performed in Figure 3 powered to detect differences in SST interneurons? There is a clear trend in Figure 3B.

We thank the reviewer for this suggestion. In Figure 3 and Extended Data Figure 2, we now indicate the sex of the mice used for cell counts, with open circles = female and closed circles = male. We increased the n to obtain at least 3 male & 3 female mice for WT and Hom cortex and hippocampus counts (with 2 males & 2 females for Het mice). We do not observe any significant differences in either the density or percentage of Tomato, PV+ or SST+ cells between male & female mice for each genotype. Because splitting the animals by sex increased the variance and reduces the power of the statistics, we still report the statistics from combining the sexes in the manuscript. However, we have performed unpaired t-tests between males & female WT and Hom mice for each cell type & brain region, and the results are very consistent between sexes: i.e., for nearly every datapoint, (1) there is a statistically significant difference between male WT vs. Hom AND female WT vs. Hom, or (2) there is NOT a statistically significant difference between male WT vs Hom and female WT vs. Hom.

Additionally, increasing the n for these cell counts now resulted in a significant decrease in the density of cortical SST+ cells in the H3.3K4M Hom mouse, rather than just a clear trend. We have updated the text in the Results, Methods and relevant figure legends to reflect these changes.

3. The characterization of hypothalamic deficits is primarily based on single-cell analyses. It would be appropriate to identify key cellular populations and examine cell densities, proportions, and other relevant characteristics using histology, which would help build a picture of the deficits more similar to what is described for cells derived from the MGE. This would allow the authors to explore common themes emerging from the analysis of the two brain regions.

MGE-derived interneurons can be cleanly classified and labeled into two large subgroups (PV+ and SST+), and an additional nNos+ population in the hippocampus. This makes analysis relatively straightforward to assess changes in the absolute number or percentage of these cell types. In the hypothalamus, Nkx2.1+ progenitors give rise to glutamatergic & GABAergic neurons cells as well as non-neuron cells (astrocytes, oligodendrocytes and tanycytes) that populate >12 different hypothalamic nuclei (Extended Data Fig 12C and related references). While it would be ideal to characterize all these cell types, it is beyond the scope of this manuscript to comprehensively examine cell densities & other characteristics of cells in so many distinct hypothalamic nuclei.

We note that our immunostaining results in old Figure 8/New Figure 10 reveal striking cellular organization changes in the hypothalamus of H3.3K4M Hom mice, with complete disorganization of tanycytes and astrocytes in the median eminence and lateral wall of the third ventricle. Additionally, we quantify increased hypothalamic astrocytes in Hom mice (Extended Data Figure 14) and altered oligodendrocyte organization and SST expression in the median eminence (Figure 10). We also observed an increase signal of Ki67 in the E13.5 hypothalamus of Hom mice (Extended Data Figure 12G). Some of these findings were predicted from our single cell sequencing data. We have now performed TUNEL staining to detect apoptotic cells at E13.5 and P4 in the hypothalamus, and like the MGE we do not detect any differences between WT and Hom mice. Thus, we believe that our data highlights important changes in cell populations and organization of the hypothalamus beyond our single cell analyses, which likely are (in part) causative of some phenotypes in the H3.3K4M Hom mice.

4. As it stands, it is unclear why studying the MGE and the hypothalamus simultaneously adds to our understanding of the role of H3K4 methylation in brain development and function.

The reviewer questions why we explored H3K4 methylation in both the MGE and hypothalamus simultaneously. The motivation behind this strategy comes from the observation that two of the most common phenotypes found in NDDs related to histone modifications are intellectual disability with epilepsy (interneuron related) and abnormal body growth (hypothalamus related) (see reviews PMID 31595951 and 28475857 highlighting this point). Thus, we were curious if targeting the H3.3K4M mutation to these two brain regions was sufficient to induce phenotypes consistent with these human diseases. And this was largely successful, as these H3.3K4M Hom mice display spontaneous seizures and/or increased seizure susceptibility, altered performance in numerous behavior assays, and abnormal body growth (initially underweight but eventually catch up and become obese). Additionally, as noted above in our response to point #1, we do not know of a genetic strategy to specifically target the H3.3K4M mutation to MGE interneurons because the *Nkx2.1-Cre* line also targets the hypothalamus (which we view as benefit for this study). And we have now included new data from *Lhx6-*

Cre;H3.3K4M Het mice (New Extended Data Figure 4), but again *Lhx6* is also expressed in a subset of hypothalamic nuclei. While our original submission noted the co-occurrence of epilepsy and abnormal body growth in NDDs associated with histone defects, we have re-emphasized this relationship at several points in the revised manuscript.

Other issues

5. *LSL-K4M* mice are not described in any detail. What locus is it used to drive the expression of *H3.3K4M*?

The *LSL-K4M* mice were generated in Dr. Kei Ge's lab and described in their original paper (Jang...Ge *Nuc Acid Res* 2019). The transgene construct is CAG promoter-LoxP-Stop-LoxP-*H3.3K4M*-FLAG/HA-polyA, using the human *H3.3b* gene sequence (<https://www.addgene.org/128743>). We confirmed with Dr. Ge that they never determined where the exogenous *H3.3K4M* cassette was integrated. A more detailed description has been added to the Methods section.

6. Does the exogenous expression of *H3.3K4M* influence the endogenous expression of *H3.3K4*? One would infer that the reduced methylation is caused by the availability of mutated *H3.3* but, is it possible that levels of wild type *H3.3* are changed in these mice? This could be explored through Western blots.

The reviewer raises a valid concern about expression of endogenous *H3.3* levels being altered in the presence of the exogenous *H3.3K4M* variant. The Western blots in Figure 1E show protein levels for both *H3K4me3* levels and total histone *H3* protein (bottom row) in the MGE and hypothalamus. For our Western Blots in Figure 1E, we extracted histone proteins using the EpiQuik Total Histone Extraction Kit (EpigenTek), as this was the only way we could obtain sufficient levels of histones to detect specific histone modifications in Western blots. We tried several different tissue preparation conditions (Repo buffer, nuclei harvesting, etc.), but we were unable to obtain strong *H3K4me3* signal in these preps for quantification. Thus we don't have a way to normalize for total histone levels in this prep, since we are only extracting histone proteins.

To determine total *H3.3* histone levels, we extracted total protein from E13.5 MGE and hypothalamus and normalized protein levels to tubulin. We detected no significant differences in *H3.3* histone levels in the MGE and hypothalamus between *H3.3K4M* WT, Het and Hom mice (see image below). This agrees with the original *LSL-K4M* manuscript (Jang...Ge *Nuc Acid Res* 2019), which performed Western blots on total *H3* protein and did not identify any changes in total *H3* histone protein levels between *Cre+* and *Cre*-negative conditions (see point #7 below). We have not included this data in the manuscript, but we can if the reviewers feel strongly about this data point.

7. The authors should also carry out Western blots for me1 and me2 to quantify how the expression of H3.3K4M impact those levels in heterozygous and homozygous mice (as shown for me3 in Figure 1E).

We performed immunostaining of H3K4me1, me2 and me3 in the MGE, with a clear downregulation of each in the MGE (Fig 1D), but we only performed Western blots on H3K4me3 to quantify this reduction. The original manuscript (Jang...*Ge Nuc Acid Res* 2019) performed H3K4me1, me2 and me3 Western blots (Figure 3A from this paper, relevant panel below). They observe a similar reduction of H3K4me1, me2 and me3 in cells from their LSL-K4M mice in their Western blots (albeit using a different Cre-driver line to target adipocytes & with no quantification). There are numerous publications with Western blots from H3K4M mutant mice or cell lines showing equivalent reduction in H3K4me1, me2 and me3 levels (image from recent *Cell* paper below), usually without quantification. This result is expected since the mutant H3.3K4M histone variant has no lysine to be methylated and thus should have an equal reduction in me1, me2 and me3 levels. Relevant to the point above, neither study observes any obvious change in H3 levels between control and H3K4M+ conditions. Therefore, we believe that collecting several litters of embryos to gather enough tissue from WT, Het and Hom mice for multiple replicates of me1 & me2 Western Blots is not necessary because (1) our immunostaining results reveal a similar decrease in me1, me2 and me3 levels in the MGE, (2) the requested Western blots were already performed in the original manuscript that generated these LSL-K4M mice (and other manuscripts as well), and (3) the loss of the H3K4 residue means that there should be no difference between me1, me2 and me3.

LSL-K4M; Cre-ER brown preadipocytes

Figure 3A from Jang...Ge
Nuc Acid Res, 2019

Figure 1C from Yagi...Hochedlinger
Cell, 2025

8. The age of the embryos analyzed in Figure 1 is not included in the text or figure legends.

We thank the reviewer for noting this omission, we have included the age E13.5 in the main text and the Figure 1 legend.

9. There are several statements at the end of some sections of the Results that belong to the Discussion. Some examples: (1) "Such increased variability in electrical properties is reminiscent of the developing hippocampus, suggesting interneuron maturation deficits in H3.3K4M Hom mice". This is speculative and not a direct conclusion of the experiments; there are several alternative explanations. (2) "As widespread perisomatic inhibition is critical for both circuit gamma entrainment and seizure control, it is likely that the deficits in hippocampal FS/PV+ cells in H3.3K4M mice directly contribute to disrupted rhythmogenesis and epilepsy". Again, this is speculative and not a direct conclusion of the experiments. The authors described defects in other interneuron populations that shape gamma rhythms. Indicating that PV interneurons cause the network dynamic changes and seizure susceptibility is a stretch, especially in the Results section. Please revise the Results to avoid these and similar statements.

We removed these 2 sentences from the Results and integrated them into the Discussion section.

Reviewer #3:

Summary of the Study: This manuscript investigates how reduced H3K4 methylation in the medial ganglionic eminence (MGE) and hypothalamus affects cortical interneuron (cIN) generation and function, as well as hypothalamic circuits controlling body growth. By using a conditional H3.3K4M mouse model driven by Nkx2.1-Cre, the authors describe reductions in interneuron numbers, altered network excitability and seizures, as well as notable changes in postnatal growth and obesity. The data include morphological, behavioral, electrophysiological, and single-cell analyses that point toward significant, cell-type-specific deficits in both MGE-derived interneurons and hypothalamic populations.

Major Points

1. *Validation of the mouse model: The authors would need to verify or test the actual timing of Cre expression. The reduction of H3K4me3 in MGE-derived cINs is not properly shown. First, I wonder why the tdTomato signal is so blurry. In numerous manuscripts, including Taniguchi et al., 2011 (DOI: 10.1016/j.neuron.2011.07.026), where Nkx2.1-Cre mediated reporter expression is shown in the embryonic MGE, discrete reporter positive cells are visible in the VZ and SVZ. However, the tdTom signal in Figure 1C and D is very blurry. This hampers proper visualization of analyzing H3K4me3 levels in the recombined cells. However, at adult stages (Fig. 3A) clear tdTom cells are visible.*

The Nkx2.1-Cre mouse was generated in Stewart Anderson's lab in 2008. Since then, it has been used by dozens of labs in >100 studies as an efficient and reliable driver of Cre expression in Nkx2.1-expressing cells in the MGE. The image below is from the original 2008 Xu...Anderson study showing excellent overlap between Nkx2.1 protein and Cre protein (B-C), as well as Cre protein and a Cre-dependent GFP reporter (G-H). Essentially all MGE cells in Nkx2.1-Cre mice express Cre except for a very small proportion of the dorsal MGE, and we mentioned this caveat in our manuscript. We believe that Figure 1C-D in our manuscript clearly show that the tdTom is present throughout the VZ, SVZ and mantle as expected. To clarify this specific point, we have added 'VZ', 'SVZ' and 'mantle' labels to the Tom panel in Figure 1C. Based on these points, we have no concern about the timing of Cre expression, and we are quite confident that basically all MGE cells in the images are Cre+/TdTom+ (and thus expressing the H3.3K4M mutant histone). The tdTom+ reporter in the Ai9 mouse line is extremely bright and fills the entire cell (not just the nuclei like the Cre immunostained images below), and it can be saturating in regions where all cells are TdTom+ like the MGE. That is likely why the tdTom panels appear blurrier in some panels, but this is indicative that all cells are expressing the bright tdTom fluorophore. As the reviewer notes, the Tom+ cells are more visible in the adult brains in Figure 3 because the Tom+ cells are significantly less dense, and thus individual Tom+ cells are clearly visible.

Modified Figure 1 from Xu...Anderson *JCN*, 2008

We also note that the Taniguchi et al., 2011 *Neuron* paper that the reviewer sites use the Nkx2.1-CreER mouse (which was created in that paper), they do not use the Nkx2.1-Cre mouse in that study. Tamoxifen induction in this Nkx2.1-CreER mouse will label only a subset of MGE cells (maybe 50%?), which is why individual cells are more clearly visible in the MGE of Figure 2 in that manuscript compared to our Nkx2.1-Cre mouse where nearly 100% of MGE cells are labeled. Thus directly comparing cell labeling in our study vs. the Taniguchi study is not valid since different Cre-driver lines were used.

2. *As Nkx2.1 is down regulated in cortical fated interneurons upon becoming postmitotic, while staying expressed in e.g. striatal fated cells, such discrete cellular signals would help to show*

downregulation in cortical fated cells, either by analyzing cells that have reached the cortex or by co-labeling e.g. with Calbindin antibodies.

The reviewer is correct that Nkx2.1 is downregulated postmitotically in cortical interneurons while it remains expressed in striatal interneurons. However, we are using an *Nkx2.1-Cre;Ai9* mouse line in which all cells that express Nkx2.1 at any point in their development will express tdTomato throughout their lifespan, regardless of whether the Nkx2.1 gene/protein are expressed or downregulated. Thus, tdTom+ cells represent Nkx2.1-lineage cells (and more relevant, Cre-lineage cells that underwent H3.3K4M recombination), rather than being a reporter of Nkx2.1 expression. And it is these tdTom+ cells that express(ed) Nkx2.1 and continually express Cre and H3.3K4M that we wish to analyze. We do not understand the reviewer's point about labeling with a calbindin antibody, as calbindin is expressed by many glutamatergic cells in the cortex and hippocampus in addition to interneurons.

3. Also, the authors state in line 124: "Having H3.3K4M active prior to cell cycle exit is critical to substitute endogenous H3.3 with mutant H3.3K4M protein due to the prolonged half-life of histone proteins in the brain..." However, there are no clear tdTom+ cells visible in the ventricular zone (VZ), where the progenitors reside. Also, the reduction of H4me2/3 level in the VZ is not properly visible (rather in the SVZ). I wonder, whether successful Cre-recombination could be shown in the sc-transcriptome analyses? This would also allow proper discrimination of progenitors versus postmitotic cINs (however, here a different mouse line was used, commented later).

As stated in response to point #1 above, the Tom signal (indicative of Cre expression) in Figure 1C-D and the HA immunostaining in Figure 1C is clearly present in the VZ and SVZ. So we do not understand why the reviewer believes that "...there are no clear tdTom+ cells visible in the VZ". As stated above, we have no concerns about Cre-recombination since Tom signal (and HA protein) is visible throughout the MGE VZ, SVZ and mantle as expected.

The reviewer is correct that the strongest decrease of H3K4 methylation levels (via immunostaining) in Figure 1D is in postmitotic cells in the mantle, while this reduction is less obvious and/or not present in the VZ/SVZ. The H3K4me1, me2 and me3 antibodies used in this study do not distinguish between histone 3 variants H3.1, H3.2 and H3.3; they recognize me1, me2 or me3 on all 3 H3 variants. As stated in response to Reviewer #2 point #5 comment above and clarified in the Methods section, the *LSL-K4M* mouse line used a human H3.3b gene sequence for the H3.3K4M mutant allele. Thus H3.1 and H3.2 histone variants should be unaffected in this mouse line. In the Introduction, we described an important study (ref 12, Funk...Kwan *PNAS* 2022) showing that H3.1 and H3.2 variants are enriched in cycling neural progenitors (VZ and SVZ cells), with the H3.3 variant replacing H3.1/2 in postmitotic cells in the mouse brain. See Figure 1B below from this paper showing very low H3.3 expression in the VZ/SVZ of the cortex, with strong upregulation of H3.3 in postmitotic neurons in the cortical plate.

B Analysis of H3 proteins in cortical development

Figure 1B from Funk...Kwan *PNAS* 2022 showing absence of H3.3 in VZ/SVZ and upregulation of H3.3 in postmitotic neurons in cortical plate

Since only variant H3.3 is genetically altered in the H3.3K4M mouse line, we would predict the strongest effect of H3.3K4 methylation to be in postmitotic cells where variant H3.3 is actively replacing H3.1/2 variants. The stronger H3K4 methylation signal in the VZ/SVZ regions in H3.3K4M Hom mice is likely from methylation on histone variants H3.1 and H3.2 that are not genetically altered in these mice. Thus, the strong reduction of H3K4 methylation in postmitotic cells in the mantle with a weaker (or no) reduction in H3K4 methylation in cycling VZ/SVZ cells is consistent with this predicted transition of histone variants. We apologize for not making this point clearer in the text, and we have attempted to clarify this logic in the Results.

4. For the hypothalamus, no immunostainings are shown in that regard, which should be provided as well. Here, I would similarly expect clear cellular tdTomato labeling, which allows determination and quantification of H3K4me3 fluorescence intensity in tdTom cells.

The bottom panels of Figure 1C show Tom expression and HA immunostaining through the WT and H3.3K4M hypothalamus, clearly showing that tdTom is expressed in the VZ, SVZ and postmitotic cells as in the MGE.

Regarding the reviewer's comment about "...quantification of H3K4me3 fluorescence intensity in tdTom cells.", we did not perform any quantification of H3K4me3 fluorescence intensity in this manuscript, as this approach is much less reliable than quantifying protein levels in Western Blots. We could generate a region of interest (ROI) in the MGE and hypothalamus and then quantify the H3K4 methylation levels within this ROI. But we argue that this is far inferior to quantification using Western Blots because: (1) each immunostaining reaction has inherent variability in terms of signal intensity, microscopy conditions, etc., (2) it's unclear how we could normalize these issues between samples, and (3) matching MGE and hypothalamic sections along the anterior-posterior axes of the brain can be very challenging, and different cellular densities along the axes of both tissues can lead to inaccurate spatial comparisons between samples. Quantifying protein levels in Western Blots avoids many of these complications and is a more standard and accepted quantitative approach in the field.

5. In addition, the authors provided Western Blot data, which showed a strong reduction for the hypothalamus in the HOM mice, but less clear reduction for the MGE. I assume from the methods,

that bulk tissue was used here. Maybe FACS-mediated enrichment of the recombined cells would yield in clearer reductions in K4me3 levels.

The reviewer is correct that the Western blots were performed on bulk tissue from dissections of E13.5 MGE and hypothalamus. In H3.3K4M Hom mice, the reduction in the hypothalamus (~80% reduction) is stronger than in the MGE (~50% reduction), but both are significantly reduced in comparison to their respective WT samples. It is possible to sort the tissue to collect only Ai9+ cells from the MGE and hypothalamus, as our dissections certainly contain some Cre-negative cells that do not express H3.3K4M protein. But since we observe the expected decrease in H3K4 methylation levels via immunostaining (unquantified) and Western Blots (quantified), we do not see a need to sort these cells.

6. Embryonic fate-mapping and characterization of potential defective processes is missing. What I miss in general in this manuscript is fate mapping of the cells at embryonic stages. By providing embryonic sections with proper tdTomato cell labeling not only allows the determination of H3K4me3 reduction in e.g. cells that have reached the cortex, it would also allow to assess which developmental defects occur that lead to reduced cIN numbers in the adult mice. Is the migration altered (e.g. misrouted)? Or do HOM-cINs die? At which stage (embryonically or postnatally)? Same for the reduced numbers of nNos+ interneurons seen in the hippocampus. More comprehensive embryonic phenotyping would also serve to better understand the findings the authors report in postnatal/adult mice, such as the reduced cortical thickness in postnatal brains of mutant mice (Fig. 2).

The reviewer raises an important question that all developmental studies struggle to comprehensively answer: defining the age and mechanism that an embryonic mutation generates a phenotype in the adult mouse. We have performed several experiments to address these concerns addressed in our response to Reviewer #1 point # 8 above and have restated them below:

1. Cell cycle dynamics: We quantified the density of PH3+ cells in the MGE and did not identify any differences in PH3+ cells between any genotypes. We also did not observe any striking differences in Ki67 expression between genotypes, though not quantified. These findings demonstrate that there are no changes in cell proliferation in H3.3K4M mutation mice. This result is not surprising since histone variant H3.3 is weakly expressed in cycling VZ/SVZ cells and gets strongly upregulated in postmitotic neurons (Funk...Kwan *PNAS* 2022, see Reviewer #3, point #3 below).
2. Apoptosis: We performed TUNEL staining to detect apoptotic cells in the E13.5 MGE, E13.5 hypothalamus and P4 cortex. We did not observe any obvious difference of TUNEL positive cells either at E13.5 MGE or P4 cortex. Since there were very few TUNEL+ cells in all genotypes, we did not perform quantitative analysis on this tissue.
3. Cell migration: We did observe clear differences in the number of tdTom+ cells migrating into the cortex, with significantly fewer interneurons entering the cortex in E13.5 H3.3K4M Hom mice compared to WT. To determine if this migration was simply delayed in mutant mice or rather led to significant reductions in interneurons, we also examined P4 WT and Hom brains. This is prior to the intrinsic programmed interneuron apoptosis that removes 20-40% of interneurons from ~P6-P12. We observe a striking decrease in Tom+ cells in the forebrain of Hom brains, most notably in the hippocampus. These findings indicate that reduced tangential migration into the cortex is likely the primary cause for reduced cell numbers in these Hom mice.

These new data are presented in New Figure 5, described in the Results and integrated into the Discussion.

Regarding nNos+ interneurons in the hippocampus, we have previously shown that nNos protein is not expressed in hippocampal interneurons at P1 (Figure S5A, Quattrocolo...Petros *Cell Reports* 2017). We do not know when nNos gets upregulated in these cells postnatally. But since nNos is not expressed embryonically, we have no way to track these cells from an embryonic fate-mapping perspective.

7. It is known that invading cINs can influence cortical progenitors (DOI: 10.1016/j.cell.2018.01.031), and thus potentially cortical neurogenesis and layer thicknesses. Embryonic fate-mapping could serve to investigate potential migration defects that correlate with changes in cortical progenitor proliferation and lamination. This would strengthen the mechanistic link to cortical thinning.

The reviewer is correct that there is crosstalk between interneuron progenitors and cortical excitatory neuron progenitors that is critical for proper maturation and cortical integration of both cell types. And perturbation of this bidirectional signaling interactions between interneurons and pyramidal cells can disrupt normal cortical maturation. We feel that an extensive characterization of this interneuron-pyramidal cell interaction is beyond the scope of this manuscript, as many of the genes and mechanisms directing this process are still being uncovered. And we are unclear exactly how “embryonic fate-mapping” would lead to a “...mechanistic link to cortical thinning”, especially considering our data where we do not see huge differences in interneuron changes between deep and superficial cortical layers. We have addressed these ideas in the Discussion.

8. Adult phenotyping: In matters of cortical thinning, it should also be clearly stated which Bregma and cortical area was analyzed. In Fig. 2F, S1 is stated. Is the effect only seen in S1?

We thank the reviewer for this suggestion. We have performed additional analysis on V1 and found that V1 thickness is also significantly reduced in H3.3K4M mice, so this effect is not specific to S1. This data has now been included in Extended Data Fig 1C. Bregma coordinates were added to the Methods.

9. Analyzing whether the reduction affects all layers or distinct layers, could provide valuable information. In the magnifications provided in 2F, it seems that II-IV are similar, and that the reduction is mainly seen in layer I and layer VI. Here, a more detailed analysis would be favorable (and could help to correlate the effects with potential embryonic defects, such as changes in neurogenesis during e.g. the formation of layer VI). To discriminate between the different layers, layer-specific markers would be needed, which is also required for the cell density measurements depicted in Extended Figure 1. Here, PV and SST+ cIN density was claimed to be analyzed in the superficial layers (I-III) and deep layers IV-VI, based on DAPI staining. However, using DAPI staining to discriminate layer IV from layer III is very imprecise...see also Figure 3A. Also I would not necessarily agree with the demarcation of the deep layer VI shown in 3A. Layer specific labeling needs to be included, such as RORB to stain layer IV or marker for Vib.

We concede that using DAPI to demarcate general layer boundaries is less precise than performing staining for layer specific cortical layer markers such as Rorb, Ctif2, Satb2, etc. However, we dispute the reviewer’s claim that “...using DAPI staining to discriminate layer IV from layer III is very imprecise”. In S1 where these counts were performed, the strong increase in DAPI+ cell density in layer IV is quite

visible by eye (and is evident in Figure 3A), making it a reliable, albeit imperfect, marker to distinguish layer III from layer IV in S1. The reviewer's concern about using DAPI to define the layer VI/subplate boundary is more valid, but as you can see from Figure 3, there are very few Tom+ cells below the yellow dotted line anyway. So defining this lower layer VI boundary would have a minimal effect on the number of cells included in Layer VI.

Performing the cell counts as we have done requires four channels for Tom (594), PV (647), SST (488) and DAPI (450) (and in the hippocampus, a fifth channel for nNos (750)). This only leaves us the far-red channel (750) for one additional fluorophore. With only 1 layer labeled per brain section, we could only count cells in 1 layer for each immunostained brain slice. This would increase the number of brain sections to be immunostained, imaged, processed and counted by 4-5 fold just to obtain roughly the same data we already have. Thus, the reviewer's suggestion to combine the PV and SST cell counts with specific cortical layer markers would be ideal. But in practice, this would be incredibly burdensome and tedious to repeat the cell counts using this strategy, which would be an incremental increase in detail to what we've already generated here.

In addition to our technical notes above, we want to emphasize that we performed these cell counts in superficial vs. deep layers because, as the reviewer implies, this could have important implications in relation to neurogenesis. As shown in Extended Data Fig. 2, we do not observe any striking differences in shifts of PV+ or SST+ cell numbers between deep and superficial layers in the H3.3K4M Hom mice; the general patterns hold true for both deep and superficial layers. And our new data in Figure 5 does not detect any changes in cell cycle dynamics (via PH3 and Ki67 immunostaining) between WT and H3.3K4M Hom mice. Therefore, we don't believe any further detailed investigation into layer-specific changes is warranted. Our goal here was not to explore differential interactions between interneurons and excitatory cells in particular cortical layers, but rather to quantify general changes in PV+ and SST+ in the H3.3K4M Hom mice.

10. For the analysis performed in 2B it needs to be mentioned in the legend or the graph axis, whether PV+ or PV/TdTom+ cells were analyzed? As not every PV cells is also positive for TdTom. Same for SST, and the analysis in Fig. 3D (Hippocampus).

We apologize for not being clearer on this point in the figure legend. In the methods, we stated "Any PV+ and SST+ cells that were TdTom- were excluded from the counts." This is because PV+ or SST+ cells that are TdTom-negative did not express Cre due to either incomplete recombination or, more likely, because these cells arise from the most dorsal MGE where Cre is not expressed in this Nkx2.1-Cre mouse line. Thus they also would not express the mutant H3.3K4M histone. We have now included this sentence in the legend for Figure 3.

11. In the hippocampus, the authors observed also PV+ cell reduction and a more severe reduction of nNOS⁺ interneurons. Mutant mice showed increased seizure susceptibility, for which PTZ injections and patch clamp-recordings in slices were performed to characterize electrophysiological properties. The authors revealed changes at the population level (greater variability) for three interneuron subtypes in H3.3K4M Hom mice, which they interpret as impaired maturation. Moreover, altered gamma oscillations were found.

We are unclear if the reviewer is asking us for something with this statement, as this seems like just a summary of our findings in this section. We apologize if we are misunderstanding the reviewer's critique or suggestion here.

12. In Figure 4 (Differences in Seizure-Related Mortality at 3 vs. 5 Months) the authors report different seizure susceptibilities and mortality rates at 3 vs. 5 months (e.g., female mortality is ~50% at 5 months, but at 3 months, males show higher seizure mortality). The text suggests that the higher PTZ dosage for males at 3 months may underlie this effect, yet it remains somewhat confusing why the absolute differences (between males and females, at each time point) shift so dramatically. It should be discussed whether the PTZ dose alone explains male–female differences or if age-related changes in brain maturity or H3K4 methylation might also be factors.

The reviewer states “...it remains somewhat confusing why the absolute differences (between males and females, at each time point) shift so dramatically”. Both the mortality rate (50% vs. 60% for Hom mice) and the time course of seizure severity (compare Figure 4C & Extended data Fig 5A) are very similar for 3-month & 5-month-old females. And the time course for seizure activity is also quite similar between male and female mice at 3-months. The most significant differences are (1) seizure strength of 5-month-old male & female Hom mice (where PTZ concentration is the same) and (2) decreased mortality rate of male Hom & Het mice from 3-months to 5-months (where PTZ concentration is different). Based on our statement, “with the increased dose in males generating similar results to female mice”, we understand how the reviewer concluded that we suggested “...the higher PTZ dosage for males at 3 months may underlie this effect.” To clarify these findings and address the reviewer's concerns, we expanded upon these points in the Discussion.

13. For PTZ experiments, did the authors used Racine scoring?

The seizure scoring system from 0-6 is detailed in our Methods section and is taken directly from the cited manuscript (Shimada & Yamagata, *JoVE* 2018). We refer to this as a modified Racine scoring system, and we have updated this terminology in the Methods.

14. Developmental vs. Adult-Stage Functional Changes: It would be helpful to discuss to what extent the observed electrophysiological and behavioral phenotypes arise from early developmental disruption of H3K4 methylation versus potential ongoing dysfunction in adult Nkx2.1-lineage cells. Since the H3.3K4M mutation is constitutively expressed, both mechanisms are possible. Although inducible Cre lines or temporally restricted models may be beyond the scope of this study, a brief discussion of this limitation, and how future studies might approach developmental versus adult-stage contributions, would strengthen the interpretation of the results.

Cells develop along a continuum, with different characteristics maturing at distinct developmental timepoints, many of which are dependent on earlier processes. In a model like this, it is always challenging to definitively assign causation of phenotypic changes to genetic perturbation during neurogenesis, cell maturation, or juvenile/adult stages. And the reviewer is correct that using inducible Cre lines and/or viruses to activate H3K4M expression at later timepoints is beyond the scope of this study (and would be a complete study on its own). While single cell sequencing data does reveal that many genes associated with interneuron maturation (*Nkx2.1*, *Lhx6*, *Dlx1*, *Gad1*), and specifically PV+ cells (*Maf*, *Mef2c*), are significantly downregulated in the MGE from H3.3K4M Hom mice, this does not

directly answer the reviewer's question about the timing of e-phys & behavior phenotypes. We now raise this issue in the Discussion.

15. Sex Differences in Phenotypes: Certain figures (e.g., Figures 4 and 5) present male and female data separately (e.g., PTZ-induced seizures and related behavioral outcomes), revealing sex-specific phenotypes. However, earlier figures (Figures 1–3), which assess H3K4 methylation levels, cell counts, cortical thickness, survival and body weight, do not consistently stratify the data by sex. This inconsistency limits the interpretation of sex-specific findings. The authors should clarify whether male vs. female comparisons were performed for these earlier datasets and either present the data accordingly or explicitly state if no sex differences were observed. As the behavioral phenotyping reveals intriguing sex-dependent effects, aligning these outcomes with corresponding cellular phenotypes would strengthen the study and provide a more integrated understanding of the observed differences.

We thank the reviewer for this suggestion. As detailed in our response to Reviewer #2 point #2 above, we've now included the sex of all animals used in P21 cortical and hippocampal interneuron cell counts, and we do not observe any differences between the density or percentage of interneuron subtypes between male and female mice.

Additionally, we have now separated male & female mice for their weight trajectory from 1-9 weeks (related to Fig. 2B) and for their body weight at 20 weeks (related to Fig. 2I). We see the same effects of decreased body weight at these ages in both male and female H3.3K4M Hom mice compared to WT controls. We had too few Hom mice survive to 12 months to separate by sex; this would be very underpowered. This new analysis is now reported in Extended Data Figure 1A-B.

Some of our data were generated prior to our knowledge of sex differences in the mice, and thus we do not have detailed sex info for all experiments. Especially for embryonic tissue, as that would require PCR genotyping at E13.5. For example, we do not know the sexes of embryonic brains used for E13.5 H3K4me immunostaining or western blots. But a visible reduction in H3K4me levels via immunostaining and H3K4me3 via WB was consistent in all brains examined, with all data points containing a mix of both male and female mice. Additionally, we did not observe a strong bias for either sex to die prematurely, as we would have noticed this over many generations of breeding. And both sexes become obese over time. Many data points in Figure 2 measuring body weight, food intake, fat/lean weight, leptin levels etc., have an $n \geq 7$ mice with combined male & female datapoints (except for 12-month Hom mice since so few of them live that long), which should minimize any sex bias in our datasets. We clarified in the Results and Figure legends when changes were observed in both sexes.

16. Although the manuscript highlights sex-specific behavioral and transcriptional effects in H3.3K4M Hom mice, the mechanistic link to H3K4 methylation remains somewhat speculative. The Discussion could be strengthened by elaborating on known epigenetic or neuroendocrine sex differences that may modulate H3K4 methylation patterns, such as hormonal control of chromatin modifiers or differences in X-inactivation escape genes. Furthermore, it would be helpful to clarify whether the authors observed comparable reductions in H3K4me3 or chromatin accessibility in male and female samples, or whether sex-specific effects may reflect differential sensitivity to epigenetic dysregulation.

We thank the reviewer for this suggestion. We originally cited 2 studies that compared differential H3K4me3 levels between males & females, including escape from X inactivation. In this revised manuscript, we have added several additional studies about the epigenetic or neuroendocrine sex differences that may modulate H3K4 methylation patterns in the Discussion. As stated above, we do not have any evidence for sex bias reduction of H3K4 methylation in our data.

17. Weight Gain: Hypothalamic vs. cIN Contributions. While the manuscript focuses on hypothalamic disruptions leading to overeating/obesity, it is worth considering the possibility that altered cIN functionality could contribute to weight changes indirectly through mood or motivational deficits (for instance, depression-like or other behavioral changes can also predispose to weight gain). The causal link between hypothalamic gene expression changes and increases in body weight could be more strengthened or discussed.

This concern about definitively assigning causation of obesity & changes in body weight to the hypothalamus was also raised by Reviewer #2 point #1 above. We have addressed this issue in a new section of the Discussion.

18. Single nuclear RNA seq experiments: In subsequent single-cell analyses (Figure 7, Extended Data 6), the manuscript transitions from using an Ai9/tdTomato reporter to a Sun1-GFP reporter system, and thus to another mouse line. Please justify why a different fluorescent reporter (Sun1-GFP) was used for the single-nucleus analysis. Clarify that the experiment is directly comparable to the earlier Ai9 lines.

In the Ai9 mouse line, Cre recombination results in untethered tdTomato expression that fills the cell bodies, axons and dendrites. However, it does not label the nuclei, which we have confirmed via flow cytometry. To harvest nuclei from Nkx2.1-lineage cells, we switched to the Sun1-sfGFP/INTACT mouse line in which GFP is tethered to the Sun1 protein that is localized to the inner nuclear membrane. In this original paper (Mo et al. *Neuron* 2015), the authors demonstrate superb specificity (>98%) of the GFP marker in VIP-Cre and PV-Cre driver lines. We (Rhodes...Petros *Frontiers* 2024) and many other labs have used this mouse line to collect fluorescent nuclei for numerous downstream assays; it is widely accepted in the field. It is quite common to switch reporter mouse lines depending on the assay being performed. As both the Ai9 and Sun1-GFP cassettes are targeted to the Rosa26 locus, we are confident that the Nkx2.1-Cre;Ai9 and Nkx2.1-Cre;Sun1-GFP lines target the same population of cells; there is no evidence to the contrary.

19. What about Sex-Specific Differences in the snRNA-Seq/ATAC-Seq analysis? The documentation of how sex was incorporated into the overall experimental design and analysis pipeline is inconsistent also for the seq-data. Specifically, the figure legends, methods and main results section do not clearly state which datasets (e.g., snRNA-seq, snATAC-seq, WNN-integrated data, pseudotime, and WGCNA analyses) were generated from male, female, or mixed-sex animals. It is also not evident whether sex was treated as a covariate during data integration and normalization, or whether sex-specific analyses were systematically performed across all comparisons. Given the increasing recognition of sex as a critical biological variable and its relevance to the phenotypes described (e.g., seizure susceptibility), I recommend that the authors provide a clearer account of the sex composition of each dataset and clarify in the figure legends

and Methods whether data were pooled, stratified, or sex-balanced. This information is essential for interpreting the generalizability of the findings and for evaluating the extent to which observed effects are sex-dependent. Same accounts for Figure 8. Information on sex should also be mentioned in the results e.g. when describing altered expression in Mef2c.

We apologize for not being clearer in our descriptions about sex composition of our single cell datasets in the text. We clarified this issue by including the statement “Data is combined male & female samples unless stated otherwise” at the start of all relevant figures. During the data integration and normalization, sex wasn’t treated as a covariate. We have added a sentence in the ‘Multiome sequencing and analysis’ section in the methods stating that male & female samples were processed separately. And we have also clarified PV-predictive genes Maf and Mef2c were each decreased in one sex. A detailed breakdown of the number of cells, reads, UMI & genes (RNA) and fragments (ATAC) for each age, genotype and sex is included in Supplementary Table 1. We also apologize for not being clearer about the datasets used in each type of analyses; we have clarified this in the Figure legends, Methods and Results.

Separating male & female datasets for all downstream analyses significantly decreases overall analytic power. Therefore, we decided to only separate sexes for generating volcano plots & clusterProfiler Go plots to characterize sex-specific DEGs in E13.5 MGE, E13.5 hypothalamus, P60 PV+, P60 SST+, P60 hypothalamus neuron and astrocytes. These sex specific volcano plots are presented in Fig. 8-9 and Extended Data Figs. 9-13. We also generated Venn diagrams displaying the percentage of common and sex-specific DEGs for PV+ and SST+ cells in Extended Figs. 10-11 and hypothalamus in Extended Figs. 12-13. We have updated the figure legends and relevant sections of the Results accordingly, and included these sex-stratified datasets in Supplementary Table 1. In sum, we believe that separating sexes for a portion of single cell analyses provided greater depth to our dataset, highlighting similarities and differences in sex-specific DEGs. We thank the reviewer for this suggestion.

20. In the provided sex-specific comparisons, where differences in gene expression (e.g., potassium channel genes in PV+ cells) are reported, refer to the H3.3K4M HOM samples. However, it remains unclear whether sex-specific differences were also present in the adult cortical control (WT) samples, or whether these only emerged in the mutant condition. This distinction is important for interpreting whether the observed differences reflect baseline sexual dimorphism or genotype–sex interactions. For PV+ cells sex specific divergent regulation was shown (Extended Data 6H). However, for SST cells, the analysis is not shown sex-specifically. Are there now sex-specific effects? Illustrating these differences more systematically and clearly could significantly strengthen the discussion about sexual dimorphism.

We thank the reviewer for raising this important point. We agree that the comparison of DEGs between Hom male and Hom female is incomplete without the same comparison between WT male and female. Since we have now generated sex-specific volcano plots & clusterProfiler Go plots (see above), this comparison is no longer necessary and we have removed these panels (Old Fig S6H-I) and relevant text from the revised manuscript. We also annotated SST+ subtypes and revealed an increase of SST_Plpp4 subtype in Hom cortex (Extended Data Fig. 11A-C). The sex-specific volcano plots for SST population were also generated (Extended Data Fig. 11D-E).

21. *In Figure 7C and Extended Data 6A, more detail on cluster annotation (including specific marker genes) would help the reader interpret the results.*

Thank you for this suggestion. In old Figure 7C, the UMAP is color coded based on genotype (WT green, Hom red) as stated in the figure panel & legend. For old Extended Data Figure 6A, UMAP is color coded based on Seurat clusters, with each cluster defined in an unbiased manner by the Seurat algorithm. We added additional details on cluster annotation with Seurat::FindAllmarker function and relevant citations (Fig. 8E). In the revised manuscript, we have classified MGE cells into 6 progenitors and 7 postmitotic cell populations. As expected, 2 major interneurons type (PV-fated and Sst-fated) and a population of GABAergic projection neurons (Lhx8+) were identified. We also showed the expression of several specific marker genes in Extended Data Fig. 9C.

22. *Does the color code presented in Figure 8C refers to the UMAP in 8A (right)? Should be mentioned in the legend.*

The reviewer is correct; we have added a line in the legend of Figure 8C noting that the color scheme matches the putative hypothalamic nuclei in panel A. We also updated the hypothalamic cell type annotation based a recent publication (Kim...Blackshaw *Cell Reports* 2025), which provided additional depth to hypothalamic nuclei classification (Extended Data Fig. 12C). We have updated the figure legends and relevant sections of the Results and Methods accordingly.

23. *Extended Fig. 5G: why are the accessible peaks different/complementary in MGE versus Hypothalamus (Numerous decreased peaks in HOM-MGE but increased in hypothalamic samples)?*

In the MGE, there is good correlation between most differentially expressed genes (Fig. 8F and Extended Data Fig. 9D) and most differentially accessible peaks (Extended Data Fig. 8G) being downregulated in H3.3K4M Hom mice. However, in the hypothalamus, this relationship is less correlated, with a relatively even mix of DEGs increased and decreased (Fig. 9D & Extended Data Fig. 12E) while most differentially accessible peaks are increased (Extended Data Fig. 8G). Many genes associated with mitotic cell cycle phase transition were upregulated in the hypothalamus of Hom mice compared to WT, which correlated with the increased proportion of neural progenitors in the hypothalamus (Fig. 9D-E & Extended Data Fig. 12E-F). Additionally, our Western Blot data reveals that the downregulation of H3K4me3 is stronger in the hypothalamus compared to the MGE (Fig. 1E), so differential expression of the mutant allele could in part explain some of these differences. The reasons for these differences are not clear, but we did devote a paragraph in the discussion to this issue and highlight that these different trends can still result in similar neurogenic phenotypes.

24. *Hypothalamic findings: The hypothalamic analysis reveals substantial changes in cellular composition and organization in H3.3K4M Hom mice, including increased NPCs, loss of ARC/PVN populations, and expansion of astrocytes. The disorganization of tanycytes and myelin in the median eminence is striking and well-illustrated histologically. However, transcriptomic analysis of tanycytes was limited by low cell numbers, with few DEGs detected. The interpretation of specific transcriptional changes, such as the downregulation of Fgf14, would benefit from orthogonal validation (e.g., in situ hybridization or qPCR) to confirm this finding independently of the single-*

nucleus dataset. Strengthening the support for these changes would enhance the mechanistic link between H3K4me dysregulation and hypothalamic function.

The power of our Multiome analysis in the embryonic hypothalamus (>17,500 cells from both WT and Hom samples) allowed us to define progenitors and postmitotic putative hypothalamic nuclei with a high degree of confidence. We have included new heatmap analyses highlighting the proportional increased NPCs and decreased ARC & VMH nuclei in the H3.3K4M Hom hypothalamus (Extended Data Fig. 12D). And we have performed new immunostaining experiments in the E13.5 hypothalamus demonstrating strong upregulation of Ki67 in H3.3K4M Hom mice (Extended Data Fig. 12G), which is further evidence of increased hypothalamic NPCs in these mice.

However, the reviewer is correct that the power of our single cell analysis of tanycytes is low due to recovering very few tanycytes in these dissections. This is expected as tanycytes make up a very small percentage of Nkx2.1-lineage hypothalamic cells. In the revised manuscript, we include new FISH data on *Fgf14*, *Vimentin* and *tdTomato* in the hypothalamus. While validating *Fgf14* expression levels in tanycytes is challenging due to the low abundance of these cells along the 3rd ventricle, we did observe less expression of *Fgf14* in ARC cells in Hom mice (Fig. 10F), as showed below.

Additionally, Sst peptide is expressed in the ME where it is involved in growth hormone release and food intake. New immunostaining data reveals a significant decrease of SST protein in the ME of H3.3K4M mice (Fig. 10C-D). We also performed TUNEL staining at E13.5 hypothalamus and did not detect any differences in hypothalamic apoptosis between WT and mutant mice (Extended Data Fig. 12J). These new results extend our characterization on how reduction of H3K4me effects hypothalamic development.

25. The reduction in Olig2+ cells is an interesting finding and points to fate regulation by H3K4me3. The authors report a “corresponding decrease in myelin basic protein (MBP) in this region (Extended Data Fig. 2).” However, I could not find a quantification here. This would need to be provided for that statement.

The reviewer is correct that our statement “...corresponding decrease in [MBP]...” implies some form of analysis and statistical quantification that we did not perform. We are confident in this MBP reduction in CA3 of H3.3K4M mice as we see this change in multiple mice, but we did not quantify this result. Thus we have softened our claim in the text to “... an apparent decrease in [MBP]...”.

26. While the expansion of hypothalamic astrocytes in H3.3K4M Hom mice is clearly documented, the manuscript would benefit from additional developmental context clarifying how Nkx2.1-expressing progenitors contribute to glial lineages, particularly in the hypothalamus. Since Nkx2.1-lineage cells are widely known for giving rise to cortical interneurons, it would be helpful to note that these progenitors can also generate astrocytes and tanycytes in ventral forebrain regions, as supported by lineage tracing studies, e.g. in the introduction. This background would help interpret whether the observed glial expansion reflects an expected regional fate or a fate shift resulting from altered H3K4 methylation.

We have added a sentence in the first paragraph of the Results stating that glia cells and tanycytes are derived from Nkx2.1+ progenitor cells. Since we don't mention Nkx2.1 in the Introduction, we felt it was best to include this concept when Nkx2.1 is introduced in the Results.

27. The Discussion clearly integrates behavioral, transcriptomic, and cellular data to support a model of H3K4 methylation-dependent regulation of interneuron and hypothalamic development. However, the authors should consider including additional developmental context for the gliogenic potential of Nkx2.1+ progenitors, particularly in the hypothalamus, to better support the interpretation of astrocyte expansion.

As noted above, we now introduce the concept of Nkx2.1-lineage cells giving rise to glia (and tanycytes) in the first paragraph of the Results section. And we have added a section to the Discussion raising the possibilities that glia changes in the H3.3K4M mice could arise from gliogenesis of Nkx2.1-lineage cells vs. compensatory changes due to altered H3K4 methylation in Nkx2.1-lineage neurons.

28. Overall, this manuscript presents an ambitious and multidimensional study of H3K4 methylation in forebrain development and function. The integration of single-cell epigenomics, electrophysiology, and behavior is commendable, and the focus on both MGE-derived interneurons and hypothalamic circuits is very comprehensive. Addressing the points raised here — particularly the developmental issues, glial fate interpretation, sex-stratified analyses, and figure clarity — will help to enhance the robustness and interpretability of the findings.

We thank the reviewer for these positive comments. We have attempted to address the reviewer's concerns, which we feel have strengthened our manuscript.

Response to Reviewers

We thank the reviewers for their positive response to our revised manuscript. We address the remaining specific comments below, with reviewers' comments indented & italicized, and our response in blue font. All significant changes in the Revised Manuscript are in red text:

Reviewer #1:

The authors have answered all my questions.

We thank the reviewer for their previous suggestions, and we're pleased to have adequately addressed them in our revised manuscript.

Reviewer #2:

The authors have addressed all the concerns I raised in the previous version of the manuscript, providing explanations for those that could not be resolved experimentally. I believe the expanded characterization of the mutant mice, including the new western blots only included in the rebuttal letter, should be incorporated into a revised version of the manuscript, as it will make the paper clearer without requiring reference to the original paper describing the generation of the mouse mutants. I have no further comments for the authors.

We thank the reviewer for their previous suggestions, and we're pleased to have adequately addressed them in our revised manuscript. We have added the Western blots showing no change in overall H3.3 levels to Extended Data Figure 1A in the revised manuscript & updated the the Methods accordingly.

Reviewer #3:

The authors have made a substantial effort to address the comments of all reviewers, and they conducted additional analyses (e.g., PH3, Ki67, and migration assays), as well as corrections/additions of text throughout the manuscript. All of this indeed strengthen the manuscripts' conclusions. However, some issues remain that, in my view, should be clarified or further improved before acceptance.

We thank the reviewer for their previous suggestions, and we're pleased to have adequately addressed most of their concerns.

1. Figure quality and interpretation of the VZ signal: In Figure 1c, the ventricular zone (VZ) still lacks a clearly discernible cellular signal (it has not been replaced), which I interpreted as background rather than specific staining. The authors state that this is a long-standing model and provided a staining form the original publication. The new Figure 5a (very left panel) actually seems to shows a more convincing VZ signal, but the resolution and magnification are too low to allow a definitive assessment. I still would recommend to replace Figure 1c with a higher-quality image that clearly shows tdTom cells in the VZ.

In Figure 1C, we have added new images showing HA staining and Tom signal in MGE sections from WT and Hom mice that we hope alleviates the reviewer's concerns. We are confident that these images depict real HA and Tomato signal in VZ, SVZ and mantle cells throughout the MGE. This is not background signal, as there is a lack of Tomato signal in portions of the DAPI+ non-MGE tissue in the merged panel for both WT and Hom mice.

2. In matters of cortical layer markers: relying on DAPI alone is not sufficient for defining laminar boundaries, albeit I do agree that layer 4 is nicely visible. However, the border of layer 6 is not clearly visible without a marker, and this might impair the density measurements. Please also

compare 3A DAPI with 2F DAPI. In 2F the cortex seems measured to more “ventral” levels. So there is inconsistency in that.

This is basically the same point that the reviewer raises in point #5, so we address the reviewer’s general concern about measuring cortical thickness below.

3. I do agree with the authors, that interneuron–progenitor communication is not the primary focus of the study, which already is very comprehensive. Yet, I felt that the observed adult phenotypes could be better contextualized by discussing potential developmental correlates. Immunolabeling for TBR1 (deep-layer neurons at E13.5) or progenitor markers such as PAX6 and EOMES (TBR2) at embryonic stages could have helped to link developmental alterations (reduced invasion of cINs into the cortex at embryonic stages) to changes in cortical progenitor populations and to the later layer shifts. I wonder, whether the newly included KI67 staining and PH3 stainings, analyzed in the MGE, could also be analyzed in the cortex? Less numbers of cycling progenitors might already provide some insights. Even if such experiments are beyond the current scope, an explicit acknowledgment of this limitation and of possible mechanistic links would strengthen the discussion.

To address the reviewer’s concerns, we examined expression of Pax6 and Ki67 in the lateral cortex (since that’s where the largest reduction in migrating interneurons was observed) of E13.5 WT and Hom mice. We did not detect any obvious differences in either marker between WT and Hom cortex. We also quantified PH3+ cells in the E13.5 lateral cortex and found no significant difference between Hom and WT cortex. See Pax6 staining and PH3 quantification below. Thus, we see no evidence that the reduction of MGE-derived interneurons migrating into the cortex affects cell cycle dynamics and neurogenesis of dorsal cortical progenitors. We feel that additional analysis looking at the interaction between postmitotic interneurons and excitatory neurons is beyond the scope of this manuscript. We do not include these new results in our manuscript, but we have expanded on the non-cell autonomous effects of reduced interneuron migration in the Discussion (lines 531-535).

4. In that matters, the authors wrote in line 528: “The primary cause for reduced forebrain interneurons was decreased cortical migration, leading to smaller brains and reduced cortical thickness.” This causality is not proven. As cINs make up rather a minority, I doubt that their reduced density makes up the reduced thickness. The possible effect on cortical progenitors and excitatory neurons needs to be at least discussed.

The reviewer is correct that we have overstated our findings here, we do not know if decreased migration is the primary cause of smaller brains & reduced cortical thickness. We have modified this

sentence from “*The primary cause...*” to “*A likely cause for reduced forebrain interneurons was decreased cortical migration.*”, which is a more accurate interpretation of our data.

In our original Response to Reviewers, we noted that there is evidence in the literature that loss of MGE-derived interneurons is sufficient to reduce cortical thickness. Mutations in *Nkx2.1* and *Arx*, two genes restricted to the MGE, result in thinner cortices and smaller brains/microcephaly (reviewed in PMID 38094004). Whether this is solely from loss of interneurons, or due to additional changes and/or loss of excitatory neurons, is not clear. And based on the data above, we do not observe any change in cell cycle dynamics of dorsal cortical progenitors. We have expanded on this point in the Discussion (lines 531-535), and see the third point below.

5. Speaking of the cortical thickness reduction: it is a strong reduction in Figure 2F, whereas this reduction is absolutely not visible in Figure 3A, although it is both S1 of 3 weeks old mice. As there is only one scale bar shown for the WT in 3A, I assume it is the same scaling. The cortex thickness in WT and HOM is nearly the same...How do the authors explain that? Layer 1 in 2F WT seem very large. Are the regions chosen for measuring well controlled and matched between the genotypes?

First, the yellow dotted lines in Fig 3A were added so that readers could visualize the deep and superficial cortical boundaries. These are not the actual boundaries used in cell counts, which were carefully drawn based on DAPI staining. We concede that the layer VI boundary in Fig 3A is not very accurate and is confusing in comparison to old Fig 2F. We apologize for this confusion. We have removed the layer VI boundary in the DAPI image of Fig 3A since this boundary is not clearly visible in both WT and Hom DAPI panels (but see below). We kept the layer III/IV boundary line which is accurate.

Second, the sections used to measure S1 and V1 cortical thickness for WT and Hom mice were well matched based on brain makers. Conversely, the panels in Figure 3A were selected to highlight the changes in PV and SST interneurons. While the Fig 3A images are both from S1, they are not necessarily matched along this same axes. That being said, we note that the white matter is present in the bottom of the Hom section in Fig 3A (note the altered DAPI staining at the bottom of the Hom DAPI panel indicative of white matter) and is not present in the WT panel. This indicates that in Fig 3A, the Hom image is in fact thinner compared to the WT image, which is consistent with our quantification indicating a moderate but significant reduction in cortical thickness in Hom mice.

Third, we note that the difference in S1 and V1 cortical thickness between WT and Hom mice, while statistically significant, is actually a relatively small reduction (Average S1 thickness: WT 801 μ m, Hom 722 μ m, difference of 79 μ m or ~10% reduction. Average V1 thickness: WT 810 μ m, Hom 774 μ m, difference of 36 μ m or ~4.5% reduction). Related to the reviewer’s comments in point #4 above, while we cannot be certain if this decrease in cortical thickness is due solely to the reduction in cortical interneurons (see Discussion lines 531-535), it is certainly possible that these small but significant reductions in cortical thickness could be directly due to a reduction of MGE-derived interneurons.

Fourth, to alleviate the reviewer’s concerns about labeling cortical layers, we have reanalyzed cortical thickness of S1 and V1 on brain sections stained with Cux1 to define layers II-IV. This allowed us to measure the total cortical thickness, as well as the thickness of layer I, layers II-IV and layers V-VI. We find that the greatest change in cortical thickness is in layers II-IV, and we do not see any change in layer I thickness in either brain region. This new Cux1+ image and cortical thickness counts are now presented in Figure 2F (lines 131-132) and Extended Data Figure 1C-D.

Response to Reviewers

We thank the reviewers for their positive response to our revised manuscript. Reviewer #3 had two additional comments that we address below.

Reviewer #3:

The revised manuscript has improved in presentation and wording and is suitable for publication. For completeness, I would like to note two points that, in my view, are not fully addressed and should be kept in mind when interpreting the data.

We are pleased that our revised manuscript has addressed the reviewer's concerns and their opinion that our manuscript is now suitable for publication.

First, the pronounced thinning of upper cortical layers is unlikely to be explained by the reported changes in interneuron abundance, which appear rather modest. Given that later embryonic stages of corticogenesis were not examined, alterations in excitatory neuron development remain a plausible alternative explanation. Excluding this possibility would have required analyses at later developmental time points, when upper cortical layers are generated.

In the previous revision, we added the following sentences in the Discussion (lines 531-536) which we feel adequately addresses this issue, and we do not feel a need to adjust this statement:

There is evidence that disruption of interneuron fate and migration can lead to non-cell autonomous changes in excitatory cells, including altered neurogenesis and defects in cortical lamination and neuron activity^{101,102}. Whether the reduced cortical thickness in H3.3K4M mutant mice is solely from loss of interneurons or also arises from altered excitatory neuron development is not clear.

Second, if the reported differences in cortical thickness are robust and systematic, they would be expected to be detectable across sections. The discrepancy between the quantitative measurements shown in Figure 2F and the qualitative appearance of cortical sections in Figure 3 therefore remains difficult to reconcile. The authors' response clarifies their interpretation but does not fully resolve this issue. I note this point for completeness. These points do not detract from the overall quality or significance of the study.

We quantified a significant reduction of cortical thickness in 2 different cortical regions (S1 and V1) presented in Figure 1 and Supplementary Figure 1. In our previous response, we noted that the overall reduced thickness is ~10% in S1 and ~4.5% in V1. Whether this should be considered "...robust and systemic..." is up to interpretation, but these differences are statistically significant. While we understand the reviewer's point here, we do not believe that there is a significant discrepancy between Figure 2F and 3A as the reviewer describes for all the reasons mentioned in our previous Response to Reviewers. As the reviewer notes that this issue does not "...detract from the overall quality or significant of the study.", we have no further comment on this issue.